palaeontology/palaeontology

Notosuchia, Crocodylomorpha, Gondwana, Kem Kem, Mesozoic, Africa

**Author for correspondence:**
Cecily S. C. Nicholl
e-mail: cecily.nicholl@ucl.ac.uk

# A second peirosaurid crocodyliform from the Mid-Cretaceous Kem Kem Group of Morocco and the diversity of Gondwanan notosuchians outside South America

Cecily S. C. Nicholl[1], Eloise S. E. Hunt[2,3], Driss Ouarhache[4] and Philip D. Mannion[1]

[1]Department of Earth Sciences, University College London, Gower Street, London WC1E 6BT, UK
[2]Department of Life Sciences, Natural History Museum, Cromwell Road, London SW7 5BD, UK
[3]Science and Solutions for a Changing Planet DTP, and the Department of Life Sciences, Imperial College London, South Kensington Campus, London SW7 2AZ, UK
[4]Laboratoire Géosystèmes, Environnement et Développement Durable, Département de Géologie, Faculté des Sciences Dhar El Mahraz, Université Sidi Mohamed Ben Abdellah, BP 1796, Atlas 30 000, Fès, Morocco

CSCN, 0000-0003-2860-2604; PDM, 0000-0002-9361-6941

Notosuchians are an extinct clade of terrestrial crocodyliforms with a particularly rich record in the late Early to Late Cretaceous (approx. 130–66 Ma) of Gondwana. Although much of this diversity comes from South America, Africa and Indo-Madagascar have also yielded numerous notosuchian remains. Three notosuchian species are currently recognized from the early Late Cretaceous (approx. 100 Ma) Kem Kem Group of Morocco, including the peirosaurid *Hamadasuchus rebouli*. Here, we describe two new specimens that demonstrate the presence of at least a fourth notosuchian species in this fauna. *Antaeusuchus taouzensis* n. gen. n. sp. is incorporated into one of the largest notosuchian-focused character-taxon matrices yet to be compiled, comprising 443 characters scored for 63 notosuchian species, with an increased sampling of African and peirosaurid species. Parsimony analyses run under equal and extended implied weighting consistently recover *Antaeusuchus* as a peirosaurid notosuchian, supported by the presence of two distinct waves on the dorsal dentary surface, a surangular which laterally overlaps the dentary above the mandibular fenestra, and a relatively broad mandibular symphysis. Within Peirosauridae, *Antaeusuchus* is recovered as the sister taxon of *Hamadasuchus*. However, it differs from

*Hamadasuchus* with respect to several features, including the ornamentation of the lateral surface of the mandible, the angle of divergence of the mandibular rami, the texture of tooth enamel and the shape of the teeth, supporting their generic distinction. We present a critical reappraisal of the non-South American Gondwanan notosuchian record, which spans the Middle Jurassic–late Eocene. This review, as well as our phylogenetic analyses, indicate the existence of at least three approximately contemporaneous peirosaurid lineages within the Kem Kem Group, alongside other notosuchians, and support the peirosaurid affinities of the 'trematochampsid' *Miadanasuchus oblita* from the Maastrichtian of Madagascar. Furthermore, the Cretaceous record demonstrates the presence of multiple lineages of approximately contemporaneous notosuchians in several African and Madagascan faunas, and supports previous suggestions regarding an undocumented pre-Aptian radiation of Notosuchia. By contrast, the post-Cretaceous record is depauperate, comprising rare occurrences of sebecosuchians in north Africa prior to their extirpation.

## 1. Introduction

Today's crocodylians are the remnants of a once much more diverse and widespread clade, Crocodyliformes [1–5]. One extinct group, Notosuchia, comprises a morphologically diverse, speciose clade of terrestrial crocodyliforms [2,6,7]. Often noted to exhibit bizarre bauplans relative to other crocodyliforms, notosuchians include species characterized by features such as 'pug-nosed' and 'duck'-like snouts (e.g. [8–10]), elongate limbs indicative of a parasagittal posture (e.g. [11–14]), mammal-like heterodont dentition (e.g. [8,15–17]) and even herbivory (e.g. [18,19]). Notosuchians have predominantly been recovered from Gondwanan landmasses, especially South America (e.g. [2,6,20]), from which more than 70% of species have been discovered [7]. Although the group had its highest apparent (i.e. 'raw' number of) species diversity in the Middle–Late Cretaceous (approx. 120–66 Ma) [7,21], notosuchians survived until the middle Miocene (approx. 12 Ma) [22–24], with putative remains extending their record back to the Middle Jurassic (approx. 168 Ma) [25].

Despite severe and pervasive under-sampling of fossiliferous localities relative to most other continents [26], diverse assemblages of extinct crocodyliforms have been discovered from several spatio-temporal intervals in Africa (e.g. [27–30]), including those yielding notosuchians. One such interval is represented by the 'middle' Cretaceous Kem Kem Group, a series of highly fossiliferous continental strata exposed in the east of Morocco along its border with Algeria, forming the northwestern edge of the Sahara Desert [31–37] (figure 1). The Kem Kem Group is generally considered to be either late Albian or Cenomanian (approx. 105–94 Ma) (e.g. [38]), with the most recent stratigraphic reappraisal favouring this younger age [37]. A diverse vertebrate fauna has been recovered from the Kem Kem Group, including sharks, bony fishes, lissamphibians, turtles, squamates, pterosaurs, non-avian dinosaurs and crocodyliforms [28,34–37,39].

The Kem Kem crocodyliforms comprise the neosuchians *Aegisuchus witmeri* [40], *Elosuchus cherifiensis* [41,42] and *Laganosuchus maghrebensis* [28], as well as three notosuchians [37]. The first of these notosuchians to be named, the peirosaurid *Hamadasuchus rebouli* [43], was erected based on a fragmentary dentary. Several specimens have since been referred to this taxon, including a nearly complete cranium and lower jaws [37,44–46]. Sereno & Larsson [28] described a second Kem Kem notosuchian species, the small-bodied uruguaysuchid *Araripesuchus rattoides*, which is currently known from several dentaries [37]. The third notosuchian species to be described, the candidodontid *Lavocatchampsa sigogneaurussellae* [38], is known from a small partial skull with unusual mammal-like multicuspid teeth. Ibrahim *et al.* [37] suggested that multicuspid crocodyliform teeth described by Larsson & Sidor [44] might represent additional notosuchian taxa. Finally, Ibrahim *et al.* [37] also noted anatomical differences between the type and referred material of *Hamadasuchus* that could indicate yet higher crocodyliform diversity in the Kem Kem Group.

Here, we describe new notosuchian remains from the Kem Kem Group of Morocco that support Ibrahim *et al.*'s [37] suggestion of higher crocodyliform diversity in this fauna. We test the phylogenetic position of these new specimens in an expanded version of an existing dataset. Finally, we provide a critical reappraisal of the Gondwanan record of non-South American notosuchians, in which we reassess the group's diversity through time and space.

Institutional abbreviations—BSPG, Bayerische Staatssammlung für Paläontologie und Geologie, Munich, Germany; CMN (formerly NMC), Canadian Museum of Nature, Ottawa, Canada; MDE,

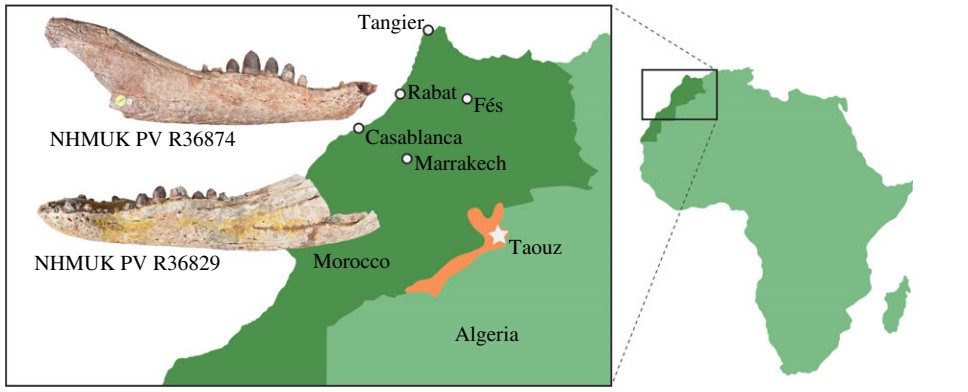

**Figure 1.** Map showing the locality of the new fossil remains. White star indicates the approximate geographical position of *Antaeusuchus taouzensis* n. gen. n. sp. (NHMUK PV R36829 and R36874) within the Kem Kem Group of Morocco.

Musée des Dinosaures, Espéraza, France; MNHM, Muséum national d'Histoire naturelle, Paris, France; NHMUK, Natural History Museum, London, UK; ROM, Royal Ontario Museum, Toronto, Canada.

## 2. Systematic palaeontology

Crocodylomorpha Walker, 1970
Crocodyliformes Hay, 1930 (*sensu* Clark in Benton and Clark, 1988)
Mesoeucrocodylia Whetstone and Whybrow, 1983
Notosuchia Gasparini, 1971
Peirosauridae Gasparini, 1982
*Antaeusuchus taouzensis* gen. et sp. nov.
urn:lsid:zoobank.org:act:62C4F680-CCFD-41CF-A328-8552E7B086C0
urn:lsid:zoobank.org:act:13959FDD-B1B1-472D-B2C6-DBD935721892

Etymology—Genus name after the giant *Antaeus* from Berber and Greek mythology, who is said to be buried at Msoura in northern Morocco, and *suchus*, from the Greek *souchos*, meaning crocodile. Species name after the township *Taouz* from where the holotype and paratype specimens were recovered.

Holotype—NHMUK PV R36829: paired mandibles, comprising an essentially complete left dentary and splenial, along with a partial angular and surangular, in articulation with the anterior portion of the right dentary and splenial.

Paratype—NHMUK PV R36874: a partial right mandible, comprising an incomplete dentary, surangular and angular.

Locality and horizon—Near to Jebel Beg'aa, Taouz township, Errachidia Province, eastern Morocco. The specimens were commercially collected and recovered from unspecified beds within the Cenomanian (lower Upper Cretaceous) Kem Kem Group.

Diagnosis—A crocodyliform characterized by the following unique combination of features: (ii) wide divergence angle (40–45°) of the mandibular rami; (ii) dorsal margin of dentary sinusoidal with two distinct waves; (iii) relatively unornamented surface texture of dentary adorned with narrow, shallow ridges; (iv) ventrolateral dentary surface anterior to mandibular fenestra transversely compressed and vertical; (v) dentary extends posteriorly beneath the mandibular fenestra; (vi) anterior alveoli of dentary strongly procumbent; (vii) concavity for the reception of the enlarged maxillary tooth lateral to the 7th alveolus of the dentary; (viii) splenial forming approximately 40% of the total mandibular anteroposterior length; (ix) surangular overlaps dentary above the mandibular fenestra; (x) rugose tooth enamel formed by anastomosing grooves and ridges; (xi) enlarged 4th and 13th dentary teeth; (xii) tooth margins in posterior region of the dentary toothrow with denticulate carinae formed by homogeneous and symmetrical denticles with a sharp cutting edge; and (xiii) sub-triangular dentary tooth crowns (in lateral view) with a gently curved apex.

## 3. Description

After a detailed description and comparison of the two specimens, we consider both NHMUK PV R36874 and R36829 to belong to the new species, *Antaeusuchus taouzensis*, and as such they are

described together. In instances where the feature being described is preserved in only one specimen, the relevant museum accession number is provided.

## 3.1. Preservation

The preserved parts of both specimens are undistorted and in good condition, such that small-scale morphological details are still visible. Damage is restricted mainly to the teeth, several of which are missing.

## 3.2. General shape

The anterior region of the mandible is characterized by a broad, 'U'-shaped symphysis that forms at least one-quarter of the total anteroposterior mandibular length. Each mandibular ramus diverges at an angle of approximately 22° from the sagittal midline. The ramus remains approximately straight along the majority of its preserved length, curving very slightly medially close to its posterior margin (visible on the left side of NHMUK PV R36829). The anterior half of the mandibular dorsal margin is characterized by two distinct 'waves', whereas the strongly sloping posterior half is largely straight, with the dentary increasing in dorsoventral height towards the surangular.

## 3.3. Dentary

The dentary is anteroposteriorly elongate, and its lateral, ventral and dorsal surfaces are sculpted by neurovascular foramina and vermiform grooves. On the lateral surface, the foramina are the largest and most numerous in the anterodorsal region of the snout. In the middle region of the snout, a series of short grooves run anteroposteriorly along the lateral surface, approximately 10 mm ventral to the toothrow. These grooves meet an anteroposteriorly elongate groove that extends to the dorsal suture of the dentary and surangular process (figures 2 and 3). Another prominent, anteroposteriorly elongate vascular groove runs from the anteriormost point of the mandibular fenestra to the level of the posterior tip of the toothrow.

In lateral view, the dentary has a sinusoidal dorsal margin composed of two distinct waves. The most anterior wave spans teeth 1–6, whereas the most posterior wave is dorsally raised between teeth 9 and 15. The dorsoventrally tallest region of each wave corresponds with the position of dentary teeth 4 and 13, the posteriormost of which is more dorsally elevated than the anterior (figures 2 and 3). The dentary forms the anterior region of a relatively wide mandibular symphysis (figure 4), the dorsal surface of which is very mildly concave. In dorsal view, the midline dentary suture extends posteriorly to the level of the 8th tooth (figure 4). The bone in this region is relatively unornamented, with the exception of a series of foramina immediately adjacent to the toothrow. On the ventral surface of the mandibular symphysis, the medial dentary suture extends posteriorly to a level between the 7th and 8th teeth. A concavity is situated lateral to the 5th–10th teeth, most likely for the reception of an enlarged maxillary tooth. The posterior region of the lateral dentary surface dorsal to the mandibular fenestra is divided into two major acute posterior processes, separated by an anterior process of the surangular (figure 2). The ventral-most dentary extension forms the anterodorsal margin of the external mandibular fenestra and is dorsoventrally wide, forming approximately three-quarters of the dorsoventral height of the mandible at the same level. The dorsal-most process is much narrower; its dorsal and ventral margins converge posteriorly to form an acute angle. An anteroposteriorly short dentary process is situated ventral to the external mandibular fenestra, although this does not contact the fenestral border. In NHMUK PV R36874, the splenial is not preserved, exposing the dentary's medial surface (figure 3).

The dentary has 18 tooth positions. On the left side of NHMUK PV R36829, whole or partial teeth are preserved in alveoli 1–16, whereas 17–18 are empty. On the right side of the specimen, 10 alveoli are preserved, with whole or partial teeth preserved in all but one (alveolus 9). Although the anterior region of NHMUK PV R36874 has broken away, the first preserved alveolus is large, and is assumed to be the 4th in the series. Whole or partial teeth are present in alveoli 5–6, 8 and 10–16 in NHMUK PV R36874. The largest tooth is the 13th, followed closely by the 4th, 11th and 12th, which are approximately equidimensional in their circumference. In dorsal view, the tooth row is slightly sinusoidal, with lateral waves corresponding to the position of the 4th and 13th teeth (figure 4). Although not fully preserved in either specimen, the anteriormost two teeth appear to be procumbent.

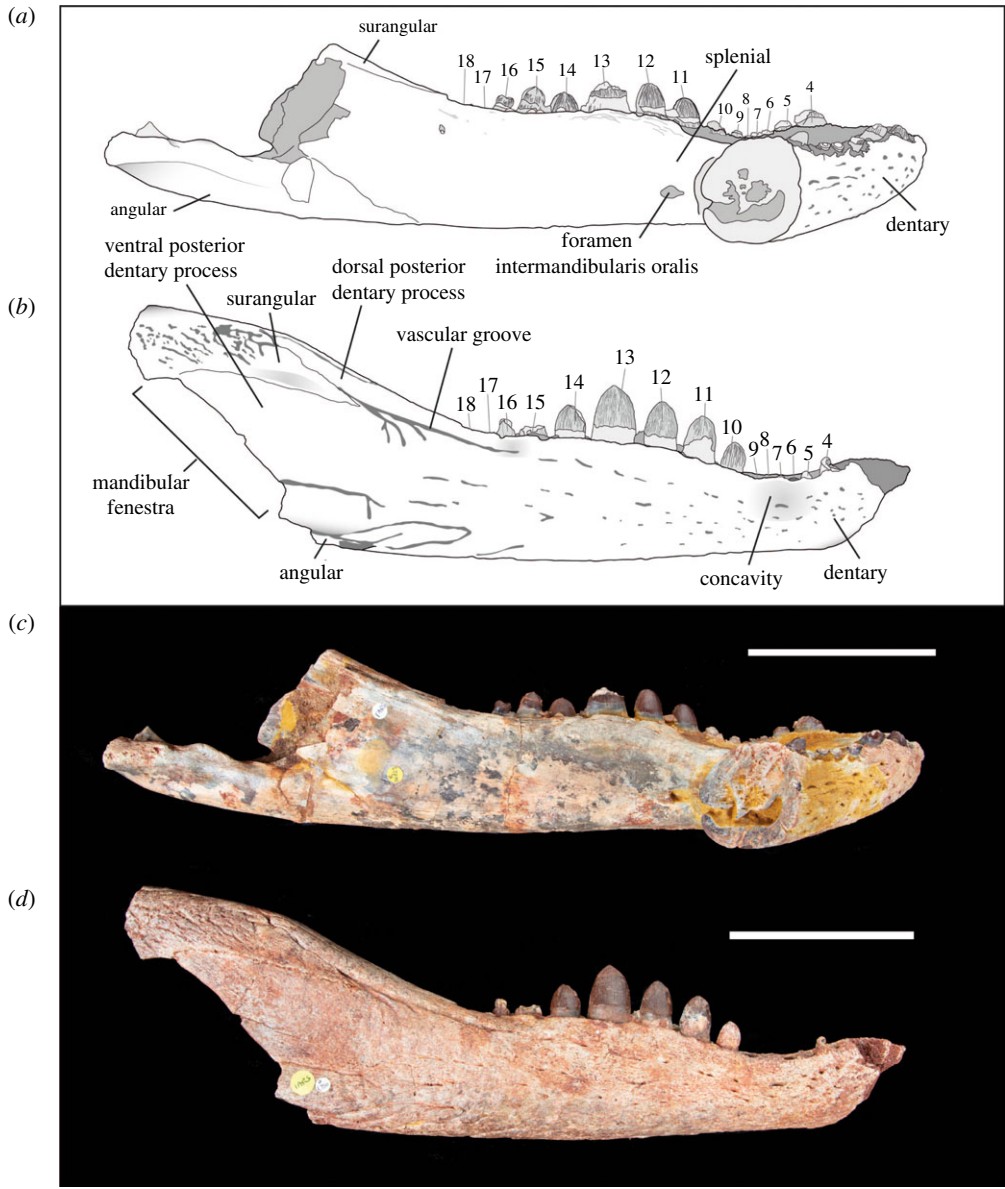

**Figure 2.** Line drawings and photographs of *Antaeusuchus taouzensis* n. gen. n. sp. (NHMUK PV R36829 (*a,c*) and NHMUK PV R36874 (*b,d*)) in right lateral view. Scale bar represents 100 mm.

Dentary teeth 3–10 project slightly anterolaterally. All of the dentary teeth are closely arranged, without the presence of diastemas.

## 3.4. Splenial

The splenials are only preserved in NHMUK PV R36829. They participate in a relatively wide mandibular symphysis (table 1) and occupy approximately 38% of the anteroposterior symphyseal length on the dorsal surface of the mandible, extending anteromedially to the position of the 8th alveolus (figure 4). On the dorsal surface of the symphysis, the splenial-dentary suture diverges gradually from the sagittal midline. This suture is slightly concave until the 11th tooth, from which point it is parallel to the tooth row. A line of small foramina run parallel to the toothrow along the lateral margin of the dorsal surface of the splenial. On the ventral surface of the mandible, the splenial occupies approximately 31% of the anteroposterior length of the symphysis, and it extends anteriorly to the position of the 9th tooth. The ventromedial splenial surface of the mandibular symphysis is dorsally displaced relative to the lateral margin. A posterior peg is located on the ventromedial surface of the symphysis (figure 4). The splenial is transversely thin and dorsoventrally

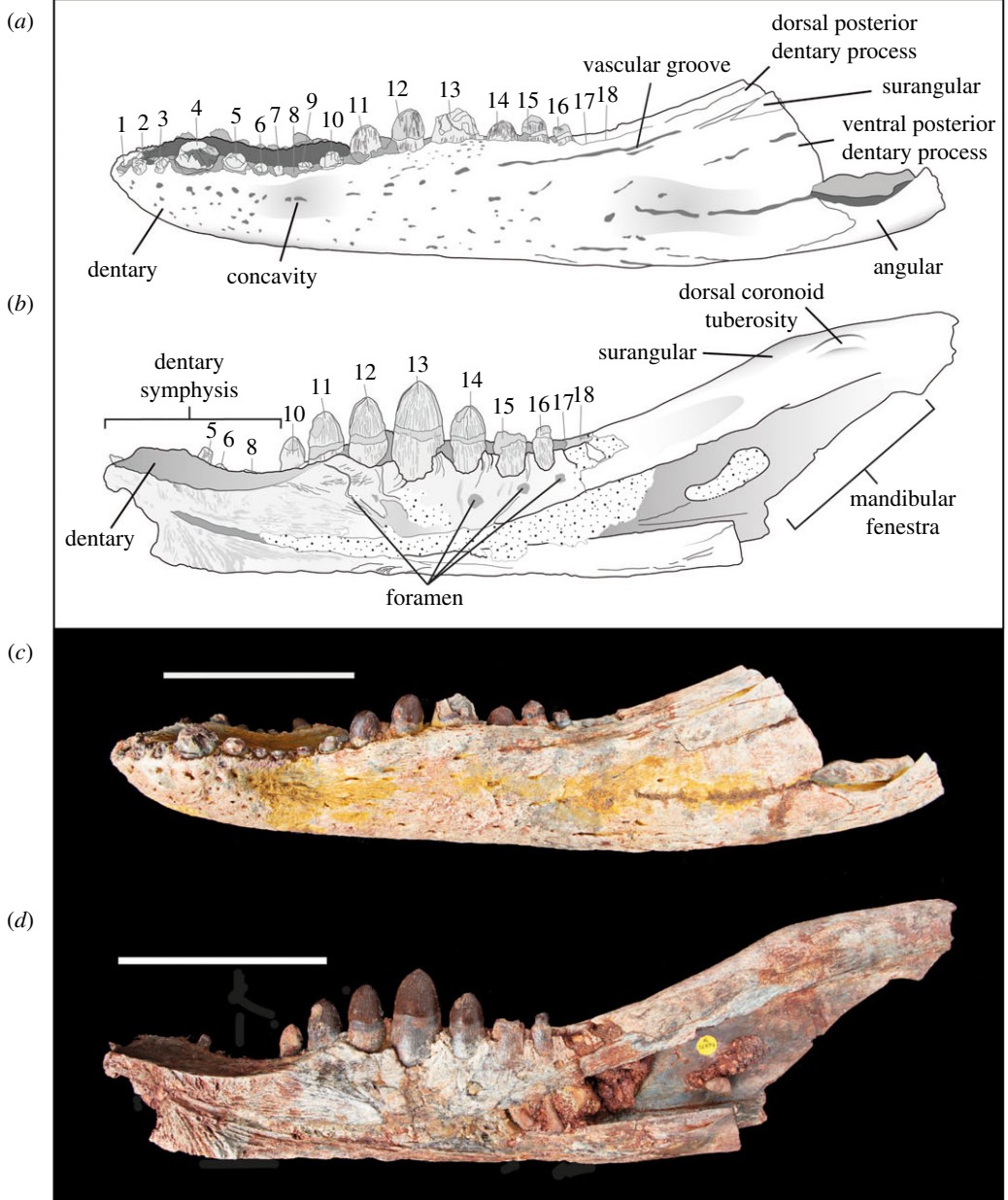

**Figure 3.** Line drawings and photographs of *Antaeusuchus taouzensis* n. gen. n. sp. (NHMUK PV R36829 (*a*,*c*) and NHMUK PV R36874 (*b*,*d*)) in left lateral view. Scale bar represents 100 mm.

tall, covering the inner surface of the mandibular ramus from the ventral margin of the dentary to the lingual alveolar groove. Positioned just posterior to the mandibular symphysis, on the medial surface of the splenial, is an opening, likely homologous to the intermandibularis oralis of living crocodylians [47]. This is elliptical, such that its anteroposterior length is approximately twice that of its dorsoventral height.

## 3.5. Surangular

The surangular is more completely preserved in NHMUK PV R36829, extending from the posterior margin of the toothrow to its broken posterior margin at the dorsal-most region of the mandibular fenestra. Its lateral surface is covered with interconnected shallow grooves. Of the surangular's two acute anterior processes, the most anteroposteriorly elongate is located on the dorsal and medial surfaces of the mandible and extends to the posterior margin of the toothrow (figure 4). A second anterior process is present on the dorsal region of the lateral mandibular surface (figures 2 and 3).

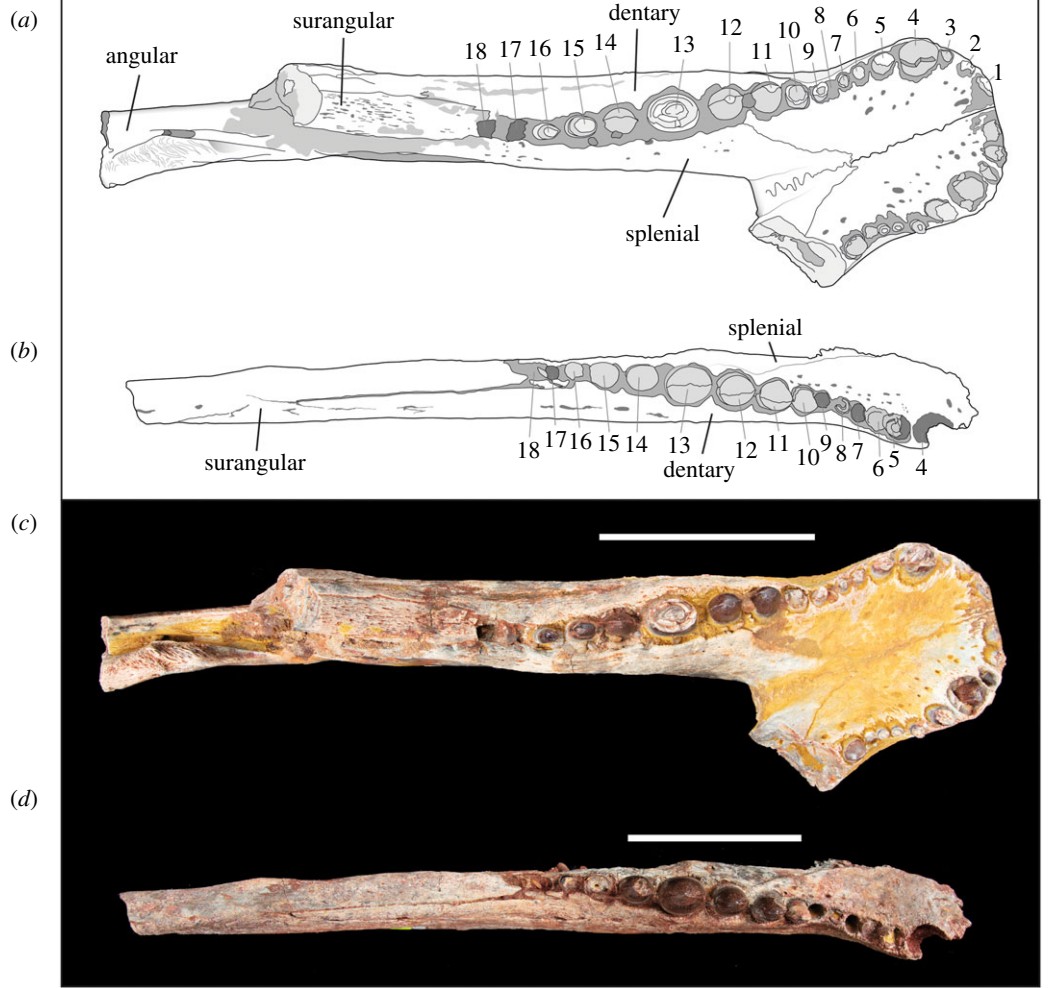

**Figure 4.** Line drawings and photographs of *Antaeusuchus taouzensis* n. gen. n. sp. (NHMUK PV R36829 (*a,c*) and NHMUK PV R36874 (*b,d*)) in dorsal view. Scale bar represents 100 mm.

**Table 1.** Mandibular measurements of the holotypic specimen (NHMUK PV R36829) of *Antaeusuchus taouzensis* n. gen. n. sp.

|  | dimension (mm) |
| --- | --- |
| maximum mandibular anteroposterior length | 415 |
| maximum mandibular symphysis anteroposterior length | 123 |
| maximum mandibular symphysis mediolateral width | 83 |
| maximum dentary anteroposterior length | 371 |
| maximum dorsoventral height of mandibular ramus | 92 |

Approximately halfway between the anterior margin of the mandibular fenestra and the posterior margin of the toothrow, the dorsal and ventral margins of this second anterior process meet anteriorly to form a sub-triangular tip. An anteroposteriorly elongate dorsal coronoid tuberosity protrudes from the dorsomedial surangular surface, running anteroposteriorly for a distance of approximately 30 mm; its anterior margin is at the same level as the posteriormost point of the posterodorsal dentary process. The surangular forms the dorsal-most margin of the mandibular fenestra.

## 3.6. Angular

Albeit highly incomplete, the angular is best preserved in NHMUK PV R36874. The angular has an elongate anterior process that extends along the ventromedial surface of the mandible to the level of

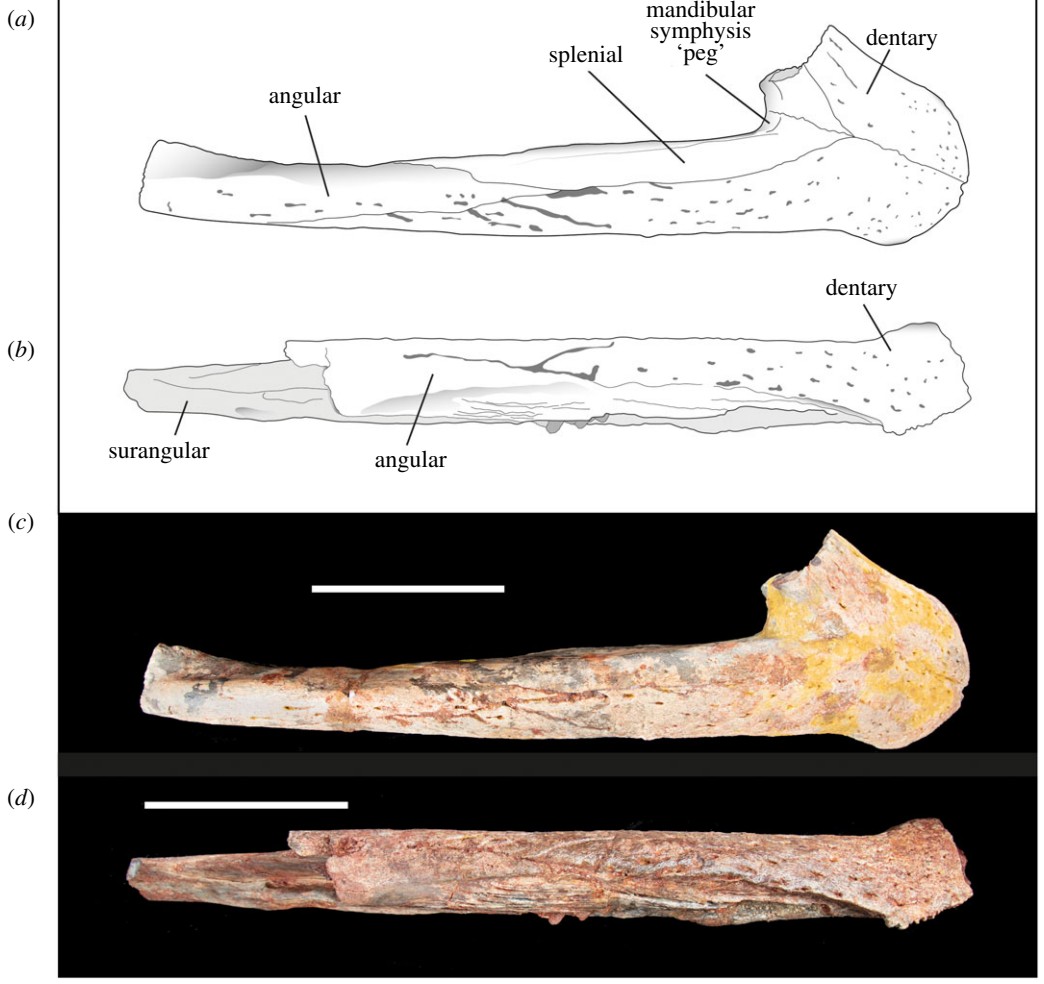

**Figure 5.** Line drawings and photographs of *Antaeusuchus taouzensis* n. gen. n. sp. (NHMUK PV R36829 (*a*,*c*) and NHMUK PV R36874 (*b*,*d*)) in ventral view. Scale bar represents 100 mm.

the 14th dentary tooth (figure 5). A second, smaller anterior process is present on the lateral surface of the skull (figure 3). This extends to the anterior margin of the mandibular fenestra, such that the angular forms the latter's entire ventral margin. The posteriormost preserved region of the angular projects laterally, forming a prominent ventrolateral ridge beneath the mandibular fenestra.

## 3.7. Mandibular fenestra

Although not fully preserved in either specimen, the mandibular fenestra can be inferred to be large and anteroventrally–posterodorsally elongate, as indicated by its extensive, straight anterodorsal margin in NHMUK PV R36874 (figures 2 and 3). Its anteriormost margin is positioned at approximately the same level as the posteriormost extension of the dorsal-most dentary process; however, the posterior fenestral margin is not preserved in either specimen.

## 3.8. Dentition

In the anterior region of the snout, the approximately circular alveoli suggest that the teeth are essentially conical; however, those towards the posterior of the toothrow (from the 10th tooth posteriorly) become more labiolingually compressed (table 2). More extreme labiolingual flattening is present on the anterior and posterior margins of all preserved teeth (figure 6). These labiolingually flattened margins are adorned with denticulated carinae forming the anteriormost and posteriormost cutting edges of the teeth. The denticles are small and subtle, showing no significant size variation along the carinae (figure 6). There are approximately 35–40 denticles per 10 mm. All preserved teeth are covered by a

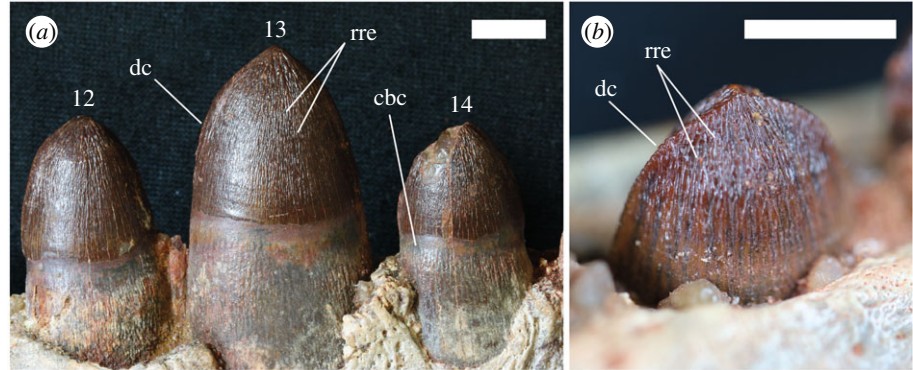

**Figure 6.** Teeth of *Antaeusuchus taouzensis* n. gen. n. sp. (*a*) Teeth 12–14 of NHMUK PV R36874. (*b*) Tooth 14 of NHMUK PV R36829. cbc, constricted base of crown; dc, denticulated carina; rre, ridged rugose enamel. Scale bar represents 10 mm.

**Table 2.** Tooth and alveolus measurements of the holotypic specimen (NHMUK PV R36829) of *Antaeusuchus taouzensis* n. gen. n. sp.

| tooth position | apicobasal length (mm) | alveolar dimension (mm) | | | |
| --- | --- | --- | --- | --- | --- |
| | | left mandible, anteroposterior length | right mandible, anteroposterior length | left mandible, mediolateral width | right mandible, mediolateral width |
| 1 | — | 11.0 | 11.5 | 14.1 | 14.1 |
| 2 | — | 7.4 | 7.0 | 8.7 | 8.4 |
| 3 | — | 5.0 | 5.0 | 5.0 | 5.4 |
| 4 | 12.1 | 17.5 | 12.5 | 16.0 | 12.0 |
| 5 | — | 10.0 | 10.0 | 10.0 | 9.5 |
| 6 | — | 5.6 | 5.5 | 6.5 | 7.0 |
| 7 | — | 4.6 | 5.2 | 6.8 | 6.1 |
| 8 | — | | 5.1 | | 6.0 |
| 9 | — | 6.6 | 5.4 | 6.5 | 6.2 |
| 10 | 11.0 | 11.2 | 11.1 | 10.5 | 10.0 |
| 11 | 18.0 | 14.2 | — | 13.0 | — |
| 12 | 19.0 | 15.6 | — | 12.2 | — |
| 13 | 16.0 | 23.1 | — | 16.2 | — |
| 14 | 10.0 | 14.4 | — | 11.5 | — |
| 15 | 13.0 | 13.0 | — | 8.5 | — |
| 16 | 8.0 | 10.0 | — | 7.0 | — |
| 17 | — | 7.0 | — | 6.0 | — |
| 18 | — | 8.0 | — | 7.0 | — |

layer of red-brown enamel upon which apicobasal striations are evident around the whole circumference of the tooth. There are approximately 3–4 bifurcating striations per 1 mm, giving the enamel a wrinkled appearance.

# 4. Phylogenetic analysis and results

## 4.1. Dataset and analytical approach

Specimens NHMUK PV R36829 and R36874 were combined as one operational taxonomic unit (OTU), *Antaeusuchus taouzensis*, into a character-taxon matrix (CTM) sampling a large number of

crocodyliforms, with particular emphasis on notosuchians. This matrix was originally published by Pol *et al.* [6] and has since formed the underlying dataset for phylogenetic analysis in a number of studies, with each one making minor additions and/or revisions to taxa and/or characters. Unfortunately, many of these iterations have occurred in parallel, rather than representing a continuous series of revisions to one dataset, meaning that there is no single dataset incorporating all of these changes to the original Pol *et al.* [6] data matrix. Here, we united many of these 'daughter' matrices, using that of Martínez *et al.* [48] as a starting point. The latter is a successive iteration of the data matrices of Leardi *et al.* [49] and Fiorelli *et al.* [50], which emanated from that of Pol *et al.* [6]. We included two additional characters, following Leardi *et al.* [51], and revised 20 existing character scores based on observations from recent studies [46,52–58] and personal observations (see Appendix for documentation of changes).

We incorporated notosuchians from parallel daughter matrices, using scores presented in those datasets, and a review of the literature. These consist of *Razanandrongobe sakalavae* from the Bathonian (Middle Jurassic) of Madagascar [25,59], the probable peirosaurids *Bayomesasuchus hernandezi* [58] and *Barrosasuchus neuquenianus* [60] from the early Late Cretaceous (Turonian and Santonian, respectively) of Argentina, the sphagesaurid *Caipirasuchus mineirus* from the late Campanian–early Maastrichtian (latest Cretaceous) of Brazil [61], and the sebecid *Ogresuchus furatus* from the early Maastrichtian of Spain [62]. We also expanded the sampling of putative peirosaurids that had not previously been incorporated into iterations of the Pol *et al.* [6] data matrix via the inclusion of *Rukwasuchus yajabalijekundu* from the Late Cretaceous of Tanzania [63] and *Miadanasuchus oblita* from the Maastrichtian of Madagascar [64]. The OTU for *H. rebouli* followed previous iterations of this data matrix, although we made a small number of character state changes (see Appendix). The resultant data matrix consists of 121 OTUs scored for 443 characters, including 63 putative notosuchian taxa. *Antaeusuchus taouzensis* could be scored for 51 of these characters.

The data matrix was analysed under maximum parsimony using the 'Stabilize Consensus' option in the 'New Technology Search' in TNT v. 1.5 [65]. The search was executed using sectorial searches, drift and tree fusing, and the consensus was stabilized five times with a factor 75, prior to using the resultant trees as the starting trees for a 'Traditional Search' using Tree Bisection-Reconstruction. Subsequently, a strict consensus tree was calculated. We applied two different weighting schemes, using equal weighting (EQW) and extended implied weighting (EIW). Shown to perform well on morphological datasets [66], EIW downweights homoplastic characters in relation to their average homoplasy, while reducing the possible impact of missing data [67]. The concavity constant, represented by the $k$-value, denotes the strength of downweighting, with lower values having been shown to downweight homoplastic characters more severely than higher values [66]. Following analytical protocols in recent analyses of neosuchians [68–71], we applied EIW to notosuchians for the first time, using $k$-values of 8 and 12. Characters with missing entries were downweighted faster assuming 50% of the homoplasy of observed entries, and weighting strength did not exceed 5 times that of characters with no missing entries. Forty-three characters representing nested sets of homologies were ordered (1, 3, 6, 10, 23, 37, 43, 44, 45,49, 65, 67, 69, 71, 73, 77, 79, 86, 90, 91, 96, 97, 105, 116, 126, 140, 142, 143, 149, 167,182, 187, 193, 197, 226, 228, 279, 339, 356, 357, 364, 368, 401). Character 5 was made inactive due to 'dependence with the modified definition of character 6' ([6]: supplementary information, p. 3). Following the identification of problematic, unstable taxa by Pol *et al.* [6], confirmed by our preliminary searches, three species known from fragmentary remains were excluded from our analyses *a priori* (i.e. *Coringasuchus anisodontis*, *Pabwehshi pakistanensis* and *Pehuenchesuchus enderi*). The character list and data matrix are provided as nexus and tnt files (electronic supplementary material), with stored settings for assigning characters as ordered and inactive.

## 4.2. Results

Under EQW, our analysis produced 11 520 trees with a tree length of 1778 steps. The overall tree topology is broadly consistent with the analyses of Pol *et al.* [6] and subsequent iterations. Notosuchia comprises the main bifurcation into Ziphosuchia (plus *Candidodon itapecuruense* and *Libycosuchus brevirostris*), and a clade in which Uruguaysuchidae is recovered as the sister taxon of Mahajangasuchidae + Peirosauridae (following the recent phylogenetic definition of [72] (see below)) (figure 7). Although *Peirosaurus torminni* is not included in our data matrix, *Uberabasuchus terrificus* has been consistently recovered as a close relative, with some authors regarding the latter as a junior synonym of the former (e.g. [46,57]). As such, we regard the *Uberabasuchus* OTU as a proxy for *Peirosaurus* in terms of identifying Peirosauridae. Bremer values are generally low across the tree, ranging from 1 to 3.

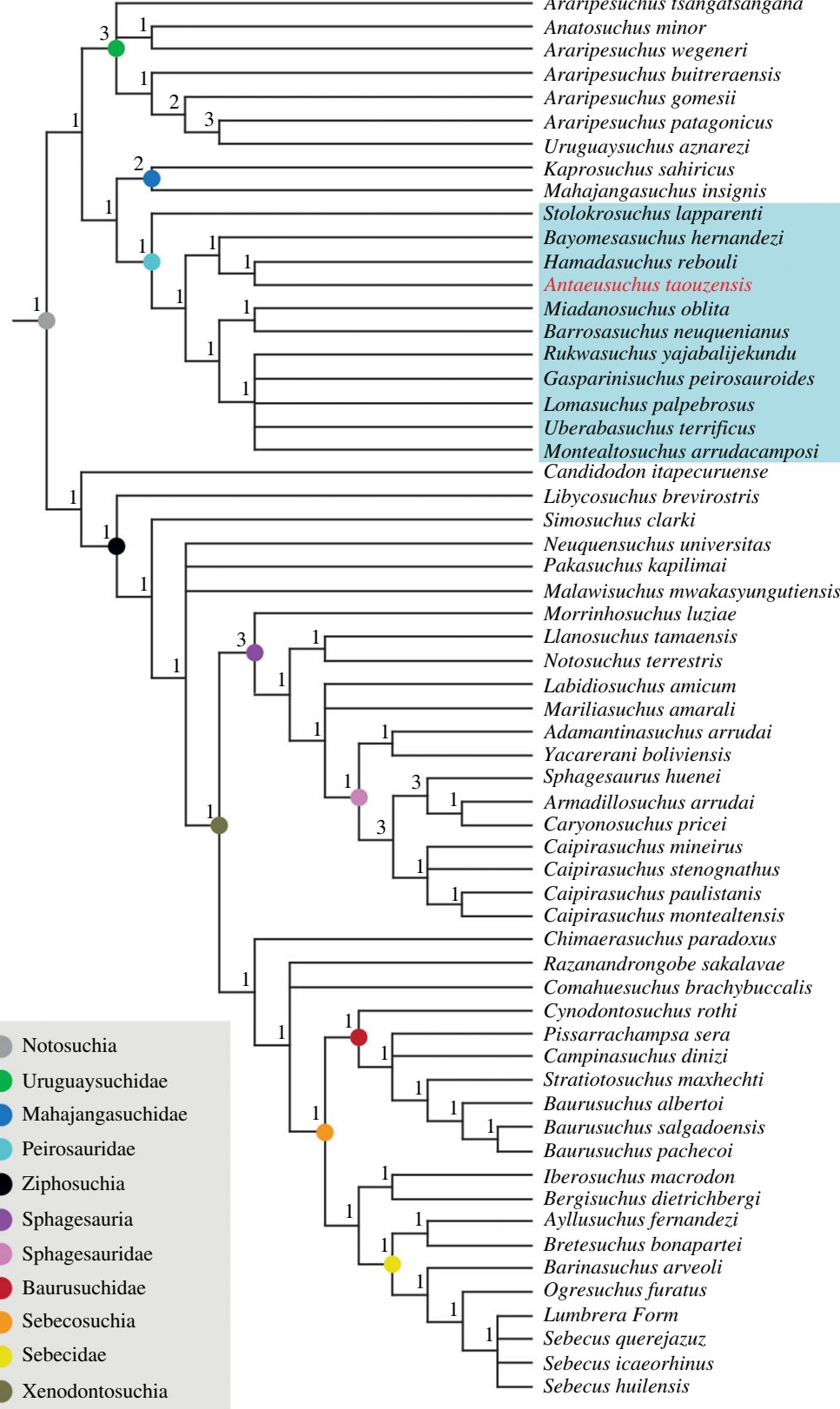

**Figure 7.** Strict consensus tree showing the relationships of notosuchians using an equal weighting of characters. Numbers at the nodes indicate Bremer support values.

*Antaeusuchus* is recovered within Peirosauridae, as the sister taxon to *Hamadasuchus* (figure 7). The two Kem Kem OTUs form a clade with *Bayomesasuchus* that is the sister group to nearly all other peirosaurids. Within this latter group, *Barrosasuchus* and *Miadanosuchus* form a clade that is the sister

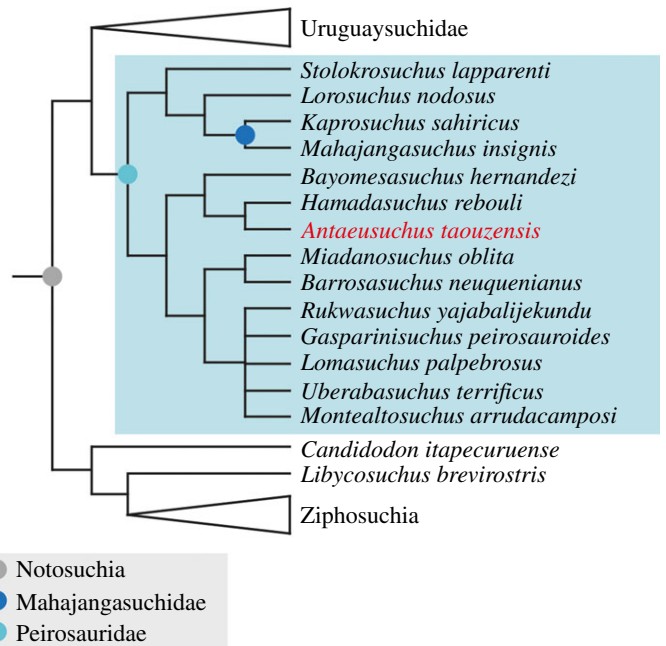

**Figure 8.** Strict consensus tree showing the relationships of notosuchians using extended implied weighting at *k*-values of 8 and 12. Some clades (Uruguaysuchidae and Ziphosuchia) have been condensed.

taxon to a polytomy comprising *Rukwasuchus*, *Gasparinisuchus peirosauroides*, *Lomasuchus palpebrosus*, *Uberabasuchus* and *Montealtosuchus arrudacamposi*. This polytomy can be resolved through the *a posteriori* pruning of *Gasparinisuchus* in the agreement subtree, which results in *Lomasuchus* and *Montealtosuchus* recovered as sister taxa, forming a clade with *Uberabasuchus*, with *Rukwasuchus* placed at the 'base' of this grouping. *Stolokrosuchus lapparenti* is recovered as the earliest diverging member of Peirosauridae.

Under EIW, with both *k*-values, the overall topology is largely similar to that recovered using EQW. With a *k*-value of 8, the analysis produced 45 trees with a tree length of 89.3, and with a *k*-value of 12, 15 MPTs were found of length 68.3. The main difference with results obtained from EQW is that the taxonomic content of Peirosauridae is now expanded, as a result of *Stolokrosuchus* being recovered in a clade with *Lorosuchus nodosus* and Mahajangasuchidae (figure 8). The phylogenetic definition proposed by Geroto & Bertini [72] means that Mahajangasuchidae is a clade within Peirosauridae according to our EIW topology.

# 5. Comparisons

Given the results of our phylogenetic analyses, which provide robust and consistent support for a peirosaurid placement for *Antaeusuchus*, we largely restrict our anatomical comparisons to members of this clade.

## 5.1. Comparisons with other peirosaurids

Characteristic of all members of Peirosauridae [57], *Antaeusuchus* possesses an undulatory dorsal margin of the dentary comprising two distinct waves, the apices of which coincide with the position of the two largest dentary teeth [6,73]. In all peirosaurids in which the relevant region is preserved, with the exception of *Stolokrosuchus* [54], the 4th dentary tooth is the largest of those in the first wave, and it coincides with the apex of the anterior dentary region. The position of the largest tooth (the apex of the posterior wave) also varies among peirosaurids [6]. The apex of this posterior wave corresponds to the 13th tooth position in *Antaeusuchus*, *Barrosasuchus*, *Gasparinisuchus* and *Kinesuchus overoi* [57,60,74], whereas it occurs at the level of the 12th tooth in *Hamadasuchus* and *Montealtosuchus* [46,56]. In *Stolokrosuchus*, the jaw gradually increases in dorsoventral height posteriorly, corresponding with a progressively larger tooth size towards the posterior region of the dentary [65]. In lateral view, the dorsoventrally expanded posterior region of the mandible of *Antaeusuchus* more closely resembles that

of *Hamadasuchus*, *Montealtosuchus* and *Uberabasuchus* [46,56,75] than it does in taxa such as *Barrosasuchus* and *Kinesuchus* (and potentially *Pepesuchus deiseae*) [60,74,76], in which the region is more dorsoventrally compressed.

All peirosaurid taxa, except for the longirostrine-snouted *Stolokrosuchus* [54], are characterized by a mediolaterally broad mandibular symphysis (figure 9). This is most prominent in *Colhuehuapisuchus lunai*, *Barrosasuchus* and *Gasparinisuchus* [57,60,77], in which the anteroposterior length to mediolateral width ratio of the symphyseal dorsal surface is less than 1.0 (values range from 0.8 to 0.9). Although still broad in comparison with many other notosuchian taxa, such as sphagesaurians [20], which often have length to width ratios exceeding 2.0 [6], *Antaeusuchus*, as well as *Hamadasuchus*, possess some of the anteroposteriorly longest mandibular symphyses among Peirosauridae, with a ratio of 1.4 and 1.5 for *Antaeusuchus* and *Hamadasuchus*, respectively [46]. The ratios are 1.2 in *Montealtosuchus* and 1.1 in both *Uberabasuchus* and *Bayomesasuchus* [56,58,75]. *Kinesuchus* preserves the anteroposteriorly longest mandibular symphyses of any peirosaurid, with a ratio of 2.0 [74].

As is the case in all notosuchians [6], the splenials of *Antaeusuchus* participate in the mandibular symphysis, although the extent to which this is the case varies between peirosaurids (figure 9). The splenial of *Antaeusuchus* occupies 39% of the anteroposterior length of the symphysis in dorsal view. A relatively long dorsal symphyseal contribution is also present in *Hamadasuchus* (49%), *Bayomesasuchus* (40%), *Kinesuchus* (44%), *Patagosuchus anielensis* (approx. 44%) and potentially *Uberabasuchus*, although the latter cannot be observed in dorsal view [58,74,75,78]. A much shorter splenial contribution to the symphysis characterizes *Gasparinisuchus* (16%), *Barrosasuchus* (21%) and *Colhuehuapisuchus* (approx. 26%) [57,60,77].

Where the splenial meets the dentary on the symphyseal dorsal surface of *Antaeusuchus*, the suture forms an approximate 'V'-shape. A similar morphology is present in *Hamadasuchus* [46], *Bayomesasuchus* [58], *Kinesuchus* [74] and *Stolokrosuchus* [54], as well as *Patagosuchus* [78] and *Montealtosuchus* [56], although the 'V' is slightly broader in the latter two species, forming a less acute angle. This morphology contrasts with that of *Barrosasuchus*, *Gasparinisuchus*, *Colhuehuapisuchus* and *Miadanasuchus* [57,60,64,77], in which the dentary-splenial suture forms a broad 'U' shape. As with all other peirosaurids [6], the dorsal surface of the mandibular symphysis on which this suture occurs is very slightly transversely concave in *Antaeusuchus*.

In *Antaeusuchus*, as well as *Hamadasuchus* [46], the dentary-splenial suture exposed on the dorsal surface of the mandibular symphysis diverges gradually from the midline until the level of the 12th dentary tooth, at which point it becomes parallel to the toothrow. A similar morphology characterizes *Kinesuchus*, except that the change in orientation of the suture is less acute in that species [74]. In *Montealtosuchus* and *Patagosuchus*, the suture becomes parallel with the toothrow at the level of the 10th dentary tooth [56,78]. By contrast, the dentary-splenial suture in *Stolokrosuchus* parallels the toothrow only at the level of the 25th tooth [54]. Although the morphology of the suture is 'U'-shaped, as opposed to the 'V'-shape that characterizes *Antaeusuchus*, it becomes approximately parallel with the toothrow at the level of the 11th and 12th tooth in *Gasparinisuchus* and *Barrosasuchus*, respectively [57,60].

Posterior to the symphysis, the mandibular rami of *Antaeusuchus* diverge at an angle of approximately 44° to each other. A value of between approximately 40 and 45° is fairly consistent among peirosaurids; this contrasts with some other notosuchians, including sphagesaurians [6,61], whereby the skull is mediolaterally broader, and the rami diverge from one another at a less acute angle. Immediately posterior to the symphysis, the splenial of *Antaeusuchus* is largely exposed in ventral view, and forms approximately 40% of the mediolateral width of the mandibular rami. A comparably broad splenial also characterizes *Uberabasuchus*, *Montealtosuchus* and *Kinesuchus* [56,74,75], whereas the splenial comprises only 25–30% of the rami transverse cross section in *Gasparinisuchus*, *Stolokrosuchus*, *Colhuehuapisuchus* and *Barrosasuchus* [54,57,60,77].

In numerous peirosaurids, including *Antaeusuchus*, *Hamadasuchus*, *Uberabasuchus*, *Montealtosuchus*, *Pepesuchus*, *Stolokrosuchus* and *Lomasuchus*, an anteroposteriorly elongate groove runs parallel to the dentary toothrow, just ventral to the dorsal margin of the lateral surface of the mandible [46,54,56,75,76,79]. The lateral surface of the dentary is also typically sculpted with pits and/or grooves in peirosaurids; however, there is interspecific variation in the degree to which this sculpting continues over the entire surface. In *Antaeusuchus*, the lateral surface of the dentary is similar in its texture and sculpting both above and below the groove. In this regard, the morphology is similar to that of *Uberabasuchus*, *Barrosasuchus* and *Kinesuchus* [60,74,75]. In *Hamadasuchus*, *Montealtosuchus*, *Pepesuchus* and *Patagosuchus*, the region above the groove is smooth, differing markedly from the remainder of the highly sculpted lateral dentary surface [46,56,76,78]. Although *Stolokrosuchus* shows

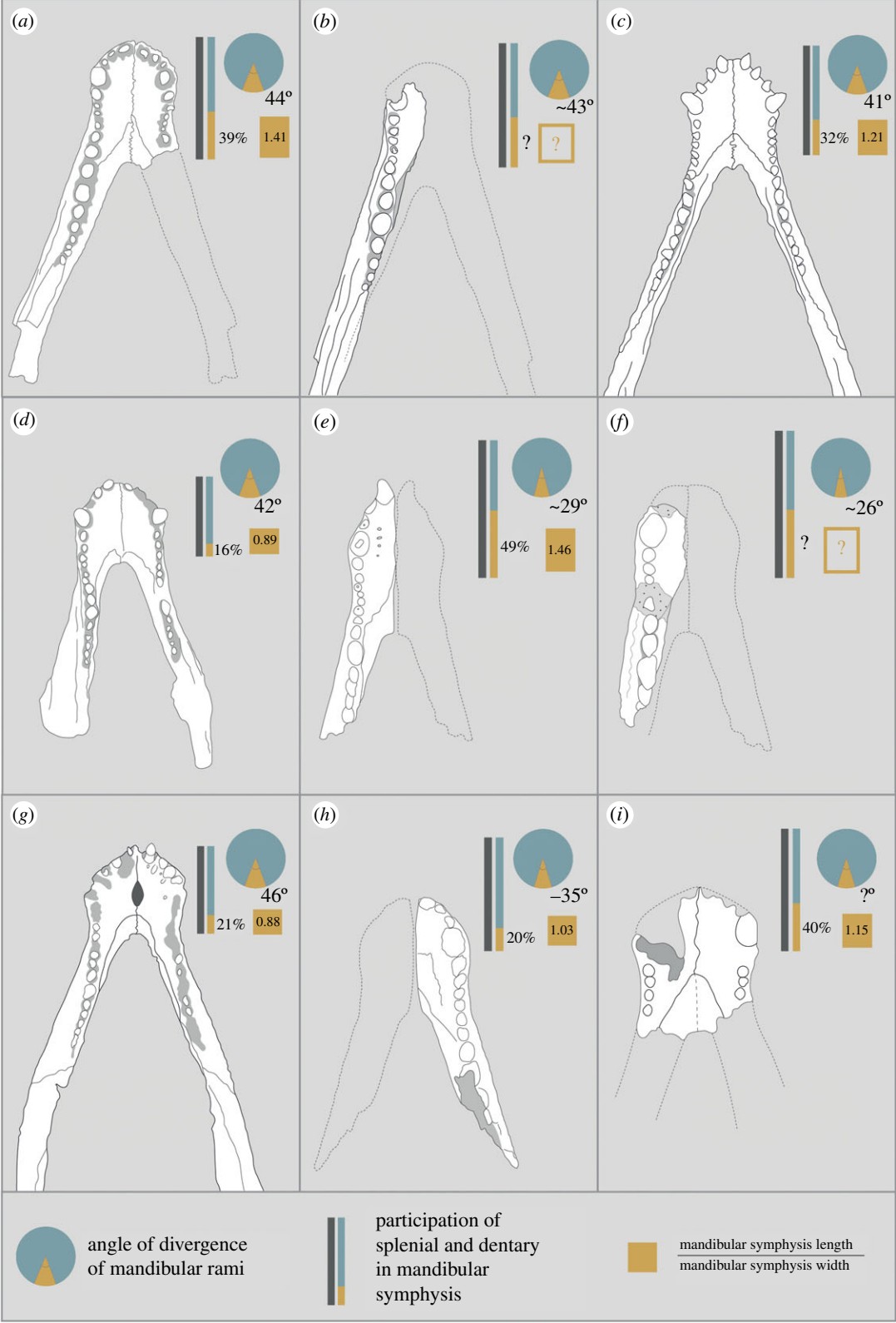

**Figure 9.** Comparison of the dorsal mandibular surfaces of several notosuchians: (*a*) *Antaeusuchus taouzensis* n. gen. n. sp. (NHMUK PV 36829*); (*b*) *Antaeusuchus taouzensis* (NHMUK PV R36874); (*c*) *Montealtosuchus arrudacamposi* (MPMA 16-0007-04*); (*d*) *Gasparinisuchus peirosauroides* (MOZ 1750 PV*); (*e*) *Hamadasuchus rebouli* (ROM 49282); (*f*) *Hamadasuchus rebouli* (MDE C001*); (*g*) *Barrosasuchus neuquenianus* (MCF-PVPH-413*); (*h*) *Araripesuchus rattoides* (CMN 41893*); (*i*) *Bayomesasuchus hernandezi* (MCF PVPH-822). Asterisk indicates a holotype specimen.

no difference in bone surface texture above and below the anteroposterior groove [80], it differs from *Antaeusuchus* in that the majority of the lateral dentary surface is smooth and unornamented.

The degree of sculpting on the lateral surface of the dentary varies across notosuchian taxa, including within Peirosauridae. Whereas the surface is covered in deep pits in *Hamadasuchus*, *Uberabasuchus*, *Montealtosuchus*, *Patagosuchus*, *Bayomesasuchus* and *Miadanasuchus* [46,56,58,64,75,78], the surface of *Antaeusuchus* is considerably smoother and is textured with narrow, shallow grooves.

Unlike *Montealtosuchus*, *Gasparinisuchus*, *Bayomesasuchus*, *Stolokrosuchus*, *Patagosuchus*, *Pepesuchus* and *Colhuehuapisuchus* [54,56,57,76,77,81], the lateral surface of the dentary adjacent to the 5th–8th teeth forms a distinct anteroposteriorly elongate concavity in *Antaeusuchus*. This is otherwise known only in *Hamadasuchus* [46], although a shallower concavity also characterizes *Barrosasuchus* [60]. This depression would likely have functioned to receive an enlarged premaxillary tooth during occlusion.

Although incompletely preserved, the mandibular fenestra in *Antaeusuchus* is almost certainly large and anteroposteriorly elongate, as is the case in *Hamadasuchus*, *Montealtosuchus* and *Uberabasuchus* [37,56,75], but differing from *Barrosasuchus* in which the fenestra is greatly reduced [60]. However, *Montealtosuchus* and *Uberabasuchus* differ from the dentaries of *Antaeusuchus* and *Hamadasuchus* [37] in that the latter two have a small posterior process that extends ventral to the mandibular fenestra. This process is absent in *Montealtosuchus* and *Uberabasuchus*, in which the dentary-angular contact is entirely anterior to the mandibular fenestra instead [56,75]. The dentary-surangular contact is similar in *Antaeusuchus*, *Hamadasuchus*, *Montealtosuchus* and *Uberabasuchus*, with the surangular contacted by two posterior processes: the dorsal process intrudes entirely into the surangular, whereas the second process forms the ventral margin of the surangular and the anterodorsal margin of the mandibular fenestra [46,56,75]. This feature cannot be assessed in other peirosaurid taxa, in which the relevant region of the mandible is not preserved.

When complete, each dentary of *Antaeusuchus* has 18 tooth positions. This count is common among peirosaurids, e.g. *Montealtosuchus*, *Gasparinisuchus*, *Kinesuchus*, *Pepesuchus* and possibly *Barrosasuchus* [56,57,60,74,76], but differs from *Stolokrosuchus*, in which there are at least 30 dentary alveoli [54]. As in all peirosaurids [6], the first two dentary teeth of *Antaeusuchus* are strongly procumbent.

The dentary teeth posterior to the 5th alveolus are closely spaced and are mostly situated in a continuous groove in *Antaeusuchus*, *Hamadasuchus* [46], *Gasparinisuchus* [57] and *Barrosasuchus* [60]. This differs from the condition in *Kinesuchus* [74] and *Patagosuchus* [78], in which the teeth are separated by distinct septa that extend fully to the dorsal margin of the dentary.

## 5.2. Detailed comparisons with *Hamadasuchus rebouli*

Although several crocodyliform taxa, including notosuchians, have been identified from the Kem Kem Group [37], only one peirosaurid species is currently recognized from these beds: *H. rebouli* [43,46]. Since the original description of the holotype dentary by Buffetaut [43], several specimens have been referred to *Hamadasuchus* [37,44–46]. In our CTM (and previous iterations), the OTU of *Hamadasuchus* comprises the holotype mandibular fragment, MDEC001, plus the cranial material, ROM 52620, referred by Larsson & Sues [46]. Almost all characters that could be assessed for *Antaeusuchus* received the same score as *Hamadasuchus*, resulting in their consistent recovery as sister taxa in our phylogenetic analyses. The only differences in scores are present in characters 77 (scored as 2 and 1 and 2 for *Antaeusuchus* and *Hamadasuchus*, respectively), 155 (scored as 1 and 0&1 for *Antaeusuchus* and *Hamadasuchus*, respectively) and 393 (scored as 0 and 0&1 for *Antaeusuchus* and *Hamadasuchus*, respectively). The score of 1&2 for character 77 reflects the fragmentary nature of the *Hamadasuchus* type specimen and uncertainty of the precise length of the splenial contribution to the mandibular symphysis, rather than representing a polymorphism, whereas the score of 0&1 for characters 155 and 393 represents the definite presence of both states in this OTU. Given the similarity of the scores of both Kem Kem specimens, and that those provided for the mandible of *Hamadasuchus* are based only on the holotype specimen and not any referred material, we provide more detailed comparisons in the following section. *Antaeusuchus* is compared to several anatomically overlapping specimens currently assigned to *Hamadasuchus*, namely the holotype dentary (MDEC001), several partial mandibles (ROM 49282, 52045 and 52047) described by Larsson & Sues [46], a complete skull and lower jaws (BSPG 2005 I 83) figured by Rauhut & López-Arbarello [45], and two mandibular symphyses (MNHN-MRS 3110 and NMC 41784) illustrated in Ibrahim *et al.* [37]. Despite being largely similar in overall morphology, *Antaeusuchus* differs in several respects from all specimens assigned to *Hamadasuchus* (figure 10).

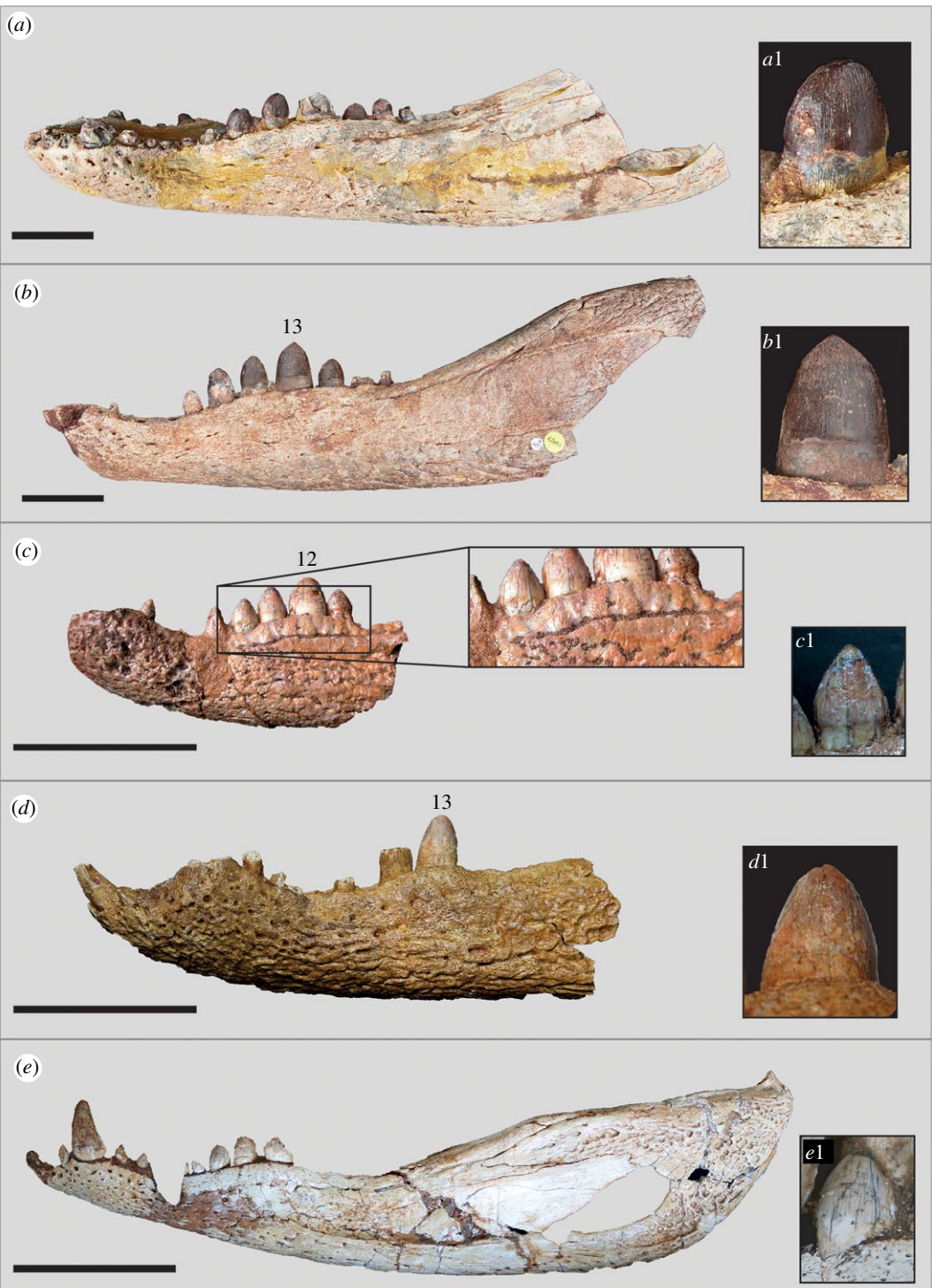

**Figure 10.** Comparison of peirosaurid mandibles from the Kem Kem Group in lateral view: (*a*) NHMUK PV 36829 (*Antaeusuchus taouzensis* n. gen. n. sp. holotype); (*b*) NHMUK PV R36874 (*Antaeusuchus taouzensis* paratype); (*c*) MDEC001 (*Hamadasuchus rebouli* holotype); (*d*) ROM 49282; (*e*) BSPG 2005 I 83. (*a*1–*e*1) show close-up images of the teeth of each respective taxon. Image (*b*) is reversed. Scale bars represent 50 mm.

Although only one dentary is preserved in the holotype (MDEC001), the angle of divergence of the mandibular rami can be inferred by measuring the angle of deviation of one dentary from the exposed symphyseal suture. Estimated mandibular rami divergence angles of approximately 20° for ROM 52047, approximately 25° for MDEC001 and approximately 30° for ROM 49282, 52045, are much narrower than that of *Antaeusuchus* (43–44°). This suggests that *Hamadasuchus* had a slenderer rostrum than that of *Antaeusuchus*.

The surface texture of the *Antaeusuchus* dentary differs from all three specimens referred to *Hamadasuchus* in that it is not covered in deep pits and grooves. Instead, it is ornamented with relatively narrow and shallow grooves. *Antaeusuchus* also differs from these specimens in the dorsal region of the dentary lateral surface. In MDEC001, the area just ventral to the toothrow is smooth and lacks pits, differing from the remainder of the lateral surface [43]. Instead, this dorsal region has a corrugated morphology, with broad, shallow dorsoventral grooves that do not correspond to the position of the dentary teeth (figure 10). A similar morphology characterizes ROM 49282, BSPG 2005 I 83 and NMC 41784, although it is not as prominent in those specimens. In *Antaeusuchus*, the degree of ornamentation is much the same across the lateral surface of the dentary and a fluted dorsal region is absent.

The apex of the second mandibular wave is marked by the position of the 13th tooth in *Antaeusuchus*, as well as ROM 49282, BSPG 2005 I 83 and probably MNHN-MRS 3110. By contrast, the tip of the second dentary wave is most likely marked by the 12th tooth in the holotype MDEC001, which is significantly larger than the 13th tooth [43].

Posterior to the 7th dentary tooth, the teeth of MDEC001 are strongly labiolingually compressed, and possess serrated carinae on their anterior and posterior cutting edges. A similar morphology is also present in *Antaeusuchus*, as well as MNHN-MRS 3110, but not in ROM 49282, in which the teeth are compressed only from the 10th tooth. The 5th–9th teeth are damaged in BSPG 2005 I 83 and so it is unclear at which tooth position the compression commences. The posterior teeth in MDEC001 have a lanceolate shape [43] that is not present in *Antaeusuchus*, but which is most reminiscent of the morphology in MNHN-MRS 3110. In NHMUK PV R36829, teeth 11, 12, 14 and 15 are the only ones which are fully preserved; the anterior two of these have rounded crowns that do not form an angular tip. Although the 14th and 15th teeth of NHMUK PV R36829 are slightly pointed at their apices, they are not comparable to the almost triangular shape of those in the holotype MDEC001 [15]. In NHMUK PV R36874, the 10th–14th teeth are slightly more angular in appearance, but their anterior and posterior margins are parallel for most of their length, converging to a point only at the crown apex (figure 10).

The only fully preserved tooth in the ROM 49282 dentary is the 13th one [46]. Comparing the broad morphology, it is most similar to the teeth at a similar dentary position in *Antaeusuchus*, in that its apical margin is rounded. However, the tooth enamel of ROM 49282 is essentially smooth. By contrast, the enamel in *Antaeusuchus* is wrinkled into anastomosing apicobasal ridges (approx. 2–3 ridges per mm) (figure 10). Both morphologies differ from that of the teeth of MDEC001, in which the enamel is textured, but has an irregular, globular pattern towards its base and anastomosing ridges towards the apex of the crown [43]. ROM 49282 also displays a unique condition in which the tooth enamel is fluted with broad apicobasal ridges around its circumference.

The teeth of all specimens assigned to *Hamadasuchus*, as well as those of *Antaeusuchus*, have very finely serrated carinae on their anterior and posterior cutting edges, with individual serrations spaced at approximately 3–5 per mm. Although most prominent in MDEC001, vertical fluting on the anterior and posterior regions of the crowns is visible in the posterior dentary teeth of all the specimens discussed in this section.

Finally, *Antaeusuchus* is significantly larger than all *Hamadasuchus* specimens, such that it is almost double the size of MDEC001, ROM 49282 and BSPG 2005 I 83. Although the size of the specimen alone should not be a reason to erect a new species (e.g. [82]), we believe it to be a valid morphological difference as part of a large, unique combination of features. Furthermore, *Hamadasuchus* is already known from an ontogenetic series, including specimens considered 'adult' [46]; as such, it is difficult to reconcile the numerous anatomical differences merely as a result of *Antaeusuchus* being an even older individual of *Hamadasuchus*).

# 6. Discussion

## 6.1. Taxonomic affinities of NHMUK PV R36829 and R36874

In all of our analyses, the NHMUK PV R36829 + R36874 OTU (i.e. *Antaeusuchus*) is recovered as the sister taxon of *Hamadasuchus*. This relationship is supported by a single unambiguous synapomorphy (a distinct concavity adjacent to the 5th–10th dentary teeth for the reception of the enlarged maxillary tooth), and one ambiguous synapomorphy (a short distance between the 4th and 5th mandibular teeth). In total, 34 characters in our CTM can be scored for both the *Hamadasuchus* OTU and

*Antaeusuchus*, with only three of these receiving different scores (characters 77, 155 and 393). The first of these describes the contribution of the splenial to the mandibular symphysis in dorsal view and only partially differs: whereas *Hamadasuchus* is polymorphic (1/2), *Antaeusuchus* is characterized solely by state 2. Nonetheless, both taxa exhibit splenials that are anteroposteriorly more elongate than other peirosaurid taxa, with the exception of *Bayomesasuchus*. The second character in which scores differ describes the sculpting of the dentary region below the toothrow. Again, the difference is only partial, with *Hamadasuchus* polymorphic (0/1) and *Antaeusuchus* possessing the derived condition (i.e. state 1). The elevated sections of this region in the *Hamadasuchus* holotype are characterized by a pitted surface, whereas the depressed areas are smooth. Finally, the third differing character describes the rugose texture of the tooth enamel, for which *Hamadasuchus* is scored as 0 and 1, whereas *Antaeusuchus* is characterized by the plesiomorphic condition (i.e. state 0). The tooth enamel in the *Hamadasuchus* holotype is more globular towards the middle and basal regions of the tooth crown, becoming more linear and ridgelike towards its apex. In *Antaeusuchus*, elongate, anastomosing ridges run from the apex to the base of the enamel.

Although there are only three differences captured in our CTM, our detailed comparisons demonstrate numerous additional features that indicate that NHMUK PV R36829 and R3687 are not referrable to *H. rebouli*. NHMUK PV R36829 and R36874 differ from *H. rebouli* in their large size as well as the possession of a unique combination of features: (i) a high angle of divergence between mandibular rami; (ii) a rugose dentary tooth enamel shaped into anastomosing apicobasal ridges; (iii) the largest dentary tooth in the second wave is located in alveolus 13; (iv) sub-triangular tooth crowns (in lateral view) with a gently curved apex; (v) minor labiolingual compression of the posterior dentary teeth; (vi) a lack of fluting on the dorsal region of the lateral dentary surface; (vii) a relatively unornamented surface texture of the dentary adorned with narrow, shallow ridges rather than deep pits or grooves; and (viii) dentary teeth more widely spaced at their base.

As such, it seems clear that NHMUK PV R36829 + R3687 represents a second peirosaurid in the Kem Kem Group, and thus supports our erection of *Antaeusuchus taouzensis* n. gen. n. sp. In addition, a material currently referred to *Hamadasuchus* also differs from the type specimen (MDEC001), as well as *Antaeusuchus*. In particular, the partial mandible, ROM 49282, described by Larsson & Sues [46], differs from both taxa in several features, including: (i) distinctive apicobasal fluting on the 13th tooth; (ii) a highly elongate contribution of the splenial to the mandibular fenestra; (iii) a mandibular rami divergence of approximately 30°; and (iv) possession of relatively smooth tooth enamel. The unique combination of characters in each of MDEC001, ROM 49282 and NHMUK PV R36829 + R36874, therefore, suggests the presence of at least three separate, albeit closely related, peirosaurid species from the Kem Kem Group. Although we erect a new name for NHMUK PV R36829 + R3687, we refrain from naming a new taxon for ROM 49282 pending the description and assessment of additional materials currently assigned to *H. rebouli* (namely BSPG 2005 I 83 and additional ROM specimens).

## 6.2. Implications for peirosaurid relationships

Peirosauridae was erected by Gasparini [83] to accommodate *Peirosaurus torminni* [84] from the late Maastrichtian Marília Formation of Brazil. This family was subsequently expanded by Gasparini *et al.* [79] to include *Lomasuchus* from the late Turonian–early Coniacian of Argentina. Geroto & Bertini [85, p. 328] provided a phylogenetic definition for Peirosauridae as 'the least inclusive clade containing *Peirosaurus tormini* [sic] [84], *Itasuchus jesuinoi* [84] and *Stolokrosuchus lapparenti* [54], but not including *Notosuchus terrestris* Woodward, 1896, *Baurusuchus pachecoi* Price, 1945, *Sphagesaurus huenei* Price, 1950, *Araripesuchus gomesii* Price, 1959, *Sebecus icaeorhinus* Simpson, 1937, *Mariliasuchus amarali* Carvalho and Bertini, 1999 and *Crocodylus niloticus* Laurent, 1768'. Although a phylogenetic definition based on two well-nested and stable species-level specifiers would be preferable (e.g. [86–88]), we follow the definition of Geroto & Bertini [85] here, pending a detailed re-evaluation of the interrelationships of this part of the notosuchian tree.

Following Geroto & Bertini's [85] definition, Peirosauridae comprises a taxonomically rich array of crocodyliforms from across the Cretaceous of South America, Africa and Madagascar (e.g. [46,56–58,60,63,74–76,78,84,89]). However, there is little consensus regarding the position of Peirosauridae. A number of analyses have recovered Peirosauridae within Notosuchia, as the sister taxon to Mahajangasuchidae (i.e. *Kaprosuchus* + *Mahajangasuchus*), with these lineages forming a clade with Uruguaysuchidae that is the sister taxon to all other notosuchians (e.g. [6,60,63]). Others have recovered Peirosauridae as part of Sebecia, forming a clade with Sebecidae (e.g. [28,46]), and

sometimes also including Mahajangasuchidae (e.g. [20,72]). Whereas some of these analyses place *Sebecia* as the sister taxon to all other notosuchians (e.g. [20,72]), others recover *Sebecia* within Neosuchia (e.g. [28,46]). Peirosauridae has also been recovered as an early diverging neosuchian clade in some studies (e.g. [89–92]).

In several recent phylogenetic analyses (e.g. [6,60,85]), *Hamadasuchus* has been recovered as the sister taxon to a group of exclusively South American Cretaceous peirosaurids (comprising various combinations of *Montealtosuchus*, *Uberabasuchus*, *Lomasuchus*, *Gasparinisuchus* and *Barcinosuchus*). Similarly, Barrios *et al.* [58] recovered *Hamadasuchus* in a polytomy with most of these taxa, along with *Bayomesasuchus* from the Turonian (Late Cretaceous) of Argentina. Sertich & O'Connor [63] recovered *Hamadasuchus* in an unresolved trichotomy with *Rukwasuchus* and *Stolokrosuchus*, forming a clade of African peirosaurids.

Here, under both EQW and EIW schemes, the position of Peirosauridae is consistent with the results of Pol *et al.* [6] and subsequent studies based on this dataset (e.g. [49–51,60,61,93]). Under its broadened taxonomic content, following the phylogenetic definition of Geroto & Bertini [85], Peirosauridae includes Mahajangasuchidae in our EIW analyses (figure 8). This occurs because *Stolokrosuchus* is recovered as more closely related to Mahajangasuchidae than to other peirosaurids in the EIW topology. Our equal weights analysis recovers *Stolokrosuchus* as the most 'basal' member of Peirosauridae instead, with Mahajangasuchidae outside of this clade (figure 7). In both cases, our peirosaurid + mahajangasuchid grouping is the sister taxon of Uruguaysuchidae, with this clade the sister taxon to all other notosuchians.

In our strict consensus trees, the clade comprising *Antaeusuchus* and *Hamadasuchus* is most closely related to *Bayomesasuchus*. This grouping is the sister taxon to other peirosaurids (excluding *Stolokrosuchus* and Mahajangasuchidae) (figure 7). The remaining South American taxa are grouped in a polytomy with the African taxon *Rukwasuchus*, with this recovered as the sister taxon of a clade comprising the Malagasy taxon *Miadanasuchus* and the Argentinean species *Barrosasuchus*. The aforementioned polytomy can be resolved via *a posteriori* pruning of *Gasparinisuchus*, resulting in *Rukwasuchus* as the sister taxon of (*Uberabasuchus* + (*Lomasuchus* + *Montealtosuchus*)).

The fact that our analyses produce topologies more consistent with those derived from the data matrix of Pol *et al.* [94] than alternative matrices is not surprising given that this is the underlying dataset for our study. As such, the interrelationships of Peirosauridae within Metasuchia will require further testing, ideally merging characters and taxa from across studies with competing hypotheses. However, the recovery of Peirosauridae as an early diverging metasuchian clade outside of the ziphosuchian notosuchian radiation is consistent across analyses, regardless of the underlying dataset.

One of the notable results of our analyses is the placement of *Miadanasuchus* within Peirosauridae, which was independently recovered in this clade by Geroto & Bertini [85]. This species from the Maastrichtian of Madagascar was originally described as *Trematochampsa oblita* [94], before being assigned to a new genus by Rasmusson Simons & Buckley [64]. The type species of *Trematochampsa*, *T. taqueti*, is based on fragmentary remains from the Coniacian–Santonian In Beceten Formation of Niger [95–97], for which the family Trematochampsidae was also erected [95]. Several additional crocodyliform taxa have been assigned to Trematochampsidae (e.g. *Amargasuchus minor* [98], *Barreirosuchus francsicoi* [99], *Hamadasuchus*, *Itasuchus*, *Mahajangasuchus*), spanning the Cretaceous of Africa, Europe, Madagascar and South America, with most of these known from fragmentary remains (see review in [100]). Buffetaut [101,102] also included *Peirosaurus torminni* as a member of Trematochampsidae, which would, therefore, have priority over Peirosauridae. However, multiple authors have questioned or rejected the monophyly of Trematochampsidae, which appears to have become a wastebasket taxon (e.g. [46,73,79,81,100,103,104]). Furthermore, Meunier & Larsson [100] demonstrated that *T. taqueti* is a nomen dubium, based on non-diagnostic, chimeric remains, with some of these displaying peirosaurid affinities. Our analyses provide further evidence that most, if not all, Cretaceous taxa previously assigned to Trematochampsidae belong to Peirosauridae, and confirm the presence of this latter clade in the Maastrichtian of Madagascar. Given the lack of diagnostic features in the type remains of '*T. taqueti*' and the absence of a formal definition for 'Trematochampsidae', coupled with its approximate synonymy with the formally defined and widely used Peirosauridae, we support the proposal of Meunier & Larsson [100] to abandon the name *Trematochampsa* and its coordinated rank taxa.

## 6.3. Gondwanan notosuchian diversity outside of South America

During the Mesozoic, notosuchians (*sensu* [6]) were the most diverse clade of Gondwanan crocodyliforms [105], although this high species richness varied through both time and space [7,106].

At least 70% of known notosuchian diversity is found on Gondwanan continents [7], with a small number of species recognized from Europe [62,107–113] and Asia [16,114]. Though most numerous in South America, Gondwanan notosuchian occurrences are also known from mainland Africa, Madagascar, India and Pakistan (table 3), as well as possibly the Arabian Peninsula. Currently, no notosuchians are known from Australasia or Antarctica, although it remains unclear whether this represents a genuine absence, perhaps pertaining to a high-latitude environmental dispersal barrier, or it reflects a sampling bias (e.g. see [118]). Here, we provide a critical reappraisal of the Gondwanan record of notosuchians outside of South America.

### 6.3.1. Jurassic

The stratigraphically oldest known notosuchian is *Razanandrongobe sakalavae* [59] from the Bathonian (Middle Jurassic) Isalo IIIb Formation in northwestern Madagascar. Originally named as an archosaur of uncertain affinities on the basis of teeth and a fragmentary maxilla [59], several more skull fragments, including a right premaxilla and an incomplete left dentary, have since been assigned to the taxon, enabling its identification as a large-bodied notosuchian [25]. Considering that the next stratigraphically oldest notosuchians are from the Aptian (late Early Cretaceous), resulting in a approximately 40 Myr ghost lineage, *Razanandrongobe* is a stratigraphic outlier and its affinities might seem doubtful. However, based on the sister taxon relationship of Notosuchia and Neosuchia, with the latter clade known from the Early Jurassic [119], *Razanandrongobe* instead partly fills the inferred ghost lineage of notosuchians, which otherwise would extend back approximately 65–75 Myr [25,26]. In the small number of phylogenetic analyses to have incorporated it [25,62], including ours, *Razanandrongobe* is recovered in a position close to the 'base' of Sebecosuchia. This nested position within Notosuchia for such a stratigraphically early species necessitates the extension of multiple unsampled lineages back into the Jurassic (figure 11). As such, the phylogenetic affinities of *Razanandrongobe* require further evaluation to test whether this poor stratigraphic fit is genuine.

### 6.3.2. Early Cretaceous

In southeastern Africa, the Aptian Dinosaur Beds of northern Malawi (figure 12) have yielded numerous remains of *Malawisuchus mwakasyungutiensis*, preserving most of the skeleton [53]. Recognized in part for its unusual mammal-like multicuspid teeth, some analyses have placed *Malawisuchus* in a nested position within Sphagesauria (e.g. [18,28,53]). However, most recent analyses typically recover *Malawisuchus* as an early diverging ziphosuchian, with spaghesaurians currently restricted to South America (e.g. [6,20]; this study). Unlike the topology of Martin & Lapparent de Broin [38], *Malawisuchus* is not recovered within Candidodontidae in our analyses (figure 7).

The Aptian–Albian Elrhaz Formation exposed at Gadoufaoua, central Niger (figure 12), has yielded the remains of three morphologically diverse notosuchian species (*Anatosuchus minor*, *Araripesuchus wegeneri* and *Stolokrosuchus lapparenti*). The bizarre, 'duck-billed' *Anatosuchus* is known from several individuals, including a skull and associated partial postcranial skeleton, as well as a skull of a juvenile animal [9,28]. *Anatosuchus* has often been recovered as a member of Uruguaysuchidae (e.g. [6,28]; some analyses have placed it outside of this clade, although these tend to recover it as a 'basal' member of Notosuchia. The small and gracile species *Ar. wegeneri* was erected from the anterior region of an articulated upper and lower snout [94]. Multiple remains have since been assigned to the taxon, including a block preserving at least five separate individuals, three of which are essentially complete, partially articulated skeletons [28]. In our analyses, *Ar. wegeneri* and *Anatosuchus* are recovered as sister taxa within Uruguaysuchidae, further questioning the monophyly of *Araripesuchus* (see [28, p. 31]). The longirostrine-snouted *Stolokrosuchus* is known from an almost complete skull [54]. Originally referred to Peirosauridae ([54]; see also [28,46,85]), subsequent analyses have shown the position of *Stolokrosuchus* to be highly labile, such that it has also been placed as an early diverging member of both Notosuchia (e.g. [25]) and Neosuchia (e.g. [105,120]). Following the definition of Peirosauridae provided by Geroto & Bertini [85], our analyses recover *Stolokrosuchus* as the earliest diverging member of this clade, which is consistent with previous analyses that have continued to place it close to the 'base' of Metasuchia.

Several isolated teeth from the Aptian–Albian Koum Formation of northeastern Cameroon (figure 12) were reported by Flynn *et al.* [121] and Congleton [122], who recognized their possible affinities with *Araripesuchus*, especially *Ar. wegeneri*. Kellner [123, p. 618] questioned this referral, suggesting that these strongly serrated, laterally compressed, leaf-shaped teeth differed from those in the posterior

**Table 3.** Spatio-temporal distribution and phylogenetic affinities of non-South American, Gondwanan named notosuchian species.

| taxon | stratigraphic and geographical provenance | age | phylogenetic position | reference |
|---|---|---|---|---|
| *Razanandrongobe sakalavae* | Isalo IIIB Fm., Madagascar | Bathonian, Middle Jurassic | Sebecosuchia? | Maganuco et al. [59] |
| *Malawisuchus mwakasyungutiensis* | Dinosaur Beds Fm., Malawi | Aptian, Early Cretaceous | Basal Ziphosuchia | Gomani [53] |
| *Stolokrosuchus lapparenti* | Elrhaz Fm., Niger | Aptian–Albian, Early Cretaceous | Peirosauridae | Larsson & Gado [54] |
| *Araripesuchus wegeneri* | Elrhaz Fm., Niger | Aptian–Albian, Early Cretaceous | Uruguaysuchidae | Buffetaut [115] |
| *Anatosuchus minor* | Elrhaz Fm., Niger | Aptian–Albian, Early Cretaceous | Uruguaysuchidae | Sereno et al. [9] |
| *Hamadasuchus rebouli* | Kem Kem Group, Morocco | Cenomanian, Late Cretaceous | Peirosauridae | Buffetaut [43] |
| *Lavocatchampsa sigogneaurusselae* | Kem Kem Group, Morocco | Cenomanian, Late Cretaceous | Basal Ziphosuchia | Martin & Lapparent de Broin [38] |
| *Araripesuchus rattoides* | Kem Kem Group, Morocco | Cenomanian, Late Cretaceous | Uruguaysuchidae | Sereno & Larsson [28] |
| *Libycosuchus brevirostris* | Bahariya Fm., Egypt | Cenomanian, Late Cretaceous | Basal Ziphosuchia | Stromer [52] |
| *Kaprosuchus saharicus* | Echkar Fm., Niger | Cenomanian, Late Cretaceous | Mahajangasuchidae | Sereno & Larsson [28] |
| *Rukwasuchus yajabalijekundu* | Galula Fm., Tanzania | Cenomanian–Campanian, Late Cretaceous | Peirosauridae | Sertich & O'Connor [63] |
| *Pakasuchus kapilimai* | Galula Fm., Tanzania | Cenomanian–Campanian, Late Cretaceous | Basal Ziphosuchia | O'Connor et al. [18] |
| *Araripesuchus tsangatsangana* | Maevarano Fm., Madagascar | Maastrichtian, Late Cretaceous | Uruguaysuchidae | Turner [116] |
| *Simosuchus clarki* | Maevarano Fm., Madagascar | Maastrichtian, Late Cretaceous | Basal Ziphosuchia | Buckley et al. [8] |
| *Mahajangasuchus insignis* | Maevarano Fm., Madagascar | Maastrichtian, Late Cretaceous | Mahajangasuchidae | Buckley & Brochu [81] |
| *Miadanosuchus oblita* | Maevarano Fm., Madagascar | Maastrichtian, Late Cretaceous | Peirosauridae | Rasmusson Simons & Buckley [64] |
| *Pabwehshi pakistanensis* | Pab Fm., Pakistan | Maastrichtian, Late Cretaceous | Sebecosuchia? | Wilson et al. [117] |
| *Eremosucus elkoholicus* | El Kohol Fm., Algeria | Ypresian, early Eocene | Sebecosuchia? | Buffetaut [102] |

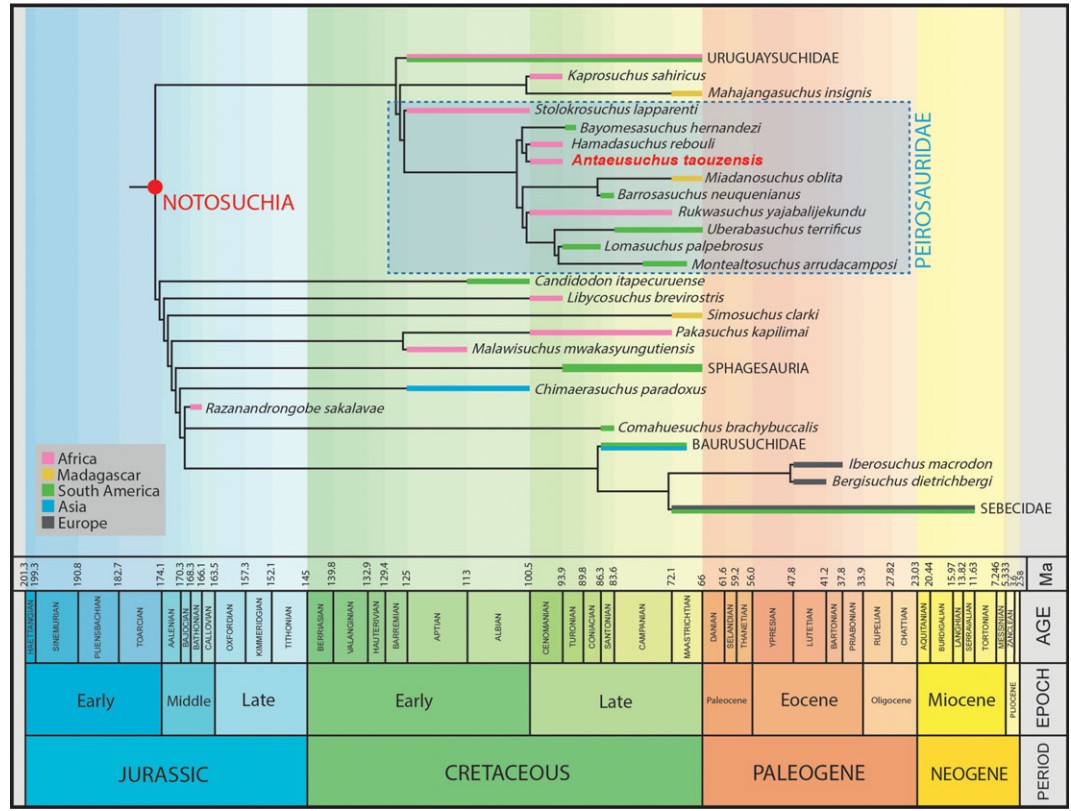

**Figure 11.** Time-calibrated phylogenetic topology showing the agreement subtree of notosuchians using an equal weighting of characters. Some clades are condensed and the polytomy including *Razanandrongobe sakalavae* is shown despite being pruned from the agreement subtree.

toothrow of *Ar. gomesii*, which he described as 'weakly serrated' and 'less leaf-shaped'. It is unclear why Kellner [123] limited comparisons to *Ar. gomesii*; nonetheless, the description of additional specimens of *Ar. wegeneri* from Niger [28], along with other species of this genus (e.g. [28,37,90,116,124,125]), allows for more thorough comparisons with the teeth from Cameroon. Given that none of the South American *Araripesuchus* species, nor *Araripesuchus tsangatsangana*, have denticles, the labiolingually compressed, the lanceolate shape of these teeth, with serrated carinae along their posteriormost and anteriormost margins, is supportive of a referral to either *Ar. wegeneri* or *Ar. rattoides* (the latter comparison is based on referred material, BSPG 2008 I 41, rather than the holotype specimen [37]). However, because of variation in crown morphology along the toothrow in all species of *Araripesuchus*, and given that teeth in the middle-to-posterior toothrow are either absent or poorly preserved in *Ar. rattoides*, it is not currently possible to provide a species-level referral.

The Albian Aïn el Guettar Formation in southern Tunisia (figure 12) has yielded numerous crocodyliform remains, including teeth assigned to *Ar. wegeneri*, *Araripesuchus* sp. and aff. *Hamadasuchus* sp. [126–128,143]. The specimens assigned to *Araripesuchus* ([127], fig. 4.7; [128], fig. 12U–X) are labiolingually compressed and triangular, with serrated carinae and relatively smooth enamel. Based on the slightly dorsoventrally constricted lanceolate shape of the teeth in lateral view, it is likely that they come from the middle region of the toothrow. All of these features support their referral to *Araripesuchus*, widening the spatial distribution of the genus to north-central Africa. Although serrated tooth margins are known to be present in *Ar. wegeneri* and a referred specimen of *Ar. rattoides* [37], we refrain from assigning these specimens beyond the generic level as was 'cautiously' proposed by Cuny *et al.* [127, p. 625] for the same reasons outlined in the preceding paragraph. A single tooth referred to aff. *Hamadasuchus* sp. is labiolingually compressed and approximately triangular in lateral view, with 'remnants of clear serration' ([127]: fig. 4.8, p. 625). Although the more extreme labiolingual compression towards the anterior and posterior margins of the tooth is reminiscent of *Hamadasuchus*, the apparent lack of rugose enamel is unusual given its presence in all teeth associated with the holotype specimen of *Hamadasuchus*. The only other named crocodyliforms from the Early Cretaceous of Africa to possess serrated carinae are *Ar. wegeneri* and

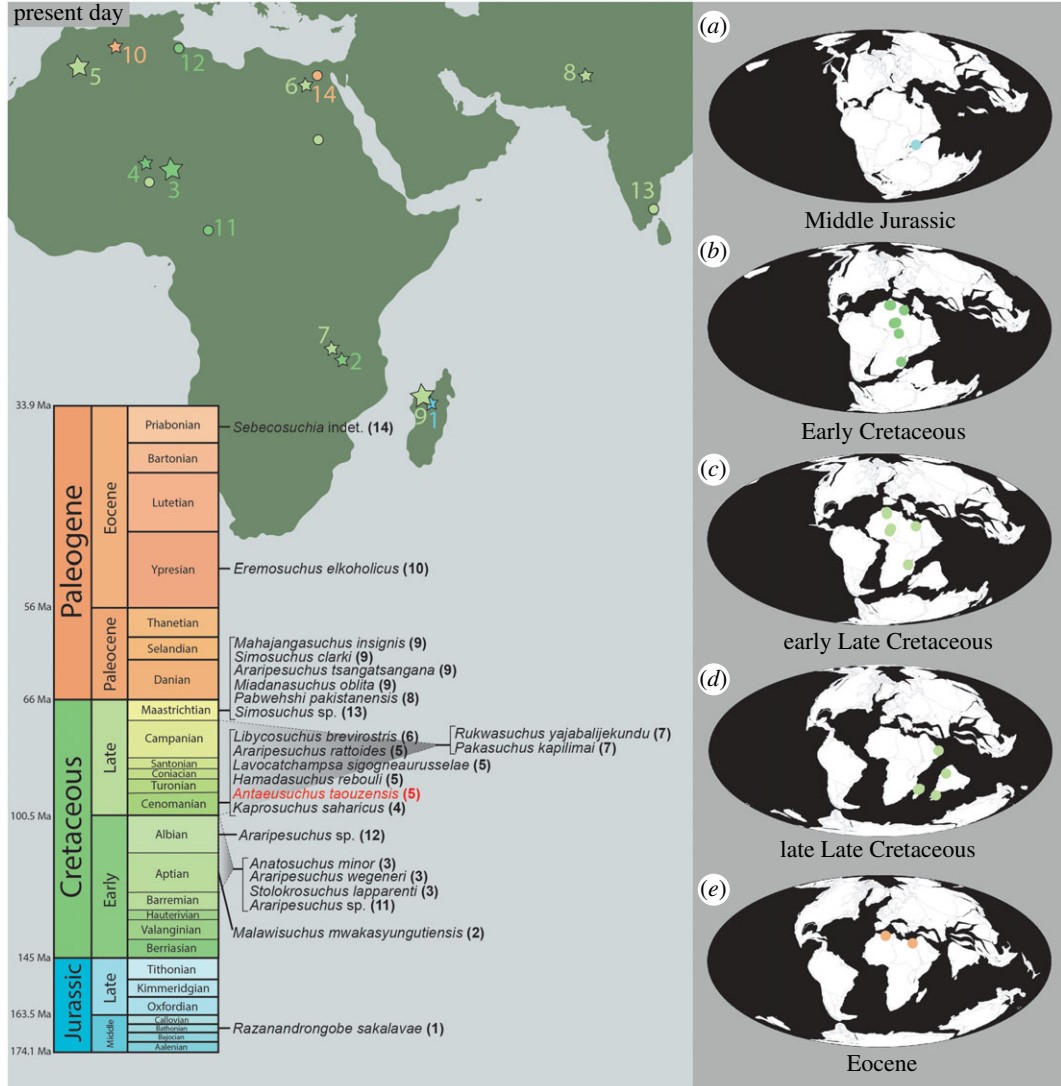

**Figure 12.** Spatio-temporal distribution of notosuchian occurrences from Africa and Indo-Madagascar. (*a*) Present-day map and stratigraphic column. Stars indicate the location of all named notosuchian taxa. The size of each star is proportional to the number of named taxa at each locality. Circles indicate other remains referred to Notosuchia. (*b*–*f*) Palaeogeographic reconstructions showing the distribution of notosuchian occurrences for the Middle Jurassic (*b*), Early Cretaceous (*c*), early Late Cretaceous (*d*), late Late Cretaceous (*e*) and Eocene (*f*). Plots modified from the Paleobiology Database Navigator (https://paleobiodb.org/navigator/).

referred material of *Ar. rattoides*, both of which possess dentition more similar in size to the tooth reported in Cuny *et al*. [127]. However, given that the Tunisian specimen is clearly well-worn and is not dissimilar in broad morphology from either *Hamadasuchus* or *Araripesuchus*, we regard this specimen as an indeterminate notosuchian. Re-evaluation and full description of specimens referred to *Hamadasuchus* that have teeth with smooth enamel (e.g. BSPG 2005 I 83) might enable referral to a particular genus.

### 6.3.3. Late Cretaceous

In northwestern Africa, the Cenomanian Kem Kem Group of Morocco has yielded three previously named notosuchian species (*Araripesuchus rattoides*, *Hamadasuchus rebouli*, *Lavocatchampsa sigogneaurussellae*), in addition to the new species, *An. taouzensis*, described herein (figure 12). *Araripesuchus rattoides* is known from the holotypic partial dentary, as well as several referred dentary fragments [28,37]. It differs from *Ar. wegeneri* in several features, including its possession of a

narrower and deeper snout, highly procumbent teeth and potentially a greater number of teeth. Though not included in our phylogenetic analyses because of its fragmentary nature, *Ar. rattoides* was recovered by Sereno & Larsson [28] in a polytomy with three *Araripesuchus* species (*Ar. gomesii* and *Ar. patagonicus* from South America, and *Ar. tsangatsangana* from Madagascar), with this the sister group to a clade comprising the remaining uruguaysuchids (including *Ar. wegeneri*). *Lavocatchampsa sigogneaurussellae* was erected based on a small anterior snout region, which displays unusually heterodont teeth that are convergent with those of mammals [38]. Using the data matrix of Pol *et al*. [6], Martin & Lapparent de Broin [38] recovered *Lavocatchampsa* as a 'basal' ziphosuchian within Candidodontidae, a small clade otherwise known only from the Cretaceous of South America [75,129].

As discussed in detail above, the peirosaurid *H. rebouli* was erected based on a single dentary fragment from the Kem Kem Group [43], but numerous cranial and mandibular remains have since been referred to this species from this stratigraphic unit [37,46], including a skull table previously assigned to *Libycosuchus* sp. [96,97]. Although we do not disagree with the referral of these remains to Peirosauridae, it is currently unclear if all of them are attributable to *H. rebouli*. Isolated teeth described by Larsson & Sidor [44] were also referred to this species. One tooth, inferred to be from the middle of the toothrow ([64]: fig. 1b), is very reminiscent of those preserved in the holotype of *H. rebouli*, based on its triangular shape in lateral view, its labiolingual compression, and the density of serrations. A second tooth shares the globular texture of the enamel towards the base of the crown, which transitions into more linear ridges towards the apex ([44]: fig. 1c), which is again consistent with a referral to *H. rebouli*. However, a conical, retro-curved caniniform tooth shows distinctive fluting ([44]: fig. 1a), which is absent from the holotypic specimen, but present in some of the specimens previously referred to the species (ROM 49282 and 52620, BSPG 2005 I 83 and possibly NMC 41892 [37]).

Larsson & Sidor [44] described several additional crocodyliform teeth from the Kem Kem Group that have been suggested to represent additional notosuchian taxa [37]. Material referred to 'Indet. crocodyliform 1' [44, p. 398] is represented by two small, sub-triangular crowns (SGM-Rep 4, SGM-Rep 5) in labiolingual view, each with three approximately parallel rows of relatively large cuspids that terminate in angular apices towards the anteroposterior midpoint of the tooth ([44], fig. 2a–d). On one of these teeth, a large planar wear facet bisects the rows of cusps on the buccal surface. A third tooth (SGM-Rep 6) referred to by Larsson & Sidor [44, p. 399] as 'Indet. crocodyliform 2' is more elliptical in dorsal view, and has a central, anteroposterior row of cuspids surrounded labially and lingually by two less dorsally raised rows of smaller cusps ([44], fig. 3). The outer two rows merge at the anteriormost and posteriormost margins of the tooth, forming a cingulum. Unlike the other multicuspid teeth, the rows of cusps in this third tooth are much closer to horizontal in their orientation, forming a less acute apex. Furthermore, the cusps of the central row are relatively larger in comparison to the tooth size and are fewer in number, forming an apex either mesially or distally (depending on tooth orientation in the jaw) rather than centrally. *Lavocatchampsa sigogneaurussellae* is the only crocodyliform from the Kem Kem Group that exhibits a multicuspid tooth morphology [38]; however, we agree with the observations of Ibrahim *et al*. [37] that both morphologies are distinct from this taxon. We do note that the less acute tooth described as 'Indet. crocodyliform 2' is most similar in its morphology to the taxon described by Martin & de Lapparent de Broin [38] based on its elliptical shape in occlusal view, and the presence of a cingulum bearing multiple cuspids that surround a central carina formed of a relatively small number (four) of cusps.

In summary, the Kem Kem Group seems to record the presence of at least seven potential notosuchians, represented by three peirosaurids (*Hamadasuchus rebouli*, *Antaeusuchus taouzensis*, and at least one unnamed species), *Araripesuchus rattoides*, and three species with multicuspid teeth (*Lavocatchampsa sigogneaurussellae* and two unnamed species). However, given poor stratigraphic constraints for many of these species, it remains possible that these were not all contemporaneous.

The Cenomanian Bahariya Formation of north-central Egypt has yielded just a single notosuchian species (figure 12), with *Libycosuchus brevirostris* represented by a complete skull and lower jaws, as well as several isolated vertebrae [52,130]. *Libycosuchus* has an anteroposteriorly short skull and is fairly consistently recovered as an early diverging ziphosuchian (e.g. [6,38,63,85]), as is also the case in our analyses.

The large-bodied species *Kaprosuchus saharicus* is the only published notosuchian currently known from the Cenomanian Echkar Formation of northwestern Niger [28] (figure 12). This species is based on an essentially complete skull and mandible and has been informally referred to as the 'boar croc' due to its enlarged caniniform teeth. *Kaprosuchus* has been consistently recovered as the sister taxon to *Mahajangasuchus insignis* from the Maastrichtian of Madagascar (see below), and is thus a member of

Mahajangasuchidae (e.g. [6,28,85]; this analysis) (figure 6). Sereno & Pol [131] reported an undescribed partial skeleton from the Echkar Formation that appears to be most closely related to the Maastrichtian Malagasy species, *Araripesuchus tsangatsangana*.

In the southeastern region of Africa, two notosuchian taxa are known from the Namba Member of the Galula Formation of western Tanzania (figure 12). Originally thought to be Aptian–Cenomanian [18,132], new dates indicate either a Cenomanian–Santonian or more likely a Campanian age for this stratigraphic unit [133]. Represented by the posterior region of the skull, the medium-to large-bodied *Rukwasuchus yajabalijekundu* was recovered by Sertich & O'Connor [63] as a peirosaurid. It had not been included in a subsequent phylogenetic analysis prior to ours, which provides further support for a peirosaurid placement (figures 6 and 7). Known from an essentially complete skeleton, *Pakasuchus kapilimai* is one of several small notosuchians with multicuspid teeth from the Cretaceous of Gondwana that appears to fill an ecological niche that would later be occupied by mammals [18,63]. As is the case in several previous studies (e.g. [6,18,38,63]), our analyses recover *Pakasuchus* as an early diverging member of Ziphosuchia, closely related to *Malawisuchus* (figure 7).

The Coniacian–Santonian In Beceten Formation of Niger (figure 12) has yielded the type material (an incomplete lacrimal) of *T. taqueti* [95]. As discussed in §6.2, Meunier & Larsson [100] demonstrated that *T. taqueti* is a nomen dubium, and suggested that isolated bones and teeth informally referred to the taxon represent at least three different small-medium sized crocodyliform species. They noted that many of these specimens show potential affinities to peirosaurids (especially *Hamadasuchus*), uruguaysuchids (especially *Araripesuchus wegeneri* and *Anatosuchus minor*), ziphosuchians and/or neosuchians, which we follow here.

The Wadi Milk Formation of northern Sudan (figure 12) has traditionally been regarded as Cenomanian (e.g. [134]), but more recent work indicates that it should be assigned to the Campanian–Maastrichtian [135]. An undescribed peirosaurid has been briefly reported, consisting of partial mandibles and part of the skull roof, and which is notable for its large size [136].

Putative notosuchian remains from the Maastrichtian Dukamaje Formation in western Niger have been mentioned in the literature, but not described. Moody & Sutcliffe [137]) listed the presence of *T. taqueti* and *Libycosuchus* sp. in this formation, but they provided no further details. We suspect that this was a mistake, with the In Beceten faunal list accidentally incorporated, but this cannot currently be confirmed.

A mandibular fragment preserving the middle portion of a right dentary could potentially represent the only occurrence of a notosuchian from the Arabian Peninsula [138]. Buscalioni *et al.* [16] tentatively assigned the specimen from the Maastrichtian Al-Khod Conglomerate Formation of northern Oman as cf. *Trematochampsa* indet. and noted similarities with *Miadanosuchus* (*Trematochampsa*) *oblita* from the Maastrichtian of Madagascar. These similarities included the presence of an enlarged tooth in the 10th alveolus, and a morphology indicative of a long and wide mandibular symphysis (despite this region not being preserved). Our analyses recover *Miadanosuchus* within Peirosauridae, a clade characterized by two distinct waves on the dorsal margin of the dentary. The very straight, only slightly inclined dorsal edge of the dentary in the Oman specimen is, therefore, not indicative of a specimen belonging to this clade, especially as the presence of an enlarged tooth would be expected to be accompanied by the dorsoventral expansion of the dentary. Furthermore, the dentary of *Miadanosuchus* maintains its mediolateral width posterior to the enlarged tenth tooth for at least the distance of two alveoli. The Oman specimen shows gradual, but distinct narrowing posterior to the enlarged tooth. Given the highly fragmentary nature of the specimen, and the few preserved anatomical features of phylogenetic relevance, we suggest that the material can only be assigned to an indeterminate crocodyliform.

The Maastrichtian Maevarano Formation that outcrops in northwestern Madagascar has thus far yielded four notosuchian taxa (figure 12). The bizarre 'pug-nosed' *Simosuchus clarki* is represented by multiple individuals preserving most of the skeleton [8,10,139,140]. Most analyses recover *Simosuchus* as an early diverging ziphosuchian (e.g. [6,85,105,141]; this study). The large-bodied *Ma. insignis* is known from an almost complete skull and much of the postcranial skeleton [81,92]. Initially thought to have affinities with 'Trematochampsidae', the taxon has since been recovered as a peirosaurid (e.g. [103]), or just outside of this clade (e.g. [6]). It is now the clade specifier for Mahajangasuchidae [28], with our analyses providing evidence for a position both within (EIW) and just outside (EQW) of Peirosauridae. *Araripesuchus tsangatsangana* is represented by a nearly complete skull, as well as a second individual preserving a nearly complete skeleton [118]. *Miadanasuchus* (*Trematochampsa*) *oblita* is known from partial dentaries, part of the skull roof, and a vertebra [64,94], and it appears to represent a peirosaurid (Geroto & Bertini [85]; this study).

*Pabwehshi pakistanensis*, recovered from the Maastrichtian Pab Formation of southwestern Pakistan (figure 12), is known from fragmentary specimens, which preserve the anterior region of the snout and the associated section of the mandible of two individuals [117]. The limited remains of *Pabwehshi* mean that its phylogenetic position is labile [6], although most authors have supported a close relationship with Baurusuchidae (e.g. [25,60,76,85,103,105,117]), which is otherwise known only from South America (e.g. [142,143]). By contrast, Larsson & Sues [46] recovered *Pabwehshi* as the most 'basal' member of Sebecia, i.e. as the sister taxon to a clade comprising Peirosauridae and Sebecidae. *Pabwehshi pakistanensis* was excluded from our analyses because of its labile position, but more complete material will ultimately be needed to robustly resolve its phylogenetic position.

An isolated tooth from the Maastrichtian Kallamedu Formation of southern India (figure 12) was described by Prasad *et al.* [144], who identified it as cf. *Simosuchus* sp. Based on comparisons with *S. clarki*, Prasad *et al.* [144] suggested that the tooth is probably from the posterior region of the dentary. We fully agree with the evaluation and assignment of this specimen.

### 6.3.4. Palaeogene

Buffetaut [102] erected *Eremosuchus elkoholicus* from the El Kohol Formation of southwest Algeria (figure 12), which is dated to the Ypresian, early Eocene [142]. This species is known from a partial mandible, teeth, vertebrae and a fibula. When initially described, *Eremosuchus* was placed in the family Trematochampsidae [102], but more recently it has been included in Sebecosuchia by several authors (e.g. [73,79,103]). However, it has not been included in most phylogenetic analyses, presumably because of its incomplete nature, and has largely been neglected in treatments of crocodyliform evolutionary history. A detailed redescription and analysis of the phylogenetic relationships of *Eremosuchus* is needed to establish its systematic and biogeographic affinities.

Finally, the late Eocene Birket Qarun Formation [145] in northeastern Egypt (figure 12) has yielded a fragmentary right dentary with ziphodont dentition [30]. Though not assigned to a genus, the specimen clearly has sebecosuchian affinities and extends the temporal range of Notosuchia in Africa [30].

### 6.3.5. Summary

Our review of the Gondwanan record of notosuchians outside of South America demonstrates their spatio-temporal distribution in the Middle Jurassic, from the Aptian–Maastrichtian and in the Eocene, with their remains known from Africa and Indo-Madagascar. A possible occurrence from the latest Cretaceous of Oman [139] cannot be confidently referred to Notosuchia. The African and Indo-Madagascan Cretaceous record indicates the presence of several lineages, all with close ties to South American clades, with many faunas demonstrating multiple sympatric species. Given that notosuchians only first appeared in the Aptian in South America (and Asia), coupled with palaeogeographic reconstructions documenting the increasing fragmentation of Gondwana at this time (e.g. [146]), this diverse record supports previous suggestions regarding an undocumented pre-Aptian radiation of Notosuchia (e.g. [26,38]). By contrast, their Gondwanan Palaeogene record outside of South America is currently limited to just two occurrences, both from the Eocene of north Africa and both belonging to Sebecosuchia. No stratigraphically younger remains have been assigned to Notosuchia from this region, with their last Laurasian occurrences from the middle Eocene of western Europe (e.g. [113]), indicating their extirpation outside of South America by the end of the Eocene.

# 7. Conclusion

Two new crocodyliform specimens from the Cenomanian Kem Kem Group of Morocco are described and incorporated into a phylogenetic analysis. Both specimens are referrable to *Antaeusuchus taouzensis* n. gen. n. sp., which is recovered within the notosuchian clade Peirosauridae, as the sister taxon to the contemporaneous *H. rebouli*. Comparisons of materials previously assigned to *Hamadasuchus* indicate the presence of at least three distinct peirosaurid species from the same spatio-temporal interval. Coupled with a critical reappraisal of the non-South American Gondwanan record of Notosuchia, we recognize a much greater taxonomic and ecomorphological diversity within this clade during the Cretaceous.

Data accessibility. The datasets supporting this article have been uploaded as part of the electronic supplementary material.

The data are provided in the electronic supplementary material [147].

Authors' contributions. C.S.C.N and P.D.M. conceived of the study and interpreted the results. All authors contributed to the design of the study and to the drafting of the manuscript. Analyses were conducted by C.S.C.N. Figures were produced by C.S.C.N. and E.S.E.H. All authors approved the final version of the manuscript.

Competing interests. We declare we have no competing interests.

Funding. C.S.C.N. is funded by a Royal Society research grant (RGF\R1\180020) awarded to P.D.M. E.S.E.H. received funding from a Palaeontological Association Undergraduate Research Bursary (grant no. PA-UB201804) and her work is supported by a Natural Environment Research Council studentship (grant no. NE/S007415/1). P.D.M.'s contribution was supported by grants from the Royal Society (grant nos UF160216, RGF\R1\180020 and RGF\EA\201037).

Acknowledgements. We are grateful to Paul Barrett and Susannah Maidment for their help in providing access to NHMUK PV R36829 and R36874, as well as to Kevin Webb for providing photographs of the specimens (all at the Natural History Museum, London). Paul Barrett also provided help with species name formulation. Access to specimens of *Hamadasuchus rebouli* and other peirosaurid material from Morocco was provided by David Evans and Brian Iwama (both Royal Ontario Museum, Toronto), as well as Oliver Rauhut (Bayerische Staatssammlung für Paläontologie und Geologie, Munich), to whom we are also grateful. Photographs of other specimens referred to *Hamadasuchus rebouli* were kindly provided by Diego Pol (Museo Paleontológico Egidio Feruglio, Trelew) including that used in figure 10*c*. We also acknowledge the Willi Hennig Society, which has sponsored the development and free distribution of TNT. Finally, we are grateful for comments provided by Mario Bronzati and one anonymous reviewer that helped to improve the quality of this manuscript.

# Appendix A

Character scores modified from the respective matrices of Martínez *et al*. [48] and are listed below:

*Hamadasuchus rebouli*:

**103** ? -> 0; **363** ? -> 0; **365** ? -> 0; **383** ? -> 0; **384** ? -> 0; **388** ? -> 0; **389** ? -> 0; **392** ? -> 1; **393** ? -> 0&1; **394** ? -> 0; **443** 0 -> 1

*Gasparinisuchus peirosauroides*:

**443** 0 -> 0&1

*Montealtosuchus arrudacamposi*:

**443** 0 -> 1

*Libycosuchus brevirostris*:

**441** ? -> 0

*Malawisuchus mwakasyungutiensis*:

**441** ? -> 0

*Caipirasuchus stenognathus*:

**441** ? -> 0

*Caipirasuchus montealtensis*:

**441** ? -> 0

*Baurusuchus salgadoensis*:

**441** ? -> 0

*Stolokrosuchus lapparenti*:

**44** 1 ? -> 0

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
