## [Peer Review File · Royal Society Open Science]

Review History

RSOS-211254.R0 (Original submission)

Review form: Reviewer 1 (Felipe Montefeltro)

Is the manuscript scientifically sound in its present form?

Yes

Are the interpretations and conclusions justified by the results?

Yes

Is the language acceptable?

Yes

Do you have any ethical concerns with this paper?

No

Have you any concerns about statistical analyses in this paper?

No

Recommendation?

Accept with minor revision (please list in comments)

Comments to the Author(s)

The MS "A second peirosaurid crocodyliform from the mid-Cretaceous Kem Kem Group of Morocco and the diversity of Gondwanan notosuchians outside South America" by Nichols et al. describes a new taxon based on two partial lower jaws that increases the understanding of notosuchian diversity in continental Africa. The MS also includes a revised phylogenetic analysis with an increased taxon sampling of peirosaurids, and a revision of African notosuchians.

The MS adds relevant information and deserve to be published by RSOS after minor review.

I did not perform a language review because I do not feel I am qualified for that. Although, from what I can judge, the language is appropriate for the scientific writing.

Perhaps the most important issue that I raise is related to the α -taxonomy of the peirosaurids from the Kem Kem beds. One of the main goals of the MS is to diagnose the new taxon *Antaeusuchus* and differentiate it from *Hamadasuchus*. However, it is not clear for me which specimens referred to *Hamadasuchus* the authors considered for their analysis. If I understood correctly, the OTU *Hamadasuchus* in their phylogenetic analysis is composed by the fragmentary holotype (MDE C001) and the skull (ROM 52620) described by Larsson & Sues (2007), while in along the comparison section, the authors also compare to a broader sampling of specimens (e.g. BSPG 2005 I 83, ROM 49282, etc). In general, I would be ok with such strategy, in fact, I used a similar *Hamadasuchus* OTU in my own phylogenetic analyses. The problem is that the preserved parts of the holotype of the taxon *Hamadasuchus* and the referred specimen ROM 52620 do not overlap. Also, Ibrahim et al. (2020) suggests that several specimens referred to *Hamadasuchus* present a morphological variation with possible taxonomical implications.

The new taxon *Antaeusuchus* can be differentiated from the holotype of *Hamadasuchus* but both cannot be differentiated from the specimen ROM 52620. I would say that the best strategy at this point is to restrict the *Hamadasuchus* OTU to only the holotype of the taxon awaiting further revision of the remaining specimens.

I detected the lack of explicit definitions of most clades used in the MS, for example figure 7. Which definition of *Notosuchia* and *Ziphosuchia* were used? In addition, which definition for *Metasuchia* was used? I was not able to find any definition in recent papers.

Other minor issues are:

Although it is not particularly relevant, the revised phylogeny presented by the MS is not the largest notosuchian-focused character-taxon matrices yet to be compiled.

We have proposed a phylogenetic definition for *Notosuchia* in our paper (Ruiz, et al. 2021), perhaps include it on the Systematic Palaeontology section.

In relation to the anatomy of *Antaeusuchus*, I suggest the authors to take a look on Pinheiro's et al. (2020, Plos One) description of the enamel of *Roxochampsia*.

Which are the parameters used during sectorial searches, drift and tree fusing?

I did not understand what the authors mean when they say that *Uberabasuchus* can be used as a proxy for *Peirosaurus*?

In the discussion about the multicusped teeth described by Larsson & Sidor (1999), I suggest the authors to also take a look on the papers by Montefeltro et al. (2009) and Pinheiro et al. (2021, *Coronelsuchus*).

The reference Evans et al. 2014 cited in the text is not listed. There is a reference cited as Montefeltro et al. (2019). I guess it is Montefeltro (2019, JVP) or Montefeltro et al. (2020, J. Anato).

Review form: Reviewer 2 (Mario Bronzati)

Is the manuscript scientifically sound in its present form?

Yes

Are the interpretations and conclusions justified by the results?

Yes

Is the language acceptable?

Yes

Do you have any ethical concerns with this paper?

No

Have you any concerns about statistical analyses in this paper?

No

Recommendation?

Accept with minor revision (please list in comments)

Comments to the Author(s)

Nicholl et al. describe a new extinct crocodyliform taxon from the famous Kem Kem Beds – Cretaceous of Morocco, and included this taxon as an OTU in a new phylogenetic analysis for Crocodyliformes, focusing on notosuchians. Furthermore, the authors also provide a review of the African notosuchians (see main concerns below).

As for many of the taxa from the Kem Kem, the two specimens assigned to the new taxon, *Antaeusuchus taouzensis*, are fragmentary/incomplete. Nevertheless, the authors did a great job in providing substantial comparisons between their material and other peirosaurids in order to consolidate the status of *Antaeusuchus* as a valid species, separated from other peirosaurids (but see below). This is important, given that their phylogenetic analyses clearly indicate the position of the new taxon within Peirosauridae.

I make most of my comments in the PDF file accompanying this review (see Appendix A). As the authors can see in there, I think that some parts of the description could be improved (e.g. by providing a more general description of each bone before going into details of their anatomy; providing more details and indications in some of the figures). Below, I make some comments that do not represent criticism of the work done by the authors, but I think that these represent points that the authors should take in consideration when submitting the revised version of their manuscript.

1. The authors used two different protocols for their phylogenetic analyses, one employing equally weighting of characters, and the other employed the extend implied weighting protocol. I do think it is interesting to see the results using two alternative methods. However, if two alternative methods are used in the paper, I think that the authors should then explain if they prefer any of the methods employed. If they do prefer one of the two methods, justify. If no, just

mention that you are using different methods because there is no study so far that says that we should 'definitely' opt for one instead of the other.

In this context, sentence like "we applied extend implied weighting to notosuchians for the first time" are not very relevant, especially when the authors use the phylogeny obtained from their analysis using equal weighting in their Figure 11. In this case, I think there should be a justification for the reason why you selected the equal weighting analysis for this figure.

2. The authors did a good job in providing comparisons between the new species with other peirosaurids. Also, by the end of this section, they list the differences between *Antaeusuchus* and the other peirosaurids from the Kem Kem, *Hamadasuchus*. The authors mention that one of the differences is that specimens assigned to *Antaeusuchus* are much larger than species assigned to *Hamadasuchus*. Thus, I think that an interesting addition to their study would be to try ruling out the possibilities that the differences between *Antaeusuchus* and *Hamadasuchus* are not related to ontogeny - i.e. that individuals of *Antaeusuchus* do not correspond to larger individuals of *Hamadasuchus*. I'm not sure it is possible to check this for all the different characteristics they mentioned based on the differences observed in extant crocodylians - but anyway, this might strength their argument to separate the two species.

3. I have mixed feelings about the last section of the discussion, which brings a revision of the notosuchians outside South America. Whereas I think it is interesting to see this kind of information compiled in a single study, I do not think that the authors used the information already available in the literature in order to provide any new insight on the evolution of African notosuchians. For example, the section dealing with the presence of *Razanandrongobe* in the Middle Jurassic of Madagascar brings no novel information that is worth being included in the discussion section of this manuscript. In sum, there is not much of new insights or new perspectives in all the sections of this part of the discussion that would justify the inclusion of this part of the manuscript together with the description of the new taxon and discussions on the phylogenetic analysis of peirosaurids. For example, section 6.3.5 would better fit as part of an introduction of a manuscript on the fossil record of notosuchians.

I like to read 'classic' palaeontological studies, on which the authors present a new taxon, discuss aspects of alpha taxonomy, and show details about the phylogenetic context of the new taxon. And, I think the authors did an excellent job in this regard, in a way that the last section seems a bit dislocated from the rest of the manuscript.

I hope that my review can be helpful and congratulate the authors on their work!

With best wishes
Mario Bronzati

Decision letter (RSOS-211254.R0)

Dear Miss Nicholl

On behalf of the Editors, we are pleased to inform you that your Manuscript RSOS-211254 "A second peirosaurid crocodyliform from the mid-Cretaceous Kem Kem Group of Morocco and the

diversity of Gondwanan notosuchians outside South America" has been accepted for publication in Royal Society Open Science subject to minor revision in accordance with the referees' reports. Please find the referees' comments along with any feedback from the Editors below my signature. One of the reviewers (reviewer 2) has kindly supplied a document along with their comments - I have attached the document to this email, but please let the editorial office know if you do not receive it.

Please submit your revised manuscript and required files (see below) no later than 7 days from today's (ie 02-Sep-2021) date. Note: the ScholarOne system will 'lock' if submission of the revision is attempted 7 or more days after the deadline. If you do not think you will be able to meet this deadline please contact the editorial office immediately.

on behalf of Dr Jennifer Botha (Associate Editor) and Kevin Padian (Subject Editor)
openscience@royalsociety.org

Associate Editor Comments to Author (Dr Jennifer Botha):

Associate Editor: 1

Comments to the Author:

The reviewers have both provided positive feedback for your paper and both suggest minor revision. There are several technical details that should be carefully considered and explained regarding the phylogenetic analyses. One reviewer also suggested deleting the review section in the discussion, but if the authors can provide justification for keeping it, I encourage them to do so

Reviewer comments to Author:

Reviewer: 1

Comments to the Author(s)

The MS "A second peirosaurid crocodyliform from the mid-Cretaceous Kem Kem Group of Morocco and the diversity of Gondwanan notosuchians outside South America" by Nichols et al. describes a new taxon based on two partial lower jaws that increases the understanding of notosuchian diversity in continental Africa. The MS also includes a revised phylogenetic analysis with an increased taxon sampling of peirosaurids, and a revision of African notosuchians.

The MS adds relevant information and deserve to be published by RSOS after minor review. I did not perform a language review because I do not feel I am qualified for that. Although, from what I can judge, the language is appropriate for the scientific writing.

Perhaps the most important issue that I raise is related to the α -taxonomy of the peirosaurids from the Kem Kem beds. One of the main goals of the MS is to diagnose the new taxon *Antaeusuchus* and differentiate it from *Hamadasuchus*. However, it is not clear for me which specimens referred to *Hamadasuchus* the authors considered for their analysis. If I understood correctly, the OTU *Hamadasuchus* in their phylogenetic analysis is composed by the fragmentary holotype (MDE C001) and the skull (ROM 52620) described by Larsson & Sues (2007), while in along the comparison section, the authors also compare to a broader sampling of specimens (e.g. BSPG 2005 I 83, ROM 49282, etc). In general, I would be ok with such strategy, in fact, I used a similar *Hamadasuchus* OTU in my own phylogenetic analyses. The problem is that the preserved parts of the holotype of the taxon *Hamadasuchus* and the referred specimen ROM 52620 do not overlap. Also, Ibrahim et al. (2020) suggests that several specimens referred to *Hamadasuchus* present a morphological variation with possible taxonomical implications.

The new taxon *Antaeusuchus* can be differentiated from the holotype of *Hamadasuchus* but both cannot be differentiated from the specimen ROM 52620. I would say that the best strategy at this point is to restrict the *Hamadasuchus* OTU to only the holotype of the taxon awaiting further revision of the remaining specimens.

I detected the lack of explicit definitions of most clades used in the MS, for example figure 7. Which definition of *Notosuchia* and *Ziphosuchia* were used? In addition, which definition for *Metasuchia* was used? I was not able to find any definition in recent papers.

Other minor issues are:

Although it is not particularly relevant, the revised phylogeny presented by the MS is not the largest notosuchian-focused character-taxon matrices yet to be compiled.

We have proposed a phylogenetic definition for *Notosuchia* in our paper (Ruiz, et al. 2021), perhaps include it on the Systematic Palaeontology section.

In relation to the anatomy of *Antaeusuchus*, I suggest the authors to take a look on Pinheiro's et al. (2020, Plos One) description of the enamel of *Roxochampsia*.

Which are the parameters used during sectorial searches, drift and tree fusing?

I did not understand what the authors mean when they say that *Uberabasuchus* can be used as a proxy for *Peirosaurus*?

In the discussion about the multicusped teeth described by Larsson & Sidor (1999), I suggest the authors to also take a look on the papers by Montefeltro et al. (2009) and Pinheiro et al. (2021, *Coronelsuchus*).

The reference Evans et al. 2014 cited in the text is not listed.

There is a reference cited as Montefeltro et al. (2019). I guess it is Montefeltro (2019, JVP) or Montefeltro et al. (2020, J. Anato).

Reviewer: 2

Comments to the Author(s)

Nicholl et al. describe a new extinct crocodyliform taxon from the famous Kem Kem Beds - Cretaceous of Morocco, and included this taxon as an OTU in a new phylogenetic analysis for

Crocodyliformes, focusing on notosuchians. Furthermore, the authors also provide a review of the African notosuchians (see main concerns below).

As for many of the taxa from the Kem Kem, the two specimens assigned to the new taxon, *Antaeusuchus taouzensis*, are fragmentary/incomplete. Nevertheless, the authors did a great job in providing substantial comparisons between their material and other peirosaurids in order to consolidate the status of *Antaeusuchus* as a valid species, separated from other peirosaurids (but see below). This is important, given that their phylogenetic analyses clearly indicate the position of the new taxon within Peirosauridae.

I make most of my comments in the PDF file accompanying this review. As the authors can see in there, I think that some parts of the description could be improved (e.g. by providing a more general description of each bone before going into details of their anatomy; providing more details and indications in some of the figures). Below, I make some comments that do not represent criticism of the work done by the authors, but I think that these represent points that the authors should take in consideration when submitting the revised version of their manuscript.

1. The authors used two different protocols for their phylogenetic analyses, one employing equally weighting of characters, and the other employed the extend implied weighting protocol. I do think it is interesting to see the results using two alternative methods. However, if two alternative methods are used in the paper, I think that the authors should then explain if they prefer any of the methods employed. If they do prefer one of the two methods, justify. If no, just mention that you are using different methods because there is no study so far that says that we should 'definitely' opt for one instead of the other.

In this context, sentence like "we applied extend implied weighting to notosuchians for the first time" are not very relevant, especially when the authors use the phylogeny obtained from their analysis using equal weighting in their Figure 11. In this case, I think there should be a justification for the reason why you selected the equal weighting analysis for this figure.

2. The authors did a good job in providing comparisons between the new species with other peirosaurids. Also, by the end of this section, they list the differences between *Antaeusuchus* and the other peirosaurids from the Kem Kem, *Hamadasuchus*. The authors mention that one of the differences is that specimens assigned to *Antaeusuchus* are much larger than species assigned to *Hamadasuchus*. Thus, I think that an interesting addition to their study would be to try ruling out the possibilities that the differences between *Antaeusuchus* and *Hamadasuchus* are not related to ontogeny - i.e. that individuals of *Antaeusuchus* do not correspond to larger individuals of *Hamadasuchus*. I'm not sure it is possible to check this for all the different characteristics they mentioned based on the differences observed in extant crocodylians - but anyway, this might strength their argument to separate the two species.

3. I have mixed feelings about the last section of the discussion, which brings a revision of the notosuchians outside South America. Whereas I think it is interesting to see this kind of information compiled in a single study, I do not think that the authors used the information already available in the literature in order to provide any new insight on the evolution of African notosuchians. For example, the section dealing with the presence of *Razanandrongobe* in the Middle Jurassic of Madagascar brings no novel information that is worth being included in the discussion section of this manuscript. In sum, there is not much of new insights or new perspectives in all the sections of this part of the discussion that would justify the inclusion of this part of the manuscript together with the description of the new taxon and discussions on the phylogenetic analysis of peirosaurids. For example, section 6.3.5 would better fit as part of an introduction of a manuscript on the fossil record of notosuchians.

I like to read 'classic' palaeontological studies, on which the authors present a new taxon, discuss aspects of alpha taxonomy, and show details about the phylogenetic context of the new taxon. And, I think the authors did an excellent job in this regard, in a way that the last section seems a bit dislocated from the rest of the manuscript.

I hope that my review can be helpful and congratulate the authors on their work!

With best wishes
Mario Bronzati

===PREPARING YOUR MANUSCRIPT===

===PREPARING YOUR REVISION IN SCHOLARONE===

Author's Response to Decision Letter for (RSOS-211254.R0)

See Appendix B.

Decision letter (RSOS-211254.R1)

Dear Miss Nicholl,

I am pleased to inform you that your manuscript entitled "A second peirosaurid crocodyliform from the mid-Cretaceous Kem Kem Group of Morocco and the diversity of Gondwanan notosuchians outside South America" is now accepted for publication in Royal Society Open Science.

on behalf of Dr Jennifer Botha (Associate Editor) and Kevin Padian (Subject Editor)
openscience@royalsociety.org

Appendix A**ROYAL SOCIETY
OPEN SCIENCE****A second peirosaurid crocodyliform from the mid-Cretaceous Kem Kem Group of Morocco and the diversity of Gondwanan notosuchians outside South America**

Journal:	Royal Society Open Science
Manuscript ID	RSOS-211254
Article Type:	Research
Date Submitted by the Author:	30-Jul-2021
Complete List of Authors:	Nicholl, Cecily; University College London, Department of Earth Sciences Hunt, Eloise S. E.; Natural History Museum, Department of Life Sciences; Imperial College London, the Department of Life Sciences Ouarhache, Driss; Université Sidi Mohamed Ben Abdellah, Laboratoire Géosystèmes, Environnement et Développement Durable. Département de Géologie Mannion, Philip; University College London, Earth Sciences
Subject:	Palaeontology < EARTH SCIENCES, palaeontology < BIOLOGY
Keywords:	Notosuchia, Crocodylomorpha, Gondwana, Kem Kem, Mesozoic, Africa
Subject Category:	Organismal and Evolutionary Biology

Author-supplied statements

Relevant information will appear here if provided.

Ethics

Does your article include research that required ethical approval or permits?:

This article does not present research with ethical considerations

Statement (if applicable):

CUST_IF_YES_ETHICS :No data available.

Data

It is a condition of publication that data, code and materials supporting your paper are made publicly available. Does your paper present new data?:

Yes

Statement (if applicable):

The datasets supporting this article have been uploaded as part of the electronic supplementary material.

Conflict of interest

I/We declare we have no competing interests

Statement (if applicable):

CUST_STATE_CONFLICT :No data available.

Authors' contributions

This paper has multiple authors and our individual contributions were as below

Statement (if applicable):

C.S.C.N and P.D.M. conceived of the study and interpreted the results. All authors contributed to the design of the study and to the drafting of the manuscript. Analyses were conducted by C.S.C.N. Figures were produced by C.S.C.N. and E.S.E.H. All authors approved the final version of the manuscript.

**A second peirosaurid crocodyliform from the mid-Cretaceous Kem Kem Group**
**of Morocco and the diversity of Gondwanan notosuchians outside South**
**America**

Cecily S. C. Nicholl^{1*}, Eloise S. E. Hunt^{2,3}, Driss Ouarhache⁴, Philip D. Mannion¹

¹ Department of Earth Sciences, University College London, Gower Street, London,
WC1E 6BT, UK.

² Department of Life Sciences, Natural History Museum, Cromwell Road, London,
SW7 5BD, UK.

³ Science and Solutions for a Changing Planet DTP, and the Department of Life
Sciences, Imperial College London, South Kensington Campus, London, SW7 2AZ,
UK.

⁴ Laboratoire Géosystèmes, Environnement et Développement Durable,
Département de Géologie, Faculté des Sciences Dhar El Mahraz, Université Sidi
Mohamed Ben Abdellah, BP 1796, Atlas, 30 000, Fès, Morocco

*Author for correspondence: Cecily S. C. Nicholl (cecily.nicholl@ucl.ac.uk)

RRH: SECOND MOROCCAN PEIROSAURID

LRH: NICHOLL, HUNT, OUARHACHE & MANNION

Abstract

Notosuchians are an extinct clade of terrestrial crocodyliforms with a particularly rich record in the late Early to Late Cretaceous of Gondwana. Although much of this diversity comes from South America, Africa and Indo-Madagascar have also yielded numerous notosuchian remains. Three notosuchian species are currently recognised from the early Late Cretaceous Kem Kem Group of Morocco, including the peirosaurid *Hamadasuchus rebouli*. Here, we describe two new specimens that demonstrate the presence of at least a fourth notosuchian species in this fauna. *Antaeosuchus taouzensis* n. gen. n. sp. is incorporated into one of the largest notosuchian-focused character-taxon matrices yet to be compiled, comprising 443 characters scored for 63 notosuchian species, with increased sampling of African and peirosaurid species. Parsimony analyses run under equal and extended implied weighting consistently recover *Antaeosuchus* as a peirosaurid notosuchian, supported by the presence of two distinct waves on the dorsal dentary surface, a surangular which laterally overlaps the dentary above the mandibular fenestra, and a relatively broad mandibular symphysis. Within Peirosauridae, *Antaeosuchus* is recovered as the sister taxon of *Hamadasuchus*. However, it differs from *Hamadasuchus* with respect to several features, including the ornamentation of the lateral surface of the mandible, the angle of divergence of the mandibular rami, the texture of tooth enamel, and the shape of the teeth, supporting their generic distinction. We present a critical reappraisal of the non-South American Gondwanan notosuchian record, which spans the Middle Jurassic–late Eocene. This indicates the existence of at least three approximately contemporaneous peirosaurid lineages within the Kem Kem Group, alongside other notosuchians, and supports the peirosaurid affinities of the ‘trematochampsid’ *Miadasuchus oblita* from the Maastrichtian of Madagascar. Furthermore, the Cretaceous record demonstrates the presence of multiple lineages of approximately contemporaneous notosuchians in several African and Madagascan faunas, and supports previous suggestions regarding an undocumented pre-Aptian radiation of Notosuchia. By contrast, the post-Cretaceous record is depauperate, comprising rare occurrences of sebecosuchians in north Africa prior to their extirpation.

Keywords: Notosuchia, Crocodylomorpha, Gondwana, Kem Kem, Mesozoic, Africa

1. Introduction

Today's crocodylians are the remnants of a once much more diverse and widespread clade, Crocodyliformes (Brochu 2003; Carvalho et al. 2010; Mannion et al. 2015; Wilberg et al. 2019; Stubbs et al. 2021). One extinct group, Notosuchia, comprises a morphologically diverse, speciose clade of terrestrial crocodyliforms (Carvalho et al. 2010; Pol et al. 2014; Pol & Leardi, 2015). Often noted to exhibit bizarre bauplans relative to other crocodyliforms, notosuchians include species characterised by features such as 'pug-nosed' and 'duck'-like snouts (e.g. Buckley et al. 2000; Sereno et al. 2003; Kley et al. 2010), elongate limbs indicative of a parasagittal posture (e.g. Gasparini 1971; Pol 2005; Riff and Kellner 2011; Godoy et al. 2016), mammal-like heterodont dentition (e.g. Carvalho 1994; Wu et al. 1995; Buckley et al. 2000; Ósi 2014), and even herbivory (e.g. O'Connor et al. 2010; Melstrom & Irmis 2019). Notosuchians have predominantly been recovered from Gondwanan landmasses, especially South America (e.g. Carvalho et al. 2010; Pol et al. 2014), from which more than 70% of species have been discovered (Pol & Leardi 2015). Although the group had its highest apparent diversity in the middle–Late Cretaceous (~120–66 Ma) (Riff et al., 2012; Pol & Leardi, 2015), notosuchians survived until the middle Miocene (~12 Ma) (Langston 1965; Langston & Gasparini 1997; Paolillo & Linares 2007), with putative remains extending their record back to the Middle Jurassic (~168 Ma) (Dal Sasso et al. 2017).

Despite severe and pervasive under-sampling of fossiliferous localities relative to most other continents (Mannion et al. 2019), diverse assemblages of extinct crocodyliforms have been discovered from several spatiotemporal intervals in Africa (e.g. Jouve 2007; Sereno & Larsson 2009; Brochu & Storrs 2012; Stefanic et al. 2020), including those yielding notosuchians. One such interval is represented by the 'middle' Cretaceous Kem Kem Group, a series of highly fossiliferous continental strata exposed in the east of Morocco along its border with Algeria, forming the northwestern edge of the Sahara Desert (Lavocat 1948; Russell 1996; Sereno et al. 1996; Bardet et al. 2010; Cavin et al. 2010; Ibrahim et al. 2020) (Fig. 1). The Kem Kem Group is generally considered to be either late Albian or Cenomanian (~105–94

1
2
3 Ma) (e.g. Martin and Lapparent de Broin 2016), with the most recent stratigraphic
reappraisal favouring this younger age (Ibrahim et al. 2020). A diverse vertebrate
fauna has been recovered from the Kem Kem Group, including sharks, bony fishes,
lissamphibians, turtles, squamates, pterosaurs, non-avian dinosaurs, and
crocodyliforms (Sereno et al. 1996; Rage & Dutheil 2008; Sereno & Larsson 2009;
Bardet et al. 2010; Cavin et al., 2010; Ibrahim et al. 2020).

The Kem Kem crocodyliforms comprise the neosuchians *Aegisuchus witmeri*
(Holliday & Gardner 2012), *Elosuchus cherifiensis* (Lavocat 1955; Lapparent de
Broin 2002), and *Laganosuchus maghrebensis* (Sereno & Larsson 2009), as well as

[revised manuscript text omitted]

2016), *Kinesuchus* (Filippi et al. 2018), and *Stolokrosuchus* (Larsson & Gado 2000),
as well as *Patagosuchus* (Lio et al. 2016) and *Montealtosuchus* (Carvalho et al.
2007), although the 'V' is slightly broader in the latter two species, forming a less
acute angle. This morphology contrasts with that of *Barrosasuchus*,
*Gasparinisuchus*, *Colhuehuapisuchus*, and *Miadanasuchus* (Rasmusson Simons &
Buckley 2009; Martinelli et al. 2012; Coria et al. 2019; Lamanna et al. 2019), in which
the dentary-splenial suture forms a broad 'U' shape. As with all other peirosaurids
(Pol et al. 2014), the dorsal surface of the mandibular symphysis on which this
suture occurs is very slightly transversely concave in *Antaeusuchus*.

In *Antaeusuchus*, as well as *Hamadasuchus* (Larsson & Sues 2007), the dentary-
splenial suture exposed on the dorsal surface of the mandibular symphysis diverges
gradually from the midline until the level of the 12th dentary tooth, at which point it
becomes parallel to the tooththrow. A similar morphology characterizes *Kinesuchus*,
except that the change in orientation of the suture is less acute in that species
(Filippi et al. 2018). In *Montealtosuchus* and *Patagosuchus*, the suture becomes
parallel with the tooththrow at the level of the 10th dentary tooth (Carvalho et al. 2007;
Lio et al. 2015). By contrast, the dentary-splenial suture in *Stolokrosuchus* parallels
the tooththrow only at the level of the 25th tooth (Larsson & Gado 2000). Although the
morphology of the suture is 'U'-shaped, as opposed to the 'V'-shape that
characterizes *Antaeusuchus*, it becomes approximately parallel with the tooththrow at
the level of the 11th and 12th tooth in *Gasparinisuchus* and *Barrosasuchus*,
respectively (Martinelli et al. 2012; Coria et al. 2019).

Posterior to the symphysis, the mandibular rami of *Antaeusuchus* diverge at an
angle of approximately 44° to each other. A value of between ~40–45° is fairly

consistent amongst peirosaurids; this contrasts with some other notosuchians,
including sphagesaurians (Pol et al. 2014; Martinelli et al. 2018), whereby the skull is
mediolaterally broader, and the rami diverge from one another at a less acute angle.
Immediately posterior to the symphysis, the splenial of *Antaeusuchus* is largely
exposed in ventral view, and forms approximately 40% of the mediolateral width of
the mandibular rami. A comparably broad splenial also characterizes
*Uberabasuchus*, *Montealtosuchus*, and *Kinesuchus* (Carvalho et al. 2004; Carvalho
et al. 2007; Filippi et al. 2018), whereas the splenial comprises only 25–30% of the
rami transverse cross section in *Gasparinisuchus*, *Stolokrosuchus*,
*Colhuehuapisuchus*, and *Barrosasuchus* (Larsson & Gado 2000; Martinelli et al.
2012; Coria et al. 2019; Lamanna et al. 2019).

In numerous peirosaurids, including *Antaeusuchus*, *Hamadasuchus*,
*Uberabasuchus*, *Montealtosuchus*, *Pepesuchus*, *Stolokrosuchus*, and *Lomasuchus*,
an anteroposteriorly elongate groove runs parallel to the dentary toothrow, just
ventral to the dorsal margin of the lateral surface of the mandible (Gasparini et al.
1991; Larsson & Gado 2000; Carvalho et al. 2004; Carvalho et al. 2007; Larsson &
Sues 2007; Campos et al. 2011). The lateral surface of the dentary is also typically
sculpted with pits and/or grooves in peirosaurids; however, there is interspecific
variation in the degree to which this sculpting continues over the entire surface. In
*Antaeusuchus*, the lateral surface of the dentary is similar in its texture and sculpting
both above and below the groove. In this regard, the morphology is similar to that of
*Uberabasuchus*, *Barrosasuchus*, and *Kinesuchus* (Carvalho et al. 2004; Filippi et al.
2018; Coria et al. 2019). In *Hamadasuchus*, *Montealtosuchus*, *Pepesuchus*, and
*Patagosuchus*, the region above the groove is smooth, differing markedly from the
remainder of the highly sculpted lateral dentary surface (Carvalho et al. 2007;
Larsson & Sues 2007; Campos et al. 2011; Lio et al. 2016). Although *Stolokrosuchus*
shows no difference in bone surface texture above and below the anteroposterior
groove (Larsson & Gado 2002), it differs from *Antaeusuchus* in that the majority of
the lateral dentary surface is smooth and unornamented.

The degree of sculpting on the lateral surface of the dentary varies across
notosuchian taxa, including within Peirosauridae. Whereas the surface is covered in
deep pits in *Hamadasuchus*, *Uberabasuchus*, *Montealtosuchus*, *Patagosuchus*,

*Bayomesasuchus*, and *Miadasuchus* (Carvalho et al. 2004; 2007; Larsson & Sues
2007; Rasmusson Simons & Buckley 2009; Barrios et al. 2016; Lio et al. 2016), the
surface of *Antaeusuchus* is considerably smoother and is textured with narrow,
shallow grooves.

Unlike *Montealtosuchus*, *Gasparinisuchus*, *Bayomesasuchus*, *Stolokrosuchus*,
*Patagosuchus*, *Pepesuchus*, and *Colhuehuapisuchus* (Buckley & Brochu 1999;
Larsson & Gado 2000; Carvalho et al. 2007; Campos et al. 2011; Martinelli et al.
2012; Lamanna et al. 2019), the lateral surface of the dentary adjacent to the 5th–8th
teeth forms a distinct anteroposteriorly elongate concavity in *Antaeusuchus*. This is
otherwise known only in *Hamadasuchus* (Larsson & Sues 2007), although a
shallower concavity also characterizes *Barrosasuchus* (Coria et al. 2019). This
depression would likely have functioned to receive an enlarged premaxillary tooth
during occlusion.

Although incompletely preserved, the mandibular fenestra in *Antaeusuchus* is almost
certainly large and anteroposteriorly elongate, as is the case in *Hamadasuchus*,
*Montealtosuchus*, and *Uberabasuchus* (Carvalho et al. 2004; Carvalho et al. 2007;
Ibrahim et al. 2020), but differing from *Barrosasuchus* in which the fenestra is greatly
reduced (Coria et al. 2019). However, *Montealtosuchus* and *Uberabasuchus* differ
from the dentaries of *Antaeusuchus* and *Hamadasuchus* (Ibrahim et al. 2020) in that
the latter two have a small posterior process that extends ventral to the mandibular
fenestra. This process is absent in *Montealtosuchus* and *Uberabasuchus*, in which
the dentary-angular contact is entirely anterior to the mandibular fenestra instead
(Carvalho et al. 2004; Carvalho et al. 2007). The dentary-surangular contact is
similar in *Antaeusuchus*, *Hamadasuchus*, *Montealtosuchus*, and *Uberabasuchus*,
with the surangular contacted by two posterior processes: the dorsal process
intrudes entirely into the surangular, whereas the second process forms the ventral
margin of the surangular and the anterodorsal margin of the mandibular fenestra
(Carvalho et al. 2004; Carvalho et al. 2007; Larsson & Sues 2007). This feature
cannot be assessed in other peirosaurid taxa, in which the relevant region of the
mandible is not preserved.

When complete, each dentary of *Antaeosuchus* has 18 tooth positions. This count is
common amongst peirosaurids, e.g. *Montealtosuchus*, *Gasparinisuchus*,
*Kinesuchus*, *Pepesuchus*, and possibly *Barrosasuchus* (Carvalho et al. 2007;
Campos et al. 2011; Martinelli et al. 2012; Filippi et al. 2018; Coria et al. 2019), but
differs from *Stolokrosuchus*, in which there are at least 30 dentary alveoli (Larsson &
Gado 2000). As in all peirosaurids (Pol et al. 2014), the first two dentary teeth of
*Antaeosuchus* are strongly procumbent.

The dentary teeth posterior to the 5th alveolus are closely spaced and are mostly
situated in a continuous groove in *Antaeosuchus*, *Hamadasuchus* (Larsson & Sues
2007), *Gasparinisuchus* (Martinelli et al. 2012), and *Barrosasuchus* (Coria et al.
2019). This differs from the condition in *Kinesuchus* (Filippi et al. 2018) and
*Patagosuchus* (Lio et al. 2016), in which the teeth are separated by distinct septa
that extend fully to the dorsal margin of the dentary.

28 29 **5.2. Detailed comparisons with *Hamadasuchus rebouli***

Although several crocodyliform taxa, including notosuchians, have been identified
from the Kem Kem Group (Ibrahim et al. 2020), only one peirosaurid species is
currently recognised from these beds: *Hamadasuchus rebouli* (Buffetaut 1994;

[revised manuscript text omitted]

mandibular features, we might expect far greater morphological variation to be
present in other regions of the skull and postcranial material. Although we erect a
new name for NHMUK PV R36829 + R3687, we refrain from naming a new taxon for

ROM 49282 pending the description and assessment of additional materials
currently assigned to *Hamadasuchus rebouli* (namely BSPG 2005 I 83 and additional
ROM specimens).

9 **6.2. Implications for peirosaurid relationships**

Peirosauridae was erected by Gasparini (1982) to accommodate *Peirosaurus*
*torminni* (Price 1955) from the late Maastrichtian Marília Formation of Brazil. This
family was subsequently expanded by Gasparini et al. (1991) to include *Lomasuchus*
from the late Turonian–early Coniacian of Argentina. Geroto & Bertini (2019 p. 328)
provided a phylogenetic definition for Peirosauridae as “the least inclusive clade
containing *P. torminni* [sic] Price, 1955, *Itasuchus jesuinoi* Price, 1955,
and *Stolokrosuchus lapparenti* Larsson & Gado, 2000, but not including *Notosuchus*
*terrestris* Woodward, 1896, *Baurusuchus pachecoi* Price, 1945, *Sphagesaurus*
*huenei* Price, 1950, *Araripesuchus gomesii* Price, 1959, *Sebecus*
*icaeorhinus* Simpson, 1937, *Mariliasuchus amarali* Carvalho & Bertini, 1999,
and *Crocodylus niloticus* Laurent, 1768”. Although a phylogenetic definition based on
two well-nested and stable species-level specifiers would be preferable (e.g. Sereno
1998, 2005; Lee 2005), we follow the definition of Geroto & Bertini (2019) here,
pending a detailed re-evaluation of the interrelationships of this part of the
notosuchian tree.

Following Geroto & Bertini’s (2019) definition, Peirosauridae comprises a
taxonomically rich array of crocodyliforms from across the Cretaceous of South
America, Africa, and Madagascar (e.g. Price 1955; Carvalho et al. 2004; Carvalho et
al. 2007; Larsson & Sues 2007; Leardi & Pol 2009; Sertich O’Connor 2014; Campos
et al. 2011; Martinelli et al. 2012; Lio et al. 2016; Barrios et al. 2016; Filippi et al.
2018; Coria et al. 2019). However, there is little consensus regarding the position of
Peirosauridae. A number of analyses have recovered Peirosauridae within
Notosuchia, as the sister taxon to Mahajangasuchidae (i.e. *Kaprosuchus* +
*Mahajangasuchus*), with these lineages forming a clade with Uruguaysuchidae that
is the sister taxon to all other notosuchians (e.g. Pol et al. 2014; Sertich & O’Connor
2014; Coria et al. 2019). Others have recovered Peirosauridae as part of Sebecia,
forming a clade with Sebecidae (e.g. Larsson & Sues 2007; Sereno & Larsson

2009), and sometimes also including Mahajangasuchidae (e.g. Geroto & Bertini
2019; Ruiz et al. 2021). Whereas some of these analyses place Sebecia as the
sister taxon to all other notosuchians (e.g. Geroto & Bertini 2019; Ruiz et al. 2021),
others recover Sebecia within Neosuchia (e.g. Larsson & Sues 2007; Sereno &
Larsson 2009). Peirosauridae has also been recovered as an early diverging
neosuchian clade in some studies (e.g. Pol & Apesteguía 2005; Gasparini 2006;
Turner & Buckley 2008; Leardi & Pol 2009).

In several recent phylogenetic analyses (e.g. Pol et al. 2014; Geroto & Bertini 2019;
Coria et al. 2019), *Hamadasuchus* has been recovered as the sister taxon to a group
of exclusively South American Cretaceous peirosaurids (comprising various
combinations of *Montealtosuchus*, *Uberabasuchus*, *Lomasuchus*, *Gasparinisuchus*,
and *Barcinosuchus*). Similarly, Barrios et al. (2016) recovered *Hamadasuchus* in a
polytomy with most of these taxa, along with *Bayomesasuchus* from the Turonian
(Late Cretaceous) of Argentina. Sertich & O'Connor (2014) recovered
*Hamadasuchus* in an unresolved trichotomy with *Rukwasuchus* and *Stolokrosuchus*,
forming a clade of African peirosaurids.

Here, under both equal and extended implied weighting schemes, the position of
Peirosauridae is consistent with the results of Pol et al. (2014) and subsequent
studies based on this dataset (e.g. Leardi et al. 2015; Fiorelli et al. 2016; Iori et al.
2018; Leardi et al. 2018; Martinelli et al. 2018; Coria et al. 2019). Under its
broadened taxonomic content, following the phylogenetic definition of Geroto &
Bertini (2019), Peirosauridae includes Mahajangasuchidae in our EIW analyses (Fig.
8). This occurs because *Stolokrosuchus* is recovered as more closely related to
Mahajangasuchidae than to other peirosaurids in the EIW topology. Our equal
weights analysis recovers *Stolokrosuchus* as the most 'basal' member of
Peirosauridae instead, with Mahajangasuchidae outside of this clade (Fig. 7). In both
cases, our peirosaurid + mahajangasuchid grouping is the sister taxon of
Uruguaysuchidae, with this clade the sister taxon to all other notosuchians.

In our strict consensus trees, the clade comprising *Antaeusuchus* and
*Hamadasuchus* is most closely related to *Bayomesasuchus*. This grouping is the
sister taxon to other peirosaurids (excluding *Stolokrosuchus* and
Mahajangasuchidae) (Fig. 7). The remaining South American taxa are grouped in a
polytomy with the African taxon *Rukwasuchus*, with this recovered as the sister
taxon of a clade comprising the Malagasy taxon *Miadasuchus* and the
Argentinean species *Barrosasuchus*. The aforementioned polytomy can be resolved
via *a posteriori* pruning of *Gasparinisuchus*, resulting in *Rukwasuchus* as the sister
taxon of a clade comprising the Malagasy taxon *Miadasuchus* and the
Argentinean species *Barrosasuchus*. The aforementioned polytomy can be resolved
via *a posteriori* pruning of *Gasparinisuchus*, resulting in *Rukwasuchus* as the sister
taxon of (*Uberabasuchus* + (*Lomasuchus* + *Montealtosuchus*)).

The fact that our analyses produce topologies more consistent with those derived
from the data matrix of Pol et al. (2014) than alternative matrices is not surprising
given that this is the underlying dataset for our study. As such, the interrelationships
of Peirosauridae within Metasuchia will require further testing, ideally merging
characters and taxa from across studies with competing hypotheses. However, the
recovery of Peirosauridae as an early diverging metasuchian clade outside of the
ziphosuchian notosuchian radiation is consistent across analyses, regardless of the
underlying dataset.

One of the notable results of our analyses is the placement of *Miadasuchus* within
Peirosauridae, which was independently recovered in this clade by Geroto and
Bertini (2019). This species from the Maastrichtian of Madagascar was originally
described as *Trematochampsia oblita* (Buffetaut & Taquet 1979), before being
assigned to a new genus by Rasmussen Simons & Buckley (2009). The type species
of *Trematochampsia*, *T. taqueti*, is based on fragmentary remains from the
Coniacian–Santonian In Beceten Formation of Niger (Buffetaut 1974; 1976a, b), for
which the family Trematochampsidae was also erected (Buffetaut 1974). Several
additional crocodyliform taxa have been assigned to Trematochampsidae (e.g.
*Amargasuchus minor* [Chiappe 1988], *Barreirosuchus franciscoi* [Lori et al. 2012],
*Hamadasuchus*, *Itasuchus*, *Mahajangasuchus*), spanning the Cretaceous of Africa,
Europe, Madagascar, and South America, with most of these known from
fragmentary remains (see review in Meunier & Larsson 2018). Buffetaut
(1988, 1989) also included *Peirosaurus torminni* as a member of
Trematochampsidae, which would therefore have priority over Peirosauridae.

However, multiple authors have questioned or rejected the monophyly of
Trematochampsidae, which appears to have become a wastebasket taxon (e.g.
Gasparini et al. 1991; Ortega et al. 1996; Buckley & Brochu 1999; Turner & Calvo
2005; Larsson & Sues 2007; Rasmusson Simons et al. 2009; Meunier & Larsson
2018). Furthermore, Meunier & Larsson (2018) demonstrated that *Trematochamps*
*taqueti* is a nomen dubium, based on non-diagnostic, chimeric remains, with some of
these displaying peirosaurid affinities. Our analyses provide further evidence that
most, if not all, Cretaceous taxa previously assigned to Trematochampsidae belong
to Peirosauridae, and confirm the presence of this latter clade in the Maastrichtian of
Madagascar. Given the lack of diagnostic features in the type remains of
'Trematochamps *taqueti*' and the absence of a formal definition for
'Trematochampsidae', coupled with its approximate synonymy with the formally
defined and widely used Peirosauridae, we support the proposal of Meunier &
Larsson (2018) to abandon the name *Trematochamps* and its coordinated rank
taxa.

**6.3. Gondwanan notosuchian diversity outside of South America**

During the Mesozoic, notosuchians (*sensu* Pol et al. 2014) were the most diverse
clade of Gondwanan crocodyliforms (Turner & Sertich 2010), although this high
species richness varied through both time and space (Pol & Leardi 2015; De Celis et
al. 2020). At least 70% of known notosuchian diversity is found on Gondwanan
continents (Pol & Leardi 2015), with a small number of species recognised from
Europe (Kuhn 1968; Antunes 1975; Rossman et al. 2000; Company et al. 2005;
Dalla Vecchia & Cau 2011; Rabi & Sebök 2015; Martin 2016; Sellés et al. 2020) and
Asia (Wu et al. 1995; Wu & Sues 1996). Though most numerous in South America,
Gondwanan notosuchian occurrences are also known from mainland Africa,
Madagascar, India, and Pakistan, as well as possibly the Arabian Peninsula.
Currently no notosuchians are known from Australasia or Antarctica, although it
remains unclear whether this represents a genuine absence, perhaps pertaining to a
high-latitude environmental dispersal barrier, or it reflects a sampling bias (e.g. see
Poropat et al. 2021). Here, we provide a critical reappraisal of the Gondwanan
record of notosuchians outside of South America.

6.3.1. Jurassic

The stratigraphically oldest known notosuchian is *Razanandrongobe sakalavae* (Maganuco et al. 2006) from the Bathonian (Middle Jurassic) Isalo IIIb Formation in northwestern Madagascar. Originally named as an archosaur of uncertain affinities on the basis of teeth and a fragmentary maxilla (Maganuco et al. 2006), several more skull fragments, including a right premaxilla and an incomplete left dentary, have since been assigned to the taxon, enabling its identification as a large-bodied notosuchian (Dal Sasso et al. 2017). Considering that the next stratigraphically oldest notosuchians are from the Aptian (late Early Cretaceous), resulting in a ~40 million-year ghost lineage, *Razanandrongobe* is a stratigraphic outlier and its affinities might seem doubtful. However, based on the sister taxon relationship of Notosuchia and Neosuchia, with the latter clade known from the Early Jurassic (Tykoski et al., 2002), *Razanandrongobe* instead partly fills the inferred ghost lineage of notosuchians, which otherwise would extend back approximately 65–75 million years (Dal Sasso et al. 2017; Mannion et al. 2019). In the small number of phylogenetic analyses to have incorporated it (Dal Sasso et al. 2017; Sellés et al. 2020), including ours, *Razanandrongobe* is recovered in a position close to the ‘base’ of Sebecosuchia. This nested position within Notosuchia for such a stratigraphically early species necessitates the extension of multiple unsampled lineages back into the Jurassic (Fig. 11). As such, the phylogenetic affinities of *Razanandrongobe* require further evaluation to test whether this poor stratigraphic fit is genuine.

6.3.2. Early Cretaceous

In southeastern Africa, the Aptian Dinosaur Beds of northern Malawi (Fig. 12) have yielded numerous remains of *Malawisuchus mwakasyungutiensis*, preserving most of the skeleton (Gomani 1997). Recognised in part for its unusual mammal-like multicuspid teeth, some analyses have placed *Malawisuchus* in a nested position within Sphagesauria (e.g. Gomani 1997; Sereno & Larsson 2009; O’Connor et al. 2010). However, most recent analyses typically recover *Malawisuchus* as an early diverging ziphosuchian, with sphagesaurians currently restricted to South America (e.g. Pol et al. 2014; Ruiz et al. 2021; this study). Unlike the topology of Martin and

Lapparent de Broin (2016), *Malawisuchus* is not recovered within Candidodontidae
in our analyses (Fig. 7).

The Aptian–Albian Elrhaz Formation exposed at Gadoufaoua, central Niger (Fig. 12),
has yielded the remains of three morphologically diverse notosuchian species
(*Anatosuchus minor*, *Araripesuchus wegneri*, and *Stolokrosuchus lapparenti*). The
bizarre, ‘duck-billed’ *Anatosuchus* is known from several individuals, including a skull
and associated partial postcranial skeleton, as well as a skull of a juvenile animal
(Sereno et al. 2003; Sereno & Larsson 2009). *Anatosuchus* has often been
recovered as a member of Uruguaysuchidae (e.g. Sereno & Larsson 2009; Pol et al.
2014); some analyses have placed it outside of this clade, although these tend to
recover it as a ‘basal’ member of Notosuchia. The small and gracile species
*Araripesuchus wegneri* was erected from the anterior region of an articulated upper
and lower snout (Buffetaut & Taquet 1979). Multiple remains have since been
assigned to the taxon, including a block preserving at least five separate individuals,
three of which are essentially complete, partially articulated skeletons (Sereno &
Larsson 2009). In our analyses, *Araripesuchus wegneri* and *Anatosuchus* are
recovered as sister taxa within Uruguaysuchidae, further questioning the monophyly
of *Araripesuchus* (see Sereno & Larsson 2009: p. 31). The longirostrine-snouted
*Stolokrosuchus* is known from an almost complete skull (Larsson & Gado 2000).
Originally referred to Peirosauridae (Larsson & Gado 2000; see also Larsson & Sues
2007; Sereno & Larsson 2009; Geroto & Bertini 2019), subsequent analyses have
shown the position of *Stolokrosuchus* to be highly labile, such that it has also been
placed as an early diverging member of both Notosuchia (e.g. Dal Sasso et al. 2017)
and Neosuchia (e.g. Turner & Sertich 2010; Andrade et al. 2011). Following the
definition of Peirosauridae provided by Geroto & Bertini (2019), our analyses recover
*Stolokrosuchus* as the earliest diverging member of this clade, which is consistent
with previous analyses that have continued to place it close to the ‘base’ of
Metasuchia.

Several isolated teeth from the Aptian–Albian Koum Formation of northeastern
Cameroon (Fig. 12) were reported by Flynn et al. (1988) and Congleton (1990), who
recognised their possible affinities with *Araripesuchus*, especially *A. wegneri*.
Kellner (1994 p. 618) questioned this referral, suggesting that these strongly

serrated, laterally compressed, leaf-shaped teeth differed from those in the posterior
toothrow of *Araripesuchus gomesii*, which he described as “weakly serrated” and
“less leaf-shaped”. It is unclear why Kellner (1994) limited comparisons to
*Araripesuchus gomesii*; nonetheless, the description of additional specimens of
*Araripesuchus wegneri* from Niger (Sereno & Larsson 2009), along with other
species of this genus (e.g. Pol & Apesteguia 2005; Ortega et al. 2000; Turner 2006;
Sereno & Larsson 2009; Dumont et al. 2020; Ibrahim et al. 2020), allows for more
thorough comparisons with the teeth from Cameroon. Given that none of the South
American *Araripesuchus* species, nor *Araripesuchus tsangatsangana*, have
denticles, the labiolingually compressed, lanceolate shape of these teeth, with
serrated carinae along their posteriormost and anteriormost margins, is supportive of
a referral to either *Araripesuchus wegneri* or *Araripesuchus rattoides* (the latter
comparison is based on referred material, BSPG 2008 I 41, rather than the holotype
specimen [Ibrahim et al. 2020]). However, because of variation in crown morphology
along the toothrow in all species of *Araripesuchus*, and given that teeth in the
middle-to-posterior toothrow are either absent or poorly preserved in *Araripesuchus*
*rattoides*, it is not currently possible to provide a species-level referral.

The Albian Aïn el Guettar Formation in southern Tunisia (Fig. 12) has yielded
numerous crocodyliform remains, including teeth assigned to *Araripesuchus*
*wegneri*, *Araripesuchus* sp., and aff. *Hamadasuchus* sp. (Le Loeuff et al. 2000;
Cuny et al. 2010; Fanti et al. 2012). The specimens assigned to *Araripesuchus*
(Cuny et al. 2010, fig. 4.7; Fanti et al. 2012, fig. 12U–X) are labiolingually
compressed and triangular, with serrated carinae and relatively smooth enamel.
Based on the slightly dorsoventrally constricted lanceolate shape of the teeth in
lateral view, it is likely that they come from the middle region of the toothrow. All of
these features support their referral to *Araripesuchus*, widening the spatial
distribution of the genus to north-central Africa. Although serrated tooth margins are
known to be present in *Araripesuchus wegneri* and a referred specimen of
*Araripesuchus rattoides* (Ibrahim et al. 2020), we refrain from assigning these
specimens beyond generic level as was “cautiously” proposed by Cuny et al. (2010:
p. 625) for the same reasons outlined in the preceding paragraph. A single tooth
referred to aff. *Hamadasuchus* sp. is labiolingually compressed and approximately
triangular in lateral view, with “remnants of clear serration” (Cuny et al. 2010: fig. 4.8,

p. 625). Although the more extreme labiolingual compression towards the anterior
and posterior margins of the tooth is reminiscent of *Hamadasuchus*, the apparent
lack of rugose enamel is unusual given its presence in all teeth associated with the
holotype specimen of *Hamadasuchus*. The only other named crocodyliforms from
the Early Cretaceous of Africa to possess serrated carinae are *Araripesuchus*
*wegeneri* and referred material of *Araripesuchus rattoides*, both of which possess
dentition more similar in size to the tooth reported in Cuny et al. (2010). However,
given that the Tunisian specimen is clearly well-worn and is not dissimilar in broad
morphology from either *Hamadasuchus* or *Araripesuchus*, we regard this specimen
as an indeterminate notosuchian. Re-evaluation and full description of specimens
referred to *Hamadasuchus* that have teeth with smooth enamel (e.g. BSPG 2005 I
83) might enable referral to a particular genus.

**6.3.3. Late Cretaceous**

In northwestern Africa, the Cenomanian Kem Kem Group of Morocco has yielded
three previously named notosuchian species (*Araripesuchus rattoides*,
*Hamadasuchus rebouli*, *Lavocatchampsa sigogneaurussellae*), in addition to the new
species, *Antaeusuchus taouzensis*, described herein (Fig. 12). *Araripesuchus*
*rattoides* is known from the holotypic partial dentary, as well as several referred
dentary fragments (Sereno & Larsson 2009; Ibrahim et al 2020). It differs from *A.*
*wegeneri* in several features, including its possession of a narrower and deeper
snout, highly procumbent teeth, and potentially a greater number of teeth. Though
not included in our phylogenetic analyses because of its fragmentary nature,
*Araripesuchus rattoides* was recovered by Sereno & Larsson (2009) in a polytomy
with three *Araripesuchus* species (*A. gomesii* and *A. patagonicus* from South
America, and *A. tsangatsangana* from Madagascar), with this the sister group to a
clade comprising the remaining uruguaysuchids (including *A. wegeneri*).
*Lavocatchampsa sigogneaurussellae* was erected based on a small anterior snout
region, which displays unusually heterodont teeth that are convergent with those of
mammals (Martin & Lapparent de Broin 2016). Using the data matrix of Pol et al.
(2014), Martin & Lapparent de Broin (2016) recovered *Lavocatchampsa* as a 'basal'
ziphosuchian within Candidodontidae, a small clade otherwise known only from the
Cretaceous of South America (Carvalho et al. 2004, Montefeltro et al. 2009).

As discussed in detail above, the peirosaurid *Hamadasuchus rebouli* was erected
based on a single dentary fragment from the Kem Kem Group (Buffetaut 1994), but
numerous cranial and mandibular remains have since been referred to this species
from this stratigraphic unit (Larsson & Sues 2007; Ibrahim et al., 2020), including a
skull table previously assigned to *Libycosuchus* sp. (Buffetaut 1976a, b). Although
we do not disagree with referral of these remains to Peirosauridae, it is currently
unclear if all of them are attributable to *Hamadasuchus rebouli*. Isolated teeth
described by Larsson & Sidor (1999) were also referred to this species. One tooth,
inferred to be from the middle of the toothrow (Larsson & Sidor 1999: fig. 1B), is very
reminiscent of those preserved in the holotype of *Hamadasuchus rebouli*, based on
its triangular shape in lateral view, its labiolingual compression, and the density of
serrations. A second tooth shares the globular texture of the enamel towards the
base of the crown, which transitions into more linear ridges towards the apex
(Larsson & Sidor 1999: fig. 1C), which is again consistent with a referral to
*Hamadasuchus rebouli*. However, a conical, retro-curved caniniform tooth shows
distinctive fluting (Larsson & Sidor 1999: fig. 1A), which is absent from the holotypic
specimen, but present in some of the specimens previously referred to the species
(ROM 49282 and 52620, BSPG 2005 I 83, and possibly NMC 41892 [Ibrahim et al.
2020]).

Larsson & Sidor (1999) described several additional crocodyliform teeth from the
Kem Kem Group that have been suggested to represent additional notosuchian taxa
(Ibrahim et al., 2020). Material referred to “Indet. crocodyliform 1” (Larsson & Sidor

[revised manuscript text omitted]

Competing interests. We declare we have no competing interests.

Funding. C.S.C.N. is funded by a Royal Society research grant (RGF\R1\180020)
awarded to P.D.M. E.S.E.H. received funding from a Palaeontological Association
Undergraduate Research Bursary (PA-UB201804) and her work is supported by a
Natural Environment Research Council studentship (NE/S007415/1). P.D.M.'s
contribution was supported by grants from the Royal Society (UF160216,
RGF\R1\180020, and RGF\EA\201037).

References

- Antunes MT. 1975. *Iberosuchus*, crocodile sebecosuchien nouveau, l'Eocene ibérique au Nord de la chaîne centrale, et l'origine du canyon de Nazaré. *Comun. Serv. Geol. Portug Comunicações dos Serviços Geológicos de Portugal* **59**, 285–330.
- Bardet N, Suberbiola XP, Jouve S, Bourdon E, Vincent P, Houssaye A, Rage JC, Jalil NE, Bouya B, Amaghazaz M. 2010. Reptilian assemblages from the latest Cretaceous–Palaeogene phosphates of Morocco: from Arambourg to present time. *Historical Biology* **22(1-3)**, 186–199. doi: 10.1080/08912961003754945
- Barrios F, Paulina-Carabajal A, Bona P. 2016. A new peirosaurid (Crocodyliformes, Mesoeucrocodylia) from the Upper Cretaceous of Patagonia, Argentina. *Ameghiniana* **53(1)**, 14–25. doi: 10.5710/AMGH.03.09.2015.2903
- Brochu CA. 2003. Phylogenetic approaches toward crocodylian history. *Annual Review of Earth and Planetary Sciences* **31(1)**, 357–397. doi: 10.1146/annurev.earth.31.100901.141308
- Brochu CA, Storrs GW. 2012. A giant crocodile from the Plio-Pleistocene of Kenya, the phylogenetic relationships of Neogene African crocodylines, and the antiquity of *Crocodylus* in Africa. *Journal of Vertebrate Paleontology* **32(3)**, 587–602. doi: 10.1080/02724634.2012.652324
- Bronzati M, Montefeltro FC, Langer MC. 2012. A species-level supertree of Crocodyliformes. *Historical Biology* **24(6)**, 598–606. doi: 10.1080/08912963.2012.662680
- Buckley GA, Brochu CA. 1999. An enigmatic new crocodile from the Upper Cretaceous of Madagascar. *Special papers in Palaeontology* **60**, 149–175.
- Buckley GA, Brochu CA, Krause DW, Pol D. 2000. A pug-nosed crocodyliform from the Late Cretaceous of Madagascar. *Nature* **405(6789)**, 941–944. doi: 10.1038/35016061
- Buffetaut E. 1974. *Trematochampsia taqueti*, un crocodilien nouveau du Sénonien inférieur du Niger. *Comptes Rendus de l'Académie des Sciences (Paris)* **279(D)**, 1749–1752.
- Buffetaut E. 1976a. Ostéologie et affinités de *Trematochampsia taqueti* (Crocodylia, Mesosuchia) du Sénonien inférieur d'In Beceten (République du Niger). *Géobios* **9**, 143–198.
- Buffetaut E. 1976b. Der Land-Krokodilier *Libycosuchus* Stromer und die Familie Libycosuchidae (Crocodylia, Mesosuchia) aus der Kreide Afrikas. *Mitteilungen der Bayerischen Staatssammlung für Paläontologie und Historische Geologie* **16**, 17–28.
- Buffetaut E, Taquet P. 1979. An early Cretaceous terrestrial crocodylian and the opening of the South Atlantic. *Nature* **280(5722)**, 486–487.

Buffetaut E. 1988. The ziphodont mesosuchian crocodile from Messel: a
reassessment. *Courier Forschungsinstitut Senckenberg* **107**, 211–221.
Buffetaut E. 1989. A new ziphodont mesosuchian crocodile from the Eocene of
Algeria. *Palaeontographica. Abteilung A, Paläozoologie, Stratigraphie* **208(1-3)**, 1–
10.
Buffetaut E, Bussert R, Brinkmann W. 1990. A new nonmarine vertebrate fauna in
the Upper Cretaceous of northern Sudan. *Berliner Geowissenschaftliche*
*Abhandlungen* **120**, 183–202.
Buffetaut E. 1994. A new crocodylian from the Cretaceous of southern Morocco.
*Comptes Rendus de l'Académie Des Sciences* **319**, 1563–1568.
Buscalioni AD, Schulp AS, Jagt JW, Hanna SS, Hartman AF. 2004. Late Cretaceous
neosuchian crocodiles from the Sultanate of Oman. *Cretaceous Research* **25(2)**,
267–275. doi: 10.1016/j.cretres.2003.12.004
Campos DA, Oliveira GR, Figueiredo RG, Riff D, Azevedo SA, Carvalho LB, Kellner
AW. 2011. On a new peirosaurid crocodyliform from the Upper Cretaceous, Bauru
Group, southeastern Brazil. *Anais da Academia Brasileira de Ciências* **83**, 317–327.
doi: 10.1590/S0001-37652011000100020
Carvalho IS, 1994. *Candidodon*: um crocodile con heterodontia (Notosuchia,
Cretaceo Inferior—Brasil). *Anais do Academia brasileira de Ciencias* **66**, 331–346.
Carvalho IS, Ribeiro LCB, Avilla LS. 2004. *Uberabasuchus terrificus sp. nov.* a new
Crocodylomorpha from the Bauru Basin (Upper Cretaceous), Brazil. *Gondwana*
*Research* **7(4)**, 975–1002. doi: 10.1016/S1342-937X(05)71079-0
Carvalho IS, Campos ACA, Nobre PH. 2005. *Baurusuchus salgadoensis*, a new
Crocodylomorpha from the Bauru Basin (Cretaceous), Brazil. *Gondwana Research*
**8**, 11–30. doi: 10.1016/S1342-937X(05)70259-8
Carvalho IS, Vasconcellos FM, Tavares SAS. 2007. *Montealtosuchus*
*arrudacamposi*, a new peirosaurid crocodile (Mesoeucrocodylia) from the Late
Cretaceous Adamantina Formation of Brazil. *Zootaxa*, **1607**, 35–46.
Carvalho IS, de Gasparini ZB, Salgado L, de Vasconcellos FM, da Silva Marinho T.
2010. Climate's role in the distribution of the Cretaceous terrestrial Crocodyliformes
throughout Gondwana. *Palaeogeography, Palaeoclimatology, Palaeoecology* **297(2)**,
252–262. doi: 10.1016/j.palaeo.2010.08.003
Cavin L, Tong H, Boudad L, Meister C, Piuz A, Tabouelle J, Aarab M, Amiot R,
Buffetaut E, Dyke G, Hua S. 2010. Vertebrate assemblages from the early Late
Cretaceous of southeastern Morocco: an overview. *Journal of African Earth*
*Sciences* **57(5)**, 391–412. doi: 10.1016/j.jafrearsci.2009.12.007

Chiappe LM. 1988. A new trematochampsid crocodile from the Early Cretaceous of
north-western Patagonia, Argentina and its palaeobiogeographical and phylogenetic
implications. *Cretaceous Research* **9(4)**, 379–389. doi: 10.1016/0195-
6671(88)90009-2

Company J, Suberbiola XP, Ruiz-Omeñaca JI, Buscalioni AD. 2005. A new species
of *Doratodon* (Crocodyliformes: Ziphosuchia) from the Late Cretaceous of Spain.
*Journal of Vertebrate Paleontology* **25(2)**, 343–53. doi: 10.1671/0272-
4634(2005)025[0343:ANSODC]2.0.CO;2

Congleton JD. 2010. *Vertebrate paleontology of the Koum Basin, northern*
*Cameroon, and archosaurian paleobiogeography in the Early Cretaceous* (Doctoral
dissertation, Southern Methodist University).

Coria RA, Ortega F, Arcucci AB, Currie PJ. 2019. A new and complete peirosaurid
(Crocodyliformes, Notosuchia) from Sierra Barrosa (Santonian, Upper Cretaceous)
of the Neuquén Basin, Argentina. *Cretaceous Research* **95**, 89–105. doi:
10.1016/j.cretres.2018.11.008

Coster P, Benammi M, Mahboubi M, Tabuce R, Adaci M, Marivaux L, Bensalah M,
Mahboubi S, Mahboubi A, Mebrouk F, Maameri C. 2010. Chronology of the Eocene
continental deposits of Africa: Magnetostratigraphy and biostratigraphy of the El
Kohol and Glib Zegdou Formations, Algeria. *Bulletin* **124(9-10)**, 1590–1606. doi:
10.1130/B30565.1

Cuny G, Cobbett AM, Meunier FJ, Benton MJ. 2010. Vertebrate microremains from
the Early Cretaceous of southern Tunisia. *Geobios* **43(6)**, 615–28. doi:
10.1016/j.geobios.2010.07.001

Dal Sasso C, Pasini G, Fleury G, Maganuco S. 2017. *Razanandrongobe sakalavae*,
a gigantic mesoeucrocodylian from the Middle Jurassic of Madagascar, is the oldest
known notosuchian. *PeerJ* **5**, e3481. doi: ~10.7717/peerj.3481

Dalla Vecchia FM, Cau A. 2011. The first record of a notosuchian crocodyliform from
Italy. *Rivista Italiana di Paleontologia e Stratigrafia* **117(2)**, 309–321.

Darlim G, Montefeltro FC, Langer MC. 2021. 3D skull modelling and description of a
new baurusuchid (Crocodyliformes, Mesoeucrocodylia) from the Late Cretaceous
(Bauru Basin) of Brazil. *Journal of Anatomy*. **00**, 1–41. doi: 10.1111/joa.13442

de Andrade MB, Edmonds R, Benton MJ, Schouten R. 2011. A new Berriasian
species of *Goniopholis* (Mesoeucrocodylia, Neosuchia) from England, and a review
of the genus. *Zoological Journal of the Linnean Society* **163(sup1)**, S66-108. doi:
10.1111/j.1096-3642.2011.00709.x

de Broin FD. 2002. *Elosuchus*, a new genus of crocodile from the Cretaceous of the
North of Africa. *Comptes Rendus Palevol* **1(5)**, 275–285. doi: 10.1016/S1631-
0683(02)00049-0

de Celis A, Narváez I, Arcucci A, Ortega F. 2020. Lagerstätte effect drives
notosuchian palaeodiversity (Crocodyliformes, Notosuchia). *Historical Biology* **15**, 1-
10. doi: 10.1080/08912963.2020.1844682

Dumont MF, Bona P, Pol D, Apesteguía S. 2020. New anatomical information on
*Araripesuchus buitreaensis* with implications for the systematics of
Uruguaysuchidae (Crocodyliformes, Notosuchia). *Cretaceous Research* **113**, 104494.
doi: 10.1016/j.cretres.2020.104494

Fanti F, Contessi M, Franchi F. 2012. The “Continental Intercalaire” of southern
Tunisia: stratigraphy, paleontology, and paleoecology. *Journal of African Earth*
*Sciences* **73**, 1-23. doi: 10.1016/j.jafrearsci.2012.07.006

Filippi LS, Barrios F, Garrido AC. 2018. A new peirosaurid from the Bajo de la Carpa
Formation (Upper Cretaceous, Santonian) of Cerro Overo, Neuquén, Argentina.
*Cretaceous Research* **83**, 75–83. doi: 10.1016/j.cretres.2017.10.021

Fiorelli LE, Leardi JM, Hechenleitner EM, Pol D, Basilici G, Grellet-Tinner G. 2016. A
new Late Cretaceous crocodyliform from the western margin of Gondwana (La Rioja
Province, Argentina). *Cretaceous Research* **60**, 194–209. doi:
10.1016/j.cretres.2015.12.003

Fletcher Jr RJ, Didham RK, Banks-Leite C, Barlow J, Ewers RM, Rosindell J, Holt
RD, Gonzalez A, Pardini R, Damschen EI, Melo FP. 2018. Is habitat fragmentation
good for biodiversity?. *Biological conservation* **226**, 9–15. doi:
10.1016/j.biocon.2018.07.022

Flynn LJ, Brillanceau A, Brunet M, Coppens Y, Dejax J, Dupéron-Laudoueneix M,
Ekodeck G, Flanagan KM, Heintz E, Hell J, Jacobs LL. 1988. Vertebrate fossils from
Cameroon, West Africa. *Journal of Vertebrate Paleontology* **7(4)**, 469–71. doi:
10.1080/02724634.1988.10011676

Gasparini ZB. 1971. Los Notosuchia del Cretácico de América del Sur como un
nuevo infraorden de los Mesosuchia (Crocodylia). *Ameghiniana* **8(2)**, 83–103.

Gasparini ZB. 1982. Una nueva familia de cocodrilos zifodontes cretácicos de
América del Sur. *Actas V Congreso Latinoamericano de Geología, Buenos Aires* **4**,
317–329.

Gasparini Z, Chiappe LM, Fernandez M. 1991. A new Senonian peirosaurid
(Crocodylomorpha) from Argentina and a synopsis of the South American
Cretaceous crocodylians. *Journal of Vertebrate Paleontology* **11(3)**, 316–333. doi:
10.1080/02724634.1991.10011401

Georgi JA, Krause DW. 2010. Postcranial axial skeleton of *Simosuchus clarki*
(Crocodyliformes: Notosuchia) from the Late Cretaceous of Madagascar. *Journal of*
*Vertebrate Paleontology* **30(sup1)**, 99–121. doi: 10.1080/02724634.2010.519172

Geroto CF, Bertini RJ. 2018. New material of *Pepesuchus* (Crocodyliformes;
Mesoeucrocodylia) from the Bauru Group: implications about its phylogeny and the
age of the Adamantina Formation. *Zoological Journal of the Linnean Society* **185(2)**,
312–334. doi: 10.1093/zoolinnean/zly037

Godoy PL, Bronzati M, Eltink E, Júlio CDA, Cidade GM, Langer MC, Montefeltro FC.
2016. Postcranial anatomy of *Pissarrachampsia sera* (Crocodyliformes,
Baurusuchidae) from the Late Cretaceous of Brazil: insights on lifestyle and
phylogenetic significance. *PeerJ* **4**, e2075. doi: 10.7717/peerj.2075

Goloboff PA, Farris JS, Nixon KC. 2008. TNT, a free program for phylogenetic
analysis. *Cladistics* **24(5)**, 774–786. doi: 10.1111/j.1096-0031.2008.00217.x

Goloboff PA, Torres A, Arias JS. 2018. Weighted parsimony outperforms other
methods of phylogenetic inference under models appropriate for morphology.
*Cladistics* **34(4)**, 407–437. doi: 10.1111/cla.12205

Gomani EM. 1997. A crocodyliform from the Early Cretaceous dinosaur beds,
northern Malawi. *Journal of Vertebrate Paleontology* **17(2)**, 280–294. doi:
10.1080/02724634.1997.10010975

Groh SS, Upchurch P, Barrett PM, Day JJ. 2020. The phylogenetic relationships of
neosuchian crocodiles and their implications for the convergent evolution of the
longirostrine condition. *Zoological Journal of the Linnean Society* **188(2)**, 473–506.
doi: 10.1093/zoolinnean/zlz117

Hill RV. 2010. Osteoderms of *Simosuchus clarki* (Crocodyliformes: Notosuchia) from
the late cretaceous of Madagascar. *Journal of Vertebrate Paleontology* **30(sup1)**,
154–176. doi: 10.1080/02724634.2010.518110

Holliday CM, Gardner NM. 2012. A new eusuchian crocodyliform with novel cranial
integument and its significance for the origin and evolution of Crocodylia. *PloS one*
**7(1)**, e30471. doi: 10.1371/journal.pone.0030471

Ibrahim N, Sereno PC, Varricchio DJ, Martill DM, Dutheil DB, Unwin DM, Baidder L,
Larsson HC, Zouhri S, Kaoukaya A. 2020. Geology and paleontology of the upper
cretaceous Kem Kem group of eastern Morocco. *ZooKeys* **928**, 1–216. doi:
10.3897/zookeys.928.47517

Iori FV, Garcia KL. 2012. *Barreirosuchus franciscoi*, um novo Crocodylomorpha
Trematochampsidae da Bacia Bauru, Brasil. *Brazilian Journal of Geology* **42(2)**,
397–410. doi: 10.5327/Z0375-75362012000200013

Iori FV, da Silva Marinho T, de Souza Carvalho I, dos Santos Frare LA. 2018.
Cranial morphology of *Morrinhosuchus luziae* (Crocodyliformes, Notosuchia) from
the Upper Cretaceous of the Bauru Basin, Brazil. *Cretaceous Research* **86**, 41–52.
doi: 10.1016/j.cretres.2018.02.010

Jouve S. 2007. Taxonomic revision of the dyrosaurid assemblage (Crocodyliformes:
Mesoeucrocodylia) from the Paleocene of the Iullemeden Basin, West Africa.

*Journal of Paleontology* **81(1)**, 163–175. doi: 10.1666/0022-
3360(2007)81[163:TROTDA]2.0.CO;2

Kellner AW. 1994. Comments on the paleobiogeography of Cretaceous archosaurs
during the opening of the South Atlantic Ocean. *Acta Geologica Leopoldensia*
**17(39/2)**, 615–625.

Kley NJ, Sertich JJ, Turner AH, Krause DW, O'Connor PM, Georgi JA. 2010.
Craniofacial morphology of *Simosuchus clarki* (Crocodyliformes: Notosuchia) from
the late Cretaceous of Madagascar. *Journal of Vertebrate Paleontology* **30(sup1)**,
13–98. doi: 10.1080/02724634.2010.532674

Kuhn O. 1968. Die Vorzeitlichen Krokodile. *Krailing: Oeben* 124 p.

Lamanna MC, Casal GA, Ibiricu LM, Martínez RD. 2019. A New Peirosaurid
Crocodyliform from the Upper Cretaceous Lago Colhué Huapi Formation of Central
Patagonia, Argentina. *Annals of Carnegie Museum* **85(3)**, 193–211. Doi:
10.2992/007.085.0301

Langston W. 1965. Fossils crocodylians from Colombia and the Cenozoic history of
the Crocodylia. *University of California Publications in Geological Sciences* **52**, 1–
157.

Langston W, Gasparini Z. 1997. Crocodylians, *Gryposuchus*, and the South
American gavials. In *Vertebrate Paleontology in the Neotropics: The Miocene Fauna*
*of La Venta, Colombia*, ed. RF Kay, RH Madden, RL Cifelli, JJ Flynn, pp. 113–
54. Washington, DC: Smithsonian. Inst.

Larsson HC, Sidor CA. 1999. Unusual crocodyliform teeth from the Late Cretaceous
(Cenomanian) of southeastern Morocco. *Journal of Vertebrate Paleontology* **19(2)**,
398–401. doi: 10.1080/02724634.1999.10011152

Larsson HC, Gado B. 2000. A new Early Cretaceous crocodyliform from Niger.
*Neues Jahrbuch für Geologie und Paläontologie-Abhandlungen* **217**, 131–141. doi:
10.1127/njgpa/217/2000/131

Larsson HC, Sues HD. 2007. Cranial osteology and phylogenetic relationships of
*Hamadasuchus rebouli* (Crocodyliformes: Mesoeucrocodylia) from the Cretaceous of
Morocco. *Zoological Journal of the Linnean Society* **149(4)**, 533–567. doi:
10.1111/j.1096-3642.2007.00271.x

Lavocat R. 1948. Découverte de Crétacé à vertébrés dans le soubassement de
l'Hammada du Guir (Sud Marocain). *Comptes rendus de l'Académie des Sciences*
*de Paris* **226**, 1291–1292.

Lavocat R. 1955. Découverte d'un Crocodylien du genre *Thoracosaurus* dans le
Crétacé supérieur d'Afrique. *Bulletin du Muséum national d'Histoire naturelle* **27**,
338–340.

Leardi JM, Pol D. 2009. The first crocodyliform from the Chubut Group (Chubut
Province, Argentina) and its phylogenetic position within basal Mesoeucrocodylia.
*Cretaceous Research* **30(6)**, 1376–1386. doi: 10.1016/j.cretres.2009.08.002

Leardi JM, Pol D, Novas FE, Suárez Riglos M. 2015. The postcranial anatomy of
*Yacarerani boliviensis* and the phylogenetic significance of the notosuchian
postcranial skeleton. *Journal of Vertebrate Paleontology* **35(6)**, e995187. doi:
10.1080/02724634.2014.995187

Leardi JM, Pol D, Gasparini Z. 2018. New Patagonian baurusuchids
(Crocodylomorpha; Notosuchia) from the Bajo de la Carpia Formation (Upper
Cretaceous; Neuquén, Argentina): new evidences of the early sebecosuchian
diversification in Gondwana. *Comptes Rendus Palevol* **17(8)**, 504–21. doi:
10.1016/j.crpv.2018.02.002

Lee MSY. 2005. Choosing reference taxa in phylogenetic nomenclature. *Zoologica*
*Scripta* **34**: 329–331. doi: 10.1111/j.1463-6409.2005.00196.x

Lio G, Agnolín FL, Valieri RJ, Filippi L, Rosales D. 2016. A new peirosaurid
(Crocodyliformes) from the Late Cretaceous (Turonian–Coniacian) of Patagonia,
Argentina. *Historical Biology* **28(6)**, 835–841. doi: 10.1080/08912963.2015.1043999

Maganuco S, Dal Sasso C, Pasini G. 2006. A new large predatory archosaur from
the Middle Jurassic (Bathonian) of Madagascar. *Atti Soc. Ital. Sci. Nat. Mus. Civ. St.*
*Nat. Milano* **147(1)**, 19–51.

Mannion PD, Benson RB, Carrano MT, Tennant JP, Judd J, Butler RJ. 2015. Climate
constrains the evolutionary history and biodiversity of crocodylians. *Nature*
*Communications* **6(1)**, 1–9. doi: 10.1038/ncomms9438

Mannion PD, Chiarenza AA, Godoy PL, Cheah YN. 2019. Spatiotemporal sampling
patterns in the 230 million year fossil record of terrestrial crocodylomorphs and their
impact on diversity. *Palaeontology* **62(4)**, 615–637. doi: **10.1111/pala.12419**

Martin JE, De Lapparent De Broin F. 2016. A miniature notosuchian with multicuspid
teeth from the Cretaceous of Morocco. *Journal of Vertebrate Paleontology* **36(6)**,
e1211534. doi: 10.1080/02724634.2016.1211534

Martinelli AG, Sertich JJ, Garrido AC, Praderio ÁM. 2012. A new peirosaurid from
the Upper Cretaceous of Argentina: Implications for specimens referred to
*Peirosaurus torminni* Price (Crocodyliformes: Peirosauridae). *Cretaceous Research*
**37**, 191–200. doi: 10.1016/j.cretres.2012.03.017

Martinelli AG, Marinho TS, Iori FV, Ribeiro LC. 2018. The first Caipirasuchus
(Mesoeucrocodylia, Notosuchia) from the Late Cretaceous of Minas Gerais, Brazil:
new insights on sphagesaurid anatomy and taxonomy. *PeerJ* **6**, e5594. doi:
10.7717/peerj.5594

Martínez RN, Alcober OA, Pol D. 2018. A new protosuchid crocodyliform
(Pseudosuchia, Crocodylomorpha) from the Norian Los Colorados Formation,

northwestern Argentina. *Journal of Vertebrate Paleontology* **38(4)**, 1–12. doi:
10.1080/02724634.2018.1491047

Melstrom KM, Irmis RB. 2019. Repeated evolution of herbivorous crocodyliforms
during the age of dinosaurs. *Current Biology* **29(14)**, 2389–2395. doi:
10.1016/j.cub.2019.05.076

Meunier LM, Larsson HC. 2018. *Trematochampsia taqueti* as a nomen dubium and
the crocodyliform diversity of the Upper Cretaceous In Beceten Formation of Niger.
*Zoological Journal of the Linnean Society* **182(3)**, 659–680. doi:
10.1093/zoolinnean/zlx061

Montefeltro FC, Laurini CR, Langer MC. 2009. Multicusped crocodyliform teeth from
the Upper Cretaceous (São José do Rio Preto Formation, Bauru Group) of São
Paulo, Brazil. *Cretaceous Research* **30(5)**, 1279–1286. doi:
10.1016/j.cretres.2009.07.003

Montefeltro FC. 2019. The osteoderms of baurusuchid crocodyliforms
(Mesoeucrocodylia, Notosuchia). *Journal of Vertebrate Paleontology* **39(2)**,
e1594242. doi: 10.1080/02724634.2019.1594242

Moody RT, Sutcliffe PJ. 1991. The Cretaceous deposits of the Iullemeden basin of
Niger, central West Africa. *Cretaceous Research* **12(2)**, 137–157. doi:
10.1016/S0195-6671(05)80021-7

Nascimento PM, Zaher H. 2010. A new species of *Baurusuchus* (Crocodyliformes,
Mesoeucrocodylia) from the Upper Cretaceous of Brazil, with the first complete
postcranial skeleton described for the family Baurusuchidae. *Papéis avulsos de*
*Zoologia* **50**, 323–361. doi: 10.1590/S0031-10492010002100001

Nicholl CS, Rio JP, Mannion PD, Delfino M. 2020. A re-examination of the anatomy
and systematics of the tomistomine crocodylians from the Miocene of Italy and
Malta. *Journal of Systematic Palaeontology* **18(22)**, 1853–1889. doi:
10.1080/14772019.2020.1855603

O'Connor PM, Sertich JJ, Stevens NJ, Roberts EM, Gottfried MD, Hieronymus TL,
Jinnah ZA, Ridgely R, Ngasala SE, Temba J. 2010. The evolution of mammal-like
crocodyliforms in the Cretaceous Period of Gondwana. *Nature* **466(7307)**, 748–751.
doi: 10.1038/nature09061

Ortega F, Buscalioni AD, Gasparini Z. 1996. Reinterpretation and new denomination
of *Atacisaurus crassiproratus* (middle Eocene; Issel, France) as cf. *Iberosuchus*
(Crocodylomorpha, Metasuchia). *Geobios* **29(3)**, 353–364. doi: 10.1016/S0016-
6995(96)80037-4

Ósi A, Clark JM, Weishampel DB. 2007. First report on a new basal eusuchian
crocodyliform with multicusped teeth from the Upper Cretaceous (Santonian) of
Hungary. *Neues Jahrbuch für Geologie und Paläontologie Abhandlungen* **243(2)**,
169–177. doi: 10.1127/0077-7749/2007/0243-0169

Ósi A. 2014. The evolution of jaw mechanism and dental function in heterodont
crocodyliforms. *Historical Biology* **26(3)**, 279–414. doi:
10.1080/08912963.2013.777533

Owusu Agyemang PC, Roberts EM, Bussert R, Evans D, Müller J. 2019. U-Pb
detrital zircon constraints on the depositional age and provenance of the dinosaur-
bearing Upper Cretaceous Wadi Milk formation of Sudan. *Cretaceous Research* **97**,
52–72. doi: 10.1016/j.cretres.2019.01.005

Paolillo A, Linares OJ. 2007. Nuevos cocodrilos sebecosuchia del cenozoico
suramericano (Mesosuchia: Crocodylia). *Paleobiologia Neotropical* **3**, 1–25.

Pol D. 2005. Postcranial remains of *Notosuchus terrestris* Woodward (Archosauria:
Crocodyliformes) from the Upper Cretaceous of Patagonia, Argentina. *Ameghiniana*
**42(1)**, 21–38.

Pol D, Leardi JM, Lecuona A, Krause M. 2012. Postcranial anatomy of *Sebecus*
*icaeorhinus* (Crocodyliformes, Sebecidae) from the Eocene of Patagonia. *Journal of*
*Vertebrate Paleontology* **32(2)**, 328–354. doi: 10.1080/02724634.2012.646833

[revised manuscript text omitted]

204x210mm (300 x 300 DPI)

Appendix B

Response to Reviewer Comments for Manuscript

Reviewer 1:

1) Perhaps the most important issue that I raise is related to the α -taxonomy of the peirosaurids from the Kem Kem beds. One of the main goals of the MS is to diagnose the new taxon *Antaeusuchus* and differentiate it from *Hamadasuchus*. However, it is not clear for me which specimens referred to *Hamadasuchus* the authors considered for their analysis. If I understood correctly, the OTU *Hamadasuchus* in their phylogenetic analysis is composed by the fragmentary holotype (MDE C001) and the skull (ROM 52620) described by Larsson & Sues (2007), while in along the comparison section, the authors also compare to a broader sampling of specimens (e.g. BSPG 2005 I 83, ROM 49282, etc). In general, I would be ok with such strategy, in fact, I used a similar *Hamadasuchus* OTU in my own phylogenetic analyses. The problem is that the preserved parts of the holotype of the taxon *Hamadasuchus* and the referred specimen ROM 52620 do not overlap. Also, Ibrahim et al. (2020) suggests that several specimens referred to *Hamadasuchus* present a morphological variation with possible taxonomical implications. The new taxon *Antaeusuchus* can be differentiated from the holotype of *Hamadasuchus* but both cannot be differentiated from the specimen ROM 52620. I would say that the best strategy at this point is to restrict the *Hamadasuchus* OTU to only the holotype of the taxon awaiting further revision of the remaining specimens.

We had already noted in our “Detailed comparisons with *Hamadasuchus rebouli*” section that the OTU for *Hamadasuchus rebouli* is composed of the holotype mandibular fragment (MDEC001) plus the cranial material (ROM 52620) referred by Larsson & Sues (2007), but we have now also clarified this in our Phylogenetic methods section. Although we have changed a few scores from previous studies, this combined OTU has been used in all analyses which include *Hamadasuchus* since the publication of Larsson & Sues (2007). Given the need for a full revision of material assigned to *Hamadasuchus* (beyond the scope of our study) and the focus of our paper (i.e. the description and phylogenetic placement of *Antaeusuchus*), we would prefer to retain *Hamadasuchus* as a combined OTU. Finally, the holotype specimen of *Hamadasuchus* is very incomplete: including it as a separate OTU in a preliminary set of analyses results in far less phylogenetic resolution across our topology. As such, we think that the best solution for this MS is to retain the current *Hamadasuchus* OTU pending revision of that taxon.

2) I detected the lack of explicit definitions of most clades used in the MS, for example figure 7. Which definition of Notosuchia and Ziphosuchia were used? In addition, which definition for Metasuchia was used? I was not able to find any definition in recent papers.

The primary aim of the manuscript is to describe a new specimen and place it into a phylogenetic analysis. We therefore don't agree that it is necessary to provide explicit clade definitions, as it is not the intention of the paper to deal with broader notosuchian taxonomy. Where appropriate, and to discuss the phylogenetic positioning of the new

specimen, we provide the definition we have used for Peirosauridae, within which *Antaeosuchus* is recovered. We believe that the specific clades used throughout the manuscript are standard, widely used terminology used by authors working on this topic, and are not especially pertinent to the key focus of the paper. It is our understanding that explicit clade definitions are not usually required for work such as this. Furthermore, there is only currently a single proposed phylogenetic definition for Notosuchia, Ziphosuchia, and Metasuchia, and these have been in usage for most of the last two decades.

3) Although it is not particularly relevant, the revised phylogeny presented by the MS is not the largest notosuchian-focused character-taxon matrices yet to be compiled.

The text states that the matrix is “one of the largest” rather than “the largest” which we believe to be true given the high degree of character and taxon sampling, and our particular focus on including more peirosaurid taxa. As such, we have made no changes in response to this comment.

4) We have proposed a phylogenetic definition for Notosuchia in our paper (Ruiz, et al. 2021), perhaps include it on the Systematic Palaeontology section.

As mentioned above, the aim of this paper is to present a new crocodylomorph specimen, not to review broader notosuchian taxonomy. We do not believe the inclusion of phylogenetic definitions of all clades relating to notosuchians to be standard procedure in work such as this. Furthermore, the definition of Notosuchia in that paper is identical to the original phylogenetic definition provided by Sereno et al. (2001).

5) In relation to the anatomy of *Antaeosuchus*, I suggest the authors to take a look on Pinheiro’s et al. (2020, Plos One) description of the enamel of *Roxochampsa*.

The paper suggested by the reviewer describes a Late Cretaceous Brazilian notosuchian, which was recovered outside of Peirosauridae. In our manuscript, detailed comparisons with other crocodylomorphs are restricted to Peirosauridae given the position recovered by *Antaeosuchus* in our phylogenetic analyses. As such, we feel it is unnecessary to include text regarding *Roxochampsa* within the manuscript based on its recovery in a different clade and lack of spatiotemporal overlap. If the reviewer is suggesting potential similarities between the two taxa, we note that the enamel in *Roxochampsa* differs from that of *Antaeosuchus* in several aspects, most notably in that the apicobasal enamel ridges are themselves crenulated along their length. As such, we have not added in comparisons to this taxon.

6) Which are the parameters used during sectorial searches, drift and tree fusing?

We have modified the text to confirm the exact parameters used during our analyses. These sections have been highlighted in the marked-up draft.

7) I did not understand what the authors mean when they say that *Uberabasuchus* can be used as a proxy for *Peirosaurus*?

We described in the preceding sentence how *Uberabasuchus terrificus* has been consistently recovered as a close relative of *Peirosaurus torminni*, with some authors regarding the latter as a junior synonym of the former (e.g. Larsson & Sues 2007; Martinelli et al. 2012). As such, we think the existing text is clear and explicit in terms of what we mean, i.e. “Although *Peirosaurus torminni* is not included in our data matrix, *Uberabasuchus terrificus* has been consistently recovered as a close relative, with some authors regarding the latter as a junior synonym of the former (e.g. Larsson & Sues 2007; Martinelli et al. 2012). As such, we regard the *Uberabasuchus* OTU as a proxy for *Peirosaurus* in terms of identifying Peirosauridae.”

8) In the discussion about the multicusped teeth described by Larsson & Sidor (1999), I suggest the authors to also take a look on the papers by Montefeltro et al. (2009) and Pinheiro et al. (2021, *Coronelsuchus*).

The paper by Montefeltro et al. (2009) describes six multicuspid teeth from the Upper Cretaceous of Brazil. Though similar in broad morphology to those described from Morocco, the authors note that they are “not related to two unnamed forms” from the Kem Kem beds. Pinheiro et al. (2021) also describe a notosuchian with heterodont, multicuspid dentition from the Bauru Basin of Brazil. Although both papers describe multicuspid teeth, this is not a particularly unusual morphology amongst notosuchians. The condition is more widespread than the papers mentioned by the reviewer and so it is unclear why these two precisely have been chosen. We refrain from including these specific examples in this work as our review focuses on the Gondwanan record outside of South America.

9) The reference Evans et al. 2014 cited in the text is not listed. There is a reference cited as Montefeltro et al. (2019). I guess it is Montefeltro (2019, JVP) or Montefeltro et al. (2020, J. Anato).

We have added the appropriate citations to the manuscript. The latter has been corrected to Montefeltro et al. (2020) in the References.

Reviewer 2:

1) The authors used two different protocols for their phylogenetic analyses, one employing equally weighting of characters, and the other employed the extend implied weighting protocol. I do think it is interesting to see the results using two alternative methods. However, if two alternative methods are used in the paper, I think that the authors should then explain if they prefer any of the methods employed. If they do prefer one of the two methods, justify. If no, just mention that you are using different methods because there is no study so far that says that we should ‘definitely’ opt for one instead of

the other. In this context, sentence like “we applied extended implied weighting to notosuchians for the first time’ are not very relevant, especially when the authors use the phylogeny obtained from their analysis using equal weighting in their Figure 11. In this case, I think there should be a justification for the reason why you selected the equal weighting analysis for this figure.

We are pleased to hear that the reviewer is interested to see the results of the alternative methods in this paper. Goloboff (2014; 2017) described the potential benefits of using extended implied weighting on morphological datasets, and several recent neosuchian-focused studies (e.g. Groh et al. 2020; Rio et al. 2020; Rio and Mannion 2021) also show that analyses run using extended implied weighting (EIW) score higher in measures of phylogenetic accuracy. We do not “prefer” any particular method, nor do we analyse the benefits and pitfalls of each as it is beyond the remit of this paper. We do, however, think that it is important to mention that the method has not yet been used on notosuchians given the discrepancies seen in tree topologies between both methods. Given that we do not have a preferred analysis, the method used to illustrate the tree in Figure 11 is not important: given that our analyses had broadly congruent results we merely chose to represent one tree as a time calibrated figure given that both are already figured as cladograms.

2. The authors did a good job in providing comparisons between the new species with other peirosaurids. Also, by the end of this section, they list the differences the differences between *Antaeusuchus* and the other peirosaurids from the Kem Kem, *Hamadasuchus*. The authors mention that one of the differences is that specimens assigned to *Antaeusuchus* are much larger than species assigned to *Hamadasuchus*. Thus, I think that an interesting addition to their study would be to try ruling out the possibilities that the differences between *Antaeusuchus* and *Hamadasuchus* are not related to ontogeny – i.e. that individuals of *Antaeusuchus* do not correspond to larger individuals of *Hamadasuchus*. I’m not sure it is possible to check this for all the different characteristics they mentioned based on the differences observed in extant crocodylians – but anyway, this might strength their argument to separate the two species.

We agree with the reviewer that this would be an interesting aspect to include in the study. Griffin et al. (2020) suggested that the best measure of ontogeny in crocodylomorphs can be ascertained from long bone histology and neurocentral fusion in vertebrae, neither of which are applicable in this paper given the preserved material. The sister taxon to *Antaeusuchus*, and therefore potentially the best proxy for evaluating ontogenetic changes in the new specimens, *Hamadasuchus rebouli* is known from several specimens that are described by Larsson & Sues (2007) as representing an ontogenetic series. Within the paper they refer to several morphological differences that are evident between the various growth stages. These almost entirely relate to the cranium (e.g. the reduction of a sagittal crest on the frontoparietal suture, the reduction of medial rims of the supratemporal fenestra, and the reduction of a crest on the posterodorsal surface of the quadrate), and thus are not helpful in assessing the growth stage of the *Antaeusuchus* mandible. The final morphological feature mentioned by Larsson & Sues (2007) relates to shape changes in the posterior dentary teeth, and describes a change from more gracile, labiolingually compressed

morphology to more robust, wider teeth in relatively mature individuals. The teeth in *Antaeusuchus* are actually closer in diameter to the less mature specimens of *Hamadasuchus*, going some way to ruling out that the former is a more mature than the latter. Though we understand the reviewer's concerns about using size as a justification for the erection of a new species, *Antaeusuchus* is significantly larger than even the largest of all known specimens referred to *Hamadasuchus*, which is already known from an ontogenetic series Larsson & Sues (2007). We agree that as a sole comparison, size would be an insufficient reason to justify a new species, but as part of a large, unique combination of morphological features, we believe it to be a valid difference. A paragraph explaining this has been added to the text.

3. I have mixed feelings about the last section of the discussion, which brings a revision of the notosuchians outside South America. Whereas I think it is interesting to see this kind of information compiled in a single study, I do not think that the authors used the information already available in the literature in order to provide any new insight on the evolution of African notosuchians. For example, the section dealing with the presence of *Razanandrongo* in the Middle Jurassic of Madagascar brings no novel information that is worth being included in the discussion section of this manuscript. In sum, there is not much of new insights or new perspectives in all the sections of this part of the discussion that would justify the inclusion of this part of the manuscript together with the description of the new taxon and discussions on the phylogenetic analysis of peirosaurids. For example, section 6.3.5 would better fit as part of an introduction of a manuscript on the fossil record of notosuchians.

We believe that a revision of Gondwanan notosuchians outside of South America is an integral part of this manuscript. We provide the first ever comprehensive overview of all notosuchians from this region, several of which we reidentify. Our reappraisal deals with multiple putative remains, the assignment of which will significantly affect the outcome of macroevolutionary studies for notosuchians outside of South America. In terms of biogeography, the notosuchian record from the Arabian Peninsula is removed. Furthermore, the review provides updated stratigraphic information which is often missed by many authors. As such, we strongly disagree with the reviewer and have retained this section.

Comments from Reviewer 2 provided in the attached PDF:

1. Please add dates in millions of years - this will facilitate readers to locate themselves in geologic time.

We have included all relevant dates within the abstract as well as at their first mention in the main body of text.

2. In the way that it is written, it seems that an analysis of the fossil record is what indicates that *Miadanosuchus* might be a peirosaurid. Is that really what is intended to be said here? My view is that only a phylogenetic analysis can indicate the phylogenetic affinities of a taxon.

The wording has been altered to reflect the role of phylogenetic analyses in the designation of *Miadanosuchus* as a peirosaurid as opposed to solely the reappraisal of the non-South American notosuchian record.

3. Usually, I like to see a final sentence wrapping up the main point of the manuscript, and also bringing some directions for future studies. The authors might want to consider to include something like this to the abstract.

We feel that the abstract as a whole does a sufficient job of summarising the main points of the manuscript and so we refrain from adding a further sentence, especially given that we were already at the word limit in our original submission. Furthermore, our entire abstract summarises our study – having a final sentence that summarises the abstract seems somewhat redundant. A future studies section is not needed and we have already noted several things that need doing in the existing text (e.g. revision of *Hamadasuchus*).

4. You can also cite Ruiz et al. 2020 here.

Requested citation has been added to the text.

5. What do you mean by apparent diversity here? Number of known species? Please specify.

Text has been clarified to indicate that we are referring to the raw number of species.

6. Could you indicate where each wave starts and ends?

Information regarding the positioning of the waves has been added.

7. Is it possible that the suture extends further posteriorly? Are the splenials overlapping the dentaries in this region?

We have not amended this section as it simply describes the dentary suture visible in dorsal view. The suture is seen to extend to the 8th tooth in specimen PV R36829, and its shape can also be inferred in PV R36874. On the ventral surface the dentary extends to the 7th/8th teeth. The exposure of the suture in medial view in PV R36874 indicates that the suture did not extend further posteriorly due to intrusion of the splenials between the dentaries.

8. These two processes, the one forming the dorsal margin of the mandibular fenestra and the one dorsal to it, are they really processes or is it just the case that the surangular is overlapping the dentary in this region?

Although the surangular does marginally overlap the dentary as scored in character 366, an oblique suture can be seen in both lateral and medial view indicating that these are distinct, separate processes rather than superficial ones formed just by a surangular process overlapping the dentary.

9. It would be great to have these processes labelled in the figure.

As requested, the posterior processes have been labelled in figures 2 and 3.

10. However, shouldn't it be three processes, as you consider as the process ventral to the mandibular fenestra as a posterior process as well?

The addition of this text clarifies the specific processes we are referring to. The “third” process is described later in the text and is not referred to here as it cannot be considered “major”, forming only a short protrusion.

11. I suggest changing to: and occupy approximately 38% of the anteroposterior symphyseal length on the dorsal surface of the mandible.

The text has been changed as requested.

12. On the ventral surface.

The text has been changed as requested.

13. It is better to cite the work of Lordansky, cited in Larsson & Sues, than to cite the latter. Should then change the sentence slightly - likely homologous the foramen intermandibularis oralis of living crocodylians.

We have cited Lordansky (1973), and have changed the text as requested.

14. I couldn't see it on the figures.

A label has been added to figures 2 and 3.

15. For the angular and some of the other bones previously described, I missed a more general description of each bone, detailing the general shape of the bone and indicating the bones that it contacts.

We have added in several additional descriptions of general morphology; however, we feel that we have already been fairly consistent with describing the contact between bones.

16. I think that if you are using a whole new section for the description of the mandibular fenestra, you could provide some more details, even if it is incomplete. For example, is it possible to add some relative measurements to the description of this structure?

We have added in an additional comment although refrain from too many descriptions given the fragmentary nature of the fenestra.

17. Which are the parameters used during sectorial searches, drift and tree fusing?

Precise parameters have been included.

18. Not necessary to include this. Just cite Goloboff.

We have kept this text as we feel that it provides an overview of the benefits of using extended implied weighting on a dataset, especially as the method has not previously been applied to notosuchian crocodylomorphs. We do not believe that the inclusion of the text detracts from the manuscript in any way.

19. A better approach would be to try different values of 'k' instead of adopting the values from previous studies.

We selected these specific values as previous studies (e.g. Goloboff et al. 2017) have indicated that lower values of k can be excessive in downweighting putative homoplastic characters. Despite extended implied weighting never having been used before in a notosuchian-focused matrix, such effects can be seen in datasets for neosuchian crocodylomorphs (e.g. Rio & Mannion 2021) in which low k -values consistently produce trees that score poorly in measures of phylogenetic accuracy. Whilst attempting to avoid lower values of k , we still undertake multiple analyses to determine the possible impact of different k -values.

20. If a taxon is included in the matrix, I think that it also should be included in the analyses. So, you should either show the topology of the strict consensus with these unstable taxa, or simply generate a reduced consensus tree using the prunnelsen command (Goloboff & Szumik, 2015 or 2016). I think that the last option is more appropriate because it shows where these problematic taxa are floating on your tree.

We refrain from changing the analysis on the basis that if these taxa were not included within the matrix at all then the issue would not have been raised. We are unaware of a similar paper which includes all described species of notosuchian. We therefore follow the recommendations of Pol et al. (2014) to exclude this taxa.

21. Which are the parameters used during sectorial searches, drift and tree fusing?

Precise parameters have been included.

22. This should be tree length - not branch length.

Changed as requested.

23. I think it will be easier to follow the results if you already mention the phylogenetic definition here.

The definition is written out later on in a separate section of text relating to peirosaurids, and therefore we refrain from including it here. We feel that writing the full definition of clades in this section of the manuscript would potentially make the text harder to follow, especially given that these are generally widely used clade terms in notosuchian-focused literature.

24. How did you recover this? You used the prunnelsen command or you removed *Gasparinisuchus* and ran a new analysis? Please indicate that.

Gasparinisuchus was removed from the analysis in the agreement subtree.

25. I'm not very convinced that you can use this characteristic for comparisons between *Antaeusuchus* and other taxa, because it is not entirely preserved in the former.

We still include this morphology in our comparisons, as despite the fenestra being incomplete in either single specimen, the majority of its border is preserved across both NHMUK PV R36829 and R36874. In both specimens the preserved margins indicate that the opening is clearly larger than in *Barrosasuchus* with which it is compared.

26. I think that in this case, you should score your taxon with '?' rather than 1 and 2.

We have chosen not to alter the score to '?' as the splenial is clearly elongate and would be scored for either state 1 or 2 if complete. We feel in this instance it is better to score for either scenario rather than to exclude morphological information. Furthermore, the character construction needs to be addressed given the large gap between states 1 and 2, however, this is beyond the scope of this paper.

27. Don't you think that this polymorphism could also exist in specimens assigned to *Hamadasuchus*?

There is definitely variation amongst taxa referred to *Hamadasuchus*, as discussed in our comparisons section. Given the character state boundaries for character 77, we felt it best to provide a detailed account of the measurements in each specimen. A comprehensive review of notosuchian characters would help to produce definite scores for many taxa with polymorphic scoring; However, this is not the aim of the paper.

28. That is not necessarily true - you should remove this sentence.

We have removed the sentence as requested.